



# 1 MIS 5e sea-level history along the Pacific Coast of North America

Daniel R. Muhs[1]
[1]U.S. Geological Survey, MS 980, Box 25046, Federal Center, Denver, Colorado 80225 USA
*Correspondence to*: Daniel R. Muhs (dmuhs@usgs.gov)
**Abstract.** The primary last interglacial, marine isotope substage (MIS) 5e records on the Pacific Coast of North America, from
Washington (USA) to Baja California Sur (Mexico), are found in the deposits of erosional marine terraces. Warmer coasts
along the southern Golfo de California host both erosional marine terraces and constructional coral reef terraces. Because the
northern part of the region is tectonically active, MIS 5e terrace elevations vary considerably, from a few meters above sea
level to as much as 70 m above sea level. The primary paleo-sea level indicator is the shoreline angle, the junction of the
wave-cut platform with the former sea cliff, which forms very close to mean sea level. Most areas on the Pacific Coast of
North America have experienced uplift since MIS 5e time, but the rate of uplift varies substantially as a function of tectonic
setting. Chronology in most places is based on uranium-series ages of the solitary coral *Balanophyllia elegans* (erosional
terraces) or the colonial corals *Porites* and *Pocillopora* (constructional reefs). In areas lacking corals, correlation to MIS 5e
can sometimes be accomplished using amino acid ratios of fossil mollusks, compared to similar ratios in mollusks that also
host dated corals. U-series analyses of corals that have experienced largely closed-system histories range from ~124 to ~118
ka, in good agreement with ages from MIS 5e reef terraces elsewhere in the world. There is no geomorphic, stratigraphic, or
geochronology evidence for more than one high-sea stand during MIS 5e on the Pacific Coast of North America. However,
in areas of low uplift rate, the outer parts of MIS 5e terraces apparently were re-occupied by the high-sea stand at ~100 ka
(MIS 5c), evident from mixes of coral ages and mixes of molluscan faunas with differing thermal aspects. This sequence of
events took place because glacial isostatic adjustment processes acting on North America resulted in regional high-sea stands
at ~100 ka and ~80 ka that were higher than is the case in far-field regions, distant from large continental ice sheets. During
MIS 5e time, sea surface temperatures (SST) off the Pacific Coast of North America were higher than is the case at present,
evident from extralimital southern species of mollusks found in dated deposits. Apparently no wholesale shifts in faunal
provinces took place, but in MIS 5e time, some species of bivalves and gastropods lived hundreds of kilometers north of their
present northern limits, in good agreement with SST estimates derived from foraminiferal records and alkenone-based
reconstructions in deep-sea cores. Because many areas of the Pacific Coast of North America have been active tectonically
for much or all of the Quaternary, many earlier interglacial periods are recorded as uplifted, higher elevation terraces. In
addition, from southern Oregon to northern Baja California, there are U-series-dated corals from marine terraces that formed
~80 ka, during MIS 5a. In contrast to MIS 5e, these terrace deposits host molluscan faunas that contain extralimital northern
species, indicating cooler SST at the end of MIS 5. Here I present a standardized database of MIS 5e sea-level indicators along





the Pacific Coast of North America and the corresponding dated samples. The database is available in Muhs (2021)
[https://doi.org/10.5281/zenodo.5557355].

## 1 Introduction

Because of the prospect of future sea-level rise, there has been an increasing interest in past, but geologically recent times
of higher than present sea level. One of the best studied of these is the last interglacial complex, recognized in terrestrial
geologic records as the Sangamon (North America) or Eemian (Europe) periods. Within the deep-sea sediment core record,
Arrhenius (1952) initiated the widely accepted practice of numbering Quaternary interglacial and glacial stages, which was
encouraged with the pioneering work on oxygen isotopes in such cores by Emiliani (1955). Interglacial periods have odd
numbers and glacial periods have even numbers. Thus, the last interglacial complex (*sensu lato*) in deep-sea cores is known
as marine isotope stage (MIS) 5. Shackleton (1969) recognized five major substages of the MIS 5 complex (5e, 5d, 5c, 5b,
5a, from oldest to youngest), and those substages are now widely recognized by marine stratigraphers and paleoclimatologists.
Another nomenclature suggested by Martinson et al. (1987) is followed by some investigators, with the peaks of these
substages referred to as "events" MIS 5.5, 5.4, 5.3, 5.2, and 5.1, from oldest to youngest. MIS 5e or 5.5 is considered to be the
period of peak global warmth and minimal global ice of the late Quaternary (see review in Murray-Wallace and Woodroffe,
45   2014).
The Pacific Coast of North America contains a rich record of Quaternary sea level history, particularly the peak of the last
interglacial period, MIS 5e, generally considered to date from ~130 ka to ~115 ka. Part of the richness of this sea level record
is due to the tectonic setting of North America (Fig. 1). Most of the continent is situated on the North America lithospheric
plate. However, Baja California and part of westernmost California are both located on the Pacific plate, and the southern part
of Central America is on the Caribbean plate. The boundaries between the two major (Pacific and North America) and smaller
lithospheric plates are the tectonic controls on the crustal blocks that form the Pacific Coast of North America. In southwestern
Canada and the northwestern USA, the Cascadia subduction zone occurs where the southeast-moving Juan de Fuca and Gorda
plates meet the North America plate (Fig. 2). Farther south, from northern California to the Golfo de California, the dominant
structural control is the San Andreas Fault, a major right-lateral (dextral) strike-slip system, with many smaller, subparallel
faults associated with it. Still farther south, at the head of the Golfo de California, the structural style changes again, with the
boundary between the Pacific and North America plates taking the form of a spreading center, the northernmost part of the
East Pacific Rise (Fig. 2). Finally, the structural style changes to the south once more, back to a subduction zone, in southern
Mexico and Central America. Here the Cocos plate is being subducted under the North America plate (in the northern part)
and under the Caribbean plate in the southern part (Fig. 1).
The importance of tectonic setting for studies of past shorelines, such as that of MIS 5e, is due to its influence on vertical
movement of coastal crustal blocks. In collisional zones, such as the Cascadia subduction zone, it could be expected that some
vertical movement might be found in the crust of the overriding plate. Indeed, a classical study by Uyeda and Kanamori (1979)





proposed that where the dip of the subducting plate is shallow, rapid uplift should be seen in the overriding plate. However,
detailed studies of marine terraces in northern California and Oregon by H.M. Kelsey and his colleagues (Kelsey, 1990;
McInelly and Kelsey, 1990; Kelsey and Bockheim, 1994; Kelsey et al., 1994, 1996; Polenz and Kelsey, 1999) have shown
convincingly that it is actually local structures (faults and folds) *within* the upper plate that control the rates of marine terrace
uplift seen along much of the Cascadia subduction zone. Farther south, within the San Andreas Fault zone, rates of uplift are
highly variable (see summary in Muhs et al., 2014b). Along much of the coast bordering this fault zone, uplift rates are modest,
likely (though not yet proven) because movements along faults that have a predominantly strike-slip (horizontal) sense of
movement have a small vertical component. Exceptions to this occur where there are restraining bends in these faults, the
most famous of which is the "big bend" area of the San Andreas Fault zone (Fig. 2). Here, crustal compression results in
extremely high rates of uplift. Away from the zone of maximum uplift south of the big bend in the San Andreas Fault, Shaw
and Suppe (1994) proposed that uplift of the Santa Cruz Island and Anacapa Island shelf area of southern California is due to
movement on an underlying, blind thrust fault. High rates of uplift can also be found on coastlines adjacent to triple junctions,
such as the Mendocino triple junction (Fig. 2) and the Panama triple junction ("PTJ" in Fig. 1). Along coastlines bordering a
spreading-center plate boundary, such as that in the Golfo de California, crustal blocks are moving away from each other and
accommodation space is increasing, so uplift rates are not expected to be particularly high. This simplified picture is to a great
extent borne out by field studies (e.g., Ortlieb, 1991), although local structures can again play a role in generating uplift over
limited parts of such a coastline. Uplift in Central America is rapid in places, due to subduction of seamounts on the Cocos
and Nazca plates.
Marine terraces along the Pacific Coast of North America have been studied for more than a century. Lawson (1893)
considered that emergent terraces formed by episodic (and presumably rapid) uplift, what would now be referred to as
coseismic uplift. Smith (1900), studying terraces on the California islands, concurred with this hypothesis, reasoning that
episodic uplift must alternate with periods of "comparative quiescence." Interestingly, the concept of episodic rapid uplift is
now known to have validity for some parts of the Pacific Coast, in diverse tectonic settings (see discussion below on Holocene
shorelines).
Grant and Gale (1931) also considered emergent terraces to have a tectonic origin, but also pointed out the possibility of a
eustatic component. It was Davis (1933), however, studying marine terraces in the Malibu, California region, who was likely
the first to point out explicitly that although uplift was obviously a factor in the formation of a flight of marine terraces, a
eustatic component was important as well. Davis (1933) considered that uplift rates were likely to have geographic variability,
but he noted that eustatic records ought to be the same everywhere. Despite the publication of this important paper, there was
a return to the idea of terraces being dominantly of a tectonic origin in later studies by Putnam (1942), Woodring et al. (1946),
and Upson (1951). Woodring et al. (1946) thought that eustatic effects were either obliterated or obscured in the geologic
record of marine terraces. Although Upson (1951) considered that terraces were formed principally by episodic uplift, he
recognized that there were problems with this explanation and thought that a eustatic component was present.



Interestingly, it was a master's degree thesis at the University of California at Berkeley that articulated our current concepts
of marine terraces, uplift, and sea level clearly for the first time. Alexander (1953), working on both marine and stream terraces
in the Capitola-Watsonville area of central California, measured the maximum elevations of marine terraces and the tops of
stream-fill terraces, noting their similar elevations, and reasoning that they must have a common, eustatic control. He also
noted that in between times of stream terrace formation, there were episodes of valley cutting, which indicated periods of
eustatically lowered sea level, during glacial periods. On the other hand, multiple marine terraces indicated long-term tectonic
uplift. He (Alexander, 1953, p. 36) concluded that "Thus, the marine terraces of the Capitola-Watsonville area are regarded
as having originated under conditions of a slowly and continuously rising coast against which occurred at least three complete
cycles of eustatic changes in sea level." This is a remarkable conclusion, reached before any modern methods of
geochronology were in common use, and based only on sound field mapping, elevation measurements, and geomorphic
reasoning. It was this concept, along with uranium-series geochronology, that allowed Broecker et al. (1968), Mesolella et al.
(1969), and Veeh and Chappell (1970) to infer that coral reef terraces on the uplifting coasts of Barbados and New Guinea
recorded interglacial periods that supported the Milankovitch or orbital theory of climate change. In California, Alexander's
(1953) concept was accepted explicitly or implicitly by subsequent workers in the following decades (e.g., Vedder and Norris,
1963; Birkeland, 1972; Bradley and Griggs, 1976; Wehmiller et al., 1977a), and his contribution is now recognized in one of
the leading textbooks on geomorphology (Anderson and Anderson, 2010).
Dating of marine terraces on the Pacific Coast of North America had a development similar to that for other coastlines.
Early use of uranium-series (U-series) analyses of corals was reported by Veeh and Valentine (1967), Valentine and Veeh
(1969), and Ku and Kern (1974). In these investigations and most subsequent studies, the taxon analyzed is the solitary coral
*Balanophyllia elegans* (Gerrodette, 1979), which is by far the most common coral found in Oregon and California marine
terrace deposits. These early studies permitted an interpretation that low-elevation terraces at Cayucos, San Nicolas Island,
and Point Loma could all date to MIS 5e. Other studies attempted U-series analyses of fossil mollusks (e.g., Bradley and
Addicott, 1968; Szabo and Rosholt, 1969; Szabo and Vedder, 1971), but a seminal study by Kaufman et al. (1971), with
extensive data from California terraces, showed that mollusks are inappropriate materials for U-series geochronology.
A new development in geochronology, however, brought mollusks back to the forefront in dating marine terraces on the
Pacific Coast. Using the Cayucos, San Nicolas Island, and Point Loma U-series coral ages as calibration points, Wehmiller et
al. (1977a), Wehmiller (1982), and Kennedy et al. (1982) showed that terraces from Baja California Sur to Oregon could be
correlated to MIS 5e on the basis of amino acid ratios in fossil mollusks, a profound finding that demonstrated the extensive
nature of the last interglacial record on the Pacific Coast of North America. In addition, these studies also showed that uplift
rates on the Pacific Coast are variable, overturning a long-held concept that the lowest marine terrace is everywhere of the
same age. Indeed, terraces estimated to be as young as ~50 ka were found in areas of high uplift rate.
Concerted efforts to find corals yielded more U-series ages of marine terraces. Rockwell et al. (1989) mapped 14 terraces
on Punta Banda, Baja California, the lowest 3 of which have shoreline angle elevations of 15-17 m, 22 m, and 27-43 m. The
1st or "Lighthouse" terrace has U-series ages (by alpha spectrometry) of corals and hydrocorals of ~80 ka, the 2nd terrace is



undated, and the 3rd or "Sea Cave" terrace has ages of ~120 ka.  This was the first study on the Pacific Coast to provide
definitive geochronologic evidence of both MIS 5e and MIS 5a (as well as a likely MIS 5c at ~22 m) terraces.  Muhs et al.
(1990, 1992, 1994) reported additional U-series ages, again by alpha spectrometry, for MIS 5e terraces at Cayucos, Point San
Luis, San Nicolas Island, San Clemente Island, and Point Loma (all in California), and Punta Banda, Isla Guadalupe, and Cabo
Pulmo (in Baja California and Baja California Sur).  Terraces dating to MIS 5a were reported from Coquille Point, near
Bandon, Oregon; Point Arena, San Nicolas Island, and Point Loma (all in California); and Punta Banda (Baja California).

The development of U-series dating of corals by thermal ionization mass spectrometry (TIMS) led to a new level of

complexity in the understanding of the Pacific Coast marine terrace record.  Stein et al. (1991) redated corals from the Cayucos
and Point Loma areas, confirming that fossils dating to MIS 5e were present, but also showing the possibility that some corals
dated to MIS 5c (~100 ka).  Muhs et al. (2002a) confirmed these results for both Cayucos and Point Loma.  Kennedy et al.
(1982), in their amino acid study along the Pacific Coast, reported that terraces correlated to MIS 5a had molluscan faunas
with cool-water aspects, whereas those correlated to MIS 5e hosted molluscan faunas with warm-water aspects.  Cool-water
faunas were confirmed with corals dated to ~80 ka using TIMS by Muhs et al. (2006) in a later study, in partial support of
Kennedy et al. (1982).  However, Muhs et al. (2002a) showed that the terraces at Cayucos and Point Loma, containing both
MIS 5e and MIS 5c corals, hosted molluscan faunas with a mix of both warm-water taxa (thought to date from MIS 5e) and
cool-water taxa (thought to date from MIS 5c).  This idea was explored in more detail on San Nicolas Island, where the lowest
three terraces (1, 2b, and 2a, in ascending elevation order) were mapped in detail, terraces elevations were measured precisely
with differential GPS methods, corals from all three terraces were dated with TIMS, and the faunas were characterized (Muhs
et al., 2012).  Terrace 1 dates to ~80 ka and hosts a cool-water fauna, terrace 2b has both 100 ka and 120 ka corals and hosts a
mix of cool-water and warm-water taxa, and terrace 2a has only ~120 ka corals, no cool-water taxa, but several warm-water
taxa.  This finding raised the possibility that the MIS 5c high-sea stand in this region had a paleo-sea level elevation higher
than what would have been inferred from the classic records on Barbados and New Guinea, and that this high stand overtook
at least the outer part of the MIS 5e terrace, reworking and mixing its fossils (with warm-water taxa) with shells dating to MIS
5c (with cool-water taxa).  Subsequent studies have shown that other terraces dating to MIS 5e (by TIMS U-series on corals)
or correlated to MIS 5e (by amino acids on mollusks) also contain mixes of warm-water and cool-water taxa (Muhs et al.,
2014a, 2014b; Muhs and Groves, 2018).

The main aim of this paper is to serve as a description notice accompanying a standardized database of MIS 5e sea-level

indicators compiled following the WALIS template (Rovere et al., 2020).  From the published papers in the area of interest, I
extracted sea level indicators and standardized the quantification of their elevation and indicative meaning (Shennan, 1982;
Shennan et al., 2015; Rovere et al., 2016), along with appropriate metadata.  Each sea level indicator was then associated with
one or more samples, dated with U-series of amino acid racemization (AAR) methods, that were also added to the database.
In some cases, U-series dated samples were already present in the WALIS database from the compilation of Chutcharavan and
Dutton, 2020).



## 2 Sea level indicators

As pointed out by Rovere et al. (2016), critical to reconstructing past sea level during MIS 5e (or any past high-sea stand, for that matter) is an accurate assessment of paleo-sea level indicators (Table 1). For the vast majority of MIS 5e geomorphic records along the Pacific Coast of North America, the best relative sea level (RSL) indicator is what is called the *shoreline angle*, a term that goes back to the classic study of terraces in the Malibu, California area by Davis (1933). The shoreline angle is the junction of the marine platform (or "wave cut bench"), formed in the surf zone and the sea cliff, when viewed in cross section (Fig. 3a). Davis (1933) and virtually all investigators who have followed him have generally regarded the shoreline angle as the best overall RSL, because it is considered to form at or near sea level. Kelsey (2015) points out that depending on bedrock type, structures within the local bedrock, orientation of the coast with respect to wave exposure, and other factors, shore-parallel variability of the shoreline angle elevation on modern coastlines can vary by as much as 1-4 m. In the San Diego area, however, measurements made by Kern (1977) indicate that modern shoreline angles typically form within a meter of modern sea level. Whether shoreline angles on the Pacific Coast form closest to mean sea level or high-tide level is probably not known with any certainty. In any case, however, the range of variability of shoreline angle elevations noted by Kelsey (2015) is typically greater than the mean tidal range. In southern and central California, from San Diego to San Francisco Bay, mean tidal range is typically only 1.1 to 1.2 m; in northern California, it increases to about 1.2 to 1.5 m; and in Oregon, it is 1.6 to 1.8 m (data from: https://tidesandcurrents.noaa.gov/tide_predictions.html). In most places that I have studied along the Pacific Coast, marine platforms or wave-cut benches are typically only visible at low tide (Fig. 4) and are not visible at high tide. In most of California, therefore, with a mean tidal range (low to high tide) of only about a meter, these observations suggest that Kern's (1977) observations have general validity, and shoreline angles approximate mean sea level.

For the field geomorphologist studying marine terraces, a much greater challenge lies in mapping shorelines accurately and finding good exposures of ancient shoreline angles. After terrace emergence, the wave-cut platform and the marine sediments covering it become the locus of deposition of terrestrial deposits, including alluvium, colluvium, and eolian sand (Fig. 5). Such deposits obscure the precise location of the *inner edge* of a marine terrace. The term "inner edge" is often used interchangeably with the term shoreline angle, but here it is meant to express the spatial extent of a shoreline, i.e., viewed planimetrically, in a shore-parallel sense. Put another way, it is the mapped expression of where the shoreline angle is situated, marking the former junction of land and sea. Terrestrial deposits that cover inner edges of marine terraces not only make mapping of a given terrace difficult, but also can be extensive enough that they cover two or more discrete terraces. Alluvial and eolian deposition can sometimes generate a rather smooth surface that gives the impression of being an actual marine platform surface, which may in reality be many meters below (Fig. 3b). In the example shown in Figure 3b, the unwary researcher might assume that there is only one terrace here, and also could easily assume that the "apparent inner edge" is where the actual shoreline angle is situated, when in fact it is seaward of this and at a much lower elevation. In the database,





the upper and lower limit of the indicative range for a shoreline angle were set as the Mean Lower Low Water (MLLW) and
the Mean Higher High Water (MHHW) reported for the nearest NOAA tide station.

Because most Pacific Coast marine terraces develop on a high-energy, erosive coastline, biological indicators of RSL are
rare. Typically, marine fossils are found in a poorly sorted mix of sand and gravel. As a consequence, the fossils in marine
terrace deposits, even those near the former shoreline, have been transported there by waves, sometimes from depths of 20 m
or more. Exceptions to this, while uncommon, do occur and most often take the form of rock-boring mollusks in growth
position, particularly bivalves in the Pholadidae family. A good example of this is the species *Penitella penita*. This taxon
typically occurs in the mid-intertidal zone, based on modern specimen collections in the Santa Barbara Museum of Natural
History (P. Valentich-Scott, written communication, March 2020). Only rarely is *P. penita* found below depths of ~10 m
(Coan et al., 2000). Thus, if fossil *P. penita* is found in growth position in bored holes of a wave-cut bench (Fig. 6), it is likely
that one is within 10 m of paleo-sea level. While this criterion is not as specific an RSL indicator as a shoreline angle, it is
often a complementary tool for paleo-sea level. Other species of bivalves can potentially serve as paleo-sea level indicators if
they are articulated, a characteristic not possible with gastropods. For example, the large bivalve *Saxidomus nuttalli*, which is
presently found from northern California to Baja California Sur, typically lives in muddy sediments within the intertidal zone
to ~10 m depth (Coan et al., 2000), a range similar to *Penitella penita*. Thus, if an articulated fossil specimen of *S. nuttalli* is
found, it is *possible* that it is close to where it was situated when it was living, because wave transport commonly will
disarticulate shells. However, *S. nuttalli* is not a rock-boring mollusk, so without occurrence in a hole that it has bored, one
can never be certain, even with articulated shells, that one is near the position where the specimen lived.
Farther south, along both shores of the Golfo de California and the Pacific Coast of mainland Mexico and Central America,
ocean water temperatures are higher than farther north, and hermatypic (reef-building) corals are found (Fig. 7). Although
hermatypic corals can be found throughout much of this region, true coral reefs are far less common. For example, within the
Golfo de California, although corals can be found along almost all of the Baja California coast and much of the Sonoran coast,
true coral reefs have been documented only at a few localities. The region from the upper Golfo de California to Panama does,
however, host a surprising diversity of coral species (Reyes-Bonilla and Lópéz-Pérez, 1998; Glynn and Ault, 2000; Glynn et
al., 2017; Toth et al., 2017). Some of the most important genera are *Porites* (7 species), *Pocillopora* (6 species), *Psammocora*
(4 species), and *Pavona* (5 species). *Porites panamensis* (formerly *P. californica* in some studies) is found from the upper
Golfo south to Panama, but also has a disjunct distribution, with colonies of this taxon also found in Bahía Magdalena, on the
Pacific coast of Baja California Sur (Squires, 1959). According to Glynn and Ault (2000), maximum shelf depths where coral
colonies or reefs have been observed, from the Golfo de California to Panama, are ~10 m or less. This important observation
provides a third relative sea level indicator; where fossil hermatypic corals are found in growth position, sea level was likely
no higher than ~10 m above that elevation.



## 3 Elevation measurements and geochronology

### 3.1 Elevation measurements

Virtually all of the studies cited herein provide measurements of the elevations of the RSL indicators. In most studies that were conducted before approximately 2010, measurements were typically made using contours on topographic maps, hand level and/or metered tape, transit and stadia rod, or barometric altimeter. For these studies, unless uncertainties are reported in the original manuscript (or where the shoreline angle elevation range is given), elevation uncertainties are assumed to be 20% of the original elevation. This procedure assumes that higher elevation shoreline angles will have greater uncertainties and attribution to an appropriate sea level datum. After approximately 2010, most studies provide elevation measurements done by either handheld or differential Global Positioning System (GPS) instruments (Table 2). Where elevation measurements were made with a handheld GPS instrument, uncertainties can be substantial, and here it is assumed that measurement errors are within $\pm 3$ m of the reported value. For measurements made with a differential GPS instrument, uncertainties are those given in the original study; if not reported, measurement errors are assumed to be within $\pm 0.5$ m.

### 3.2 Geochronology

All of the relative sea level (RSL) indicators that represent MIS 5e on the Pacific Coast of North America considered here have geochronological constraints based on either direct numerical dating using uranium-series (U-series) methods on corals or the correlated-age method of amino acid geochronology, with ties to nearby U-series-dated (coral) localities. As a result, each RSL data point in the database is associated with one or more fossil samples dated with either U-series or amino acid geochronology. Luminescence methods have not been widely applied in this region, although the study by Grove et al. (1995) in the Tomales Bay area provides an important exception. U-series dating of mollusks was once considered a promising method for dating marine terrace fossils in California, but the study by Kaufman et al. (1971) has shown convincingly that mollusks do not take up U during growth, and frequently behave as open systems with respect to U and its daughter products. Thus, early studies that have attempted to date marine terraces by this method are not considered reliable. More recently, cosmogenic isotopes have been attempted in developing chronologies for marine terraces in California (Perg et al., 2001). This method, while promising in theory, requires careful discrimination of which sediments are sampled for analysis. In a study by Perg et al. (2001), ages derived for the terraces near Santa Cruz, California, do not agree with U-series ages on marine terrace corals from the same area (Muhs et al., 2006). The latter investigators speculated that the sediments analyzed by Perg et al. (2001) were likely taken from the terrestrial deposits overlying the marine terrace deposits, which explains the younger than expected cosmogenic ages. Finally, the unique altitudinal-spacing method of Bull (1985) has been applied to marine terraces on the Pacific Coast of North America. Terraces correlated to MIS 5e using this method are not considered in the present review, because Bull's (1985) method assumes that the sea level history derived from the Huon Peninsula of New Guinea is a faithful representation of sea level history on all coastlines around the world (this issue is reviewed in more detail below).



### 3.3.1 Uranium-series dating

Uranium-series dating is based on the fortunate characteristic of corals (Fig. 8) to take up small amounts of U ($^{238}$U, $^{235}$U, $^{234}$U) from seawater into their aragonite skeletons during growth. The U assimilated by corals is in isotopic equilibrium with seawater. In contrast, Th and Pa are very insoluble elements, and therefore ocean water contains essentially no dissolved Th or Pa. Thus, $^{230}$Th and $^{231}$Pa atoms, absent in living corals, accumulate in a fossil, due to decay of $^{234}$U and $^{235}$U, respectively. These two "daughter-deficient" methods utilize daughter/parent activity ratios ($^{230}$Th/$^{234}$U and $^{231}$Pa/$^{235}$U) that begin with 0 in living corals and continue to increase in a fossil until equilibrium values of 1.0 are reached. In addition, $^{234}$U is present in seawater with an ~14-16% greater activity than $^{238}$U (i.e., the $^{234}$U/$^{238}$U activity value in seawater is ~1.15). In a fossil coral, the $^{234}$U/$^{238}$U activity value decreases down to an equilibrium value of 1.0 over time, resulting in a third clock, a "daughter-excess" method.

Both solitary and colonial corals take up U from seawater during growth, usually in amounts ranging from 2-3 ppm, although some genera of corals (notably species of *Acropora*) take up U in amounts ranging from 3-4 ppm. Along the northern part of the Pacific Coast of North America, from Oregon to Baja California, the most common species used for U-series dating is the solitary coral *Balanophyllia elegans* (Fig. 8). Based on studies of living and dead-collected modern specimens, *B. elegans* takes up some additional U after death, but apparently does so from seawater while still submerged and in isotopic equilibrium with U in the ocean (Muhs et al., 2002a, 2006). Farther south, where colonial, hermatypic corals are found, species of the genera *Pocillopora* (Fig. 8) and *Porites* are the taxa most commonly used for U-series dating. In practice, the two clocks used most commonly in U-series dating are $^{230}$Th/$^{234}$U and $^{234}$U/$^{238}$U. Because of the laboratory challenges in using a Pa spike, few laboratories measure $^{231}$Pa/$^{235}$U. It is a common practice to assess $^{230}$Th/$^{234}$U ages by plotting measured $^{230}$Th/$^{238}$U values against measured $^{234}$U/$^{238}$U values, along with the expected isotopic evolution pathways, assuming initial $^{234}$U/$^{238}$U values in seawater. In Figure 9a, such an array is shown for corals from Bermuda and the pathways expected from initial seawater values (for $^{234}$U/$^{238}$U 1.140-1.155; shown here from 1.140-1.160; for $^{230}$Th/$^{234}$U, the initial value is 0.0). Corals that follow these expected isotopic evolution pathways yield ages that likely have minimal bias and can be considered to have had mostly closed-system histories with respect to U-series nuclides. In Figure 9b, what is shown is a much more common situation with corals from the Pacific Coast of North America, with examples from 1st and 2nd terraces on San Nicolas Island, California. While some corals that indicate a closed-system history, similar to Bermuda, others plot above the closed-system evolution pathways. This indicates an open-system history with respect to U-series isotopes in these corals, likely due to recoil-derived additions of $^{230}$Th and $^{234}$U from dissolved U in water passing through the host sediment. An alternative method of assessing degree of closed-system history of fossil corals is to plot the apparent $^{230}$Th/$^{238}$U age as a function of its back-calculated initial $^{234}$U/$^{238}$U value, using the measured $^{234}$U/$^{238}$U value and the apparent age. Examples of this approach are given in Figure 10, where samples, if they have experienced a closed-system history, should fall within the blue-shaded bands that define the range of variability of modern seawater. As is evident from the plots shown in Figure 10, both solitary corals from the Pacific Coast of North America, and colonial corals from Barbados are prone to open-system histories, but some corals show good evidence





of a likely closed-system history. In the examples shown here, it would appear that those corals with closed-system histories
on the Pacific Coast have an age range of ~124 ka to ~114 ka.

In examining U-series data from corals of reef terraces on Barbados, Gallup et al. (1994) noted that even with open-system

histories on isotope evolution plots, a roughly linear trend was observed, with corals that plotted farther above the closed-
system pathway showing a bias to older apparent ages. On the Pacific Coast of North America, the same kind of trend is seen
as that on Barbados (see Fig. 9a), indicating that this may be a general condition in the near-surface environment where fossil
corals are found, despite substantial differences in climate, soil and groundwater hydrology, and composition of surrounding
terrains. Nevertheless, noting this typically linear trend on Barbados, Gallup et al. (1994) suggested that extrapolation of linear
trends back to a closed-system composition could yield an approximate age for a given terrace. This is also part of the basis
of the open-system method of U-series age correction devised by Thompson et al. (2003).

Because of the analytical challenges in determining $^{231}Pa/^{235}U$ ages, it has become a common practice within the U-series

geochronology community to assess the reliability of $^{230}Th/^{234}U$ ages with use of the back-calculated $^{234}U/^{238}U$ values and a
comparison to modern seawater. Although in principle this is an appropriate cross-check, it is not completely reliable. Studies
by Gallup et al. (2002) and Cutler et al. (2003) on corals from Barbados and New Guinea showed that some corals that
demonstrated concordant $^{231}Pa/^{235}U$ and $^{230}Th/^{234}U$ ages did not show back-calculated $^{234}U/^{238}U$ values within the range of
modern seawater. Conversely, some corals that did show back-calculated $^{234}U/^{238}U$ values within the range of modern seawater
did not have concordant $^{231}Pa/^{235}U$ and $^{230}Th/^{234}U$ ages.

Marine terrace corals dated by U-series methods are found within the WALIS database and/or within the compilation of

Chutcharavan and Dutton (2021). Generalized information about each U-series-dated locality can be found in Table S1.
**3.2.2 Amino acid geochronology**

In the absence of corals in a marine terrace deposit or emergent reef, mollusks, both bivalves and gastropods, can be used

for amino acid geochronology. For marine terraces on the Pacific Coast of North America, amino acid geochronology was
pioneered by John F. Wehmiller and his colleagues (Wehmiller et al., 1977a; Lajoie et al., 1980; Kennedy et al., 1982;
Wehmiller, 1982, 1992, 2013a, 2013b). The method is based on the observation that living organisms contain only amino
acids with the "L" (*levo*, or left-handed) configuration. Upon death of an organism, amino acids of the L configuration convert
to amino acids of the "D" (*dextro*, or right-handed) configuration, a reaction called racemization. Racemization is a reversible
process that results in increased D/L ratios in a fossil until an equilibrium ratio of 1.0 is reached. A related process, called
epimerization, is conversion of the amino acid L-isoleucine (found in living organisms) to D-alloisoleucine (not found in living
organisms). Epimerization, like racemization, begins with D-alloisoleucine/L-isoleucine values of 0.0 in a fossil, but this ratio
increases over time until an equilibrium value of 1.25-1.30 is reached (Miller and Mangerud, 1985). Some of the fossils that
have been most commonly used on the Pacific Coast of North America are the bivalves *Saxidomus* and *Chione* (in protected,
sandy or muddy, bay environments) and *Tegula* (in high-energy, rocky-shore environments), shown in Figure 8.

Amino acid values in fossil mollusks can be used for lateral correlation of marine terrace deposits, exploiting the fact that

both racemization and epimerization rates increase with higher diagenetic temperature histories. This means that D/L values



in shells reach equilibrium values more quickly in warmer climates than they do in cooler climates. Thus, on north-south-
trending coastlines in the Northern Hemisphere, such as the Pacific Coast of North America, shells in terrace deposits at more
southerly localities are expected to have higher D/L values than shells of the same genus but of similar age in cooler, northerly
localities. When D/L values are arrayed on a latitudinal plot or a plot of mean annual air temperatures, there should be a south-
to-north decrease in D/L values in shells of the same age. In practice, some localities along such an array have independent
age control from U-series dating of corals. If so, then a shore-parallel correlation of locality to locality, from south to north,
can be accomplished, yielding an "aminozone" corresponding to the age of the independently dated localities. Shells from
younger terraces would define an aminozone below such a zone and older terraces would define an aminozone above it.
The first major attempts at aminostratigraphic correlation along the Pacific Coast using the approach just described were
those by Wehmiller et al. (1977a), Kennedy et al. (1982), and Wehmiller (1982). The north-to-south correlation of terraces
from Kennedy et al. (1982) is shown in Figure 11a, along with three U-series-dated localities that serve as calibration points.
Kennedy et al. (1982) also noted that most localities correlated to either MIS 5a or MIS 3 hosted terrace faunas with cool-
water aspects, whereas those correlated to MIS 5e had warm-water faunas, or at least faunas that were "neutral," lacking cool
or water-water taxa. In the time since the Kennedy et al. (1982) study was conducted, more U-series ages on coral have been
reported (~120 ka, ~80 ka, and ~47 ka), many of which support the original aminostratigraphic correlations (Fig. 11b).
Nevertheless, some localities are now known to host mixes of warm and cool faunas and at least two of these have mixes of
~120 ka (MIS 5e) and ~100 ka (MIS 5c) corals (Fig. 11b). This issue is discussed in more detail below.
Even with some concerns, amino acid geochronology has been shown to be a very powerful coast-parallel correlation tool.
Even within the limited geographic range of central California to northern Baja California, there is enough of an air temperature
gradient that aminostratigraphic correlation can be accomplished. At a given locality where two terraces are found (one at a
low elevation, one at a higher elevation), MIS 5a and MIS 5e terrace deposits can usually be distinguished from one another
(Fig. 12). Furthermore, lateral correlation of MIS 5e and MIS 5a deposits from central California to northern Baja California
can be made, anchored by localities with U-series ages on corals.
Similar to U-series-dated marine terrace corals, those terrace localities correlated to MIS 5e with amino acid racemization
or epimerization methods are found within the WALIS database, along with linkage to the U-series-dated localities that served
as calibration. Generalized information about each locality correlated to MIS 5e with amino acid geochronology can be found
in Table S2.
**3.2.3 Zoogeographic aspects of terrace faunas**
In a pioneering study of marine terraces on the Pacific Coast of North America, Kennedy et al. (1982) used the
aminostratigraphic approach described above to extend earlier work by Wehmiller et al. (1977a). Both studies established
that the lowest marine terrace along the Pacific Coast of North America is not the same age at all localities, due to varying
rates of uplift from one reach of coast to another. In addition, Kennedy et al. (1982) noted that localities dated (by U-series
on coral) to or correlated with MIS 5e host either zoogeographically "neutral" molluscan fossil faunas or faunas that contain





extralimital southern species. In contrast, localities that were either dated or correlated to the ~80 ka MIS 5a host molluscan
fossil faunas with several extralimital northern species (Fig. 11). Extralimital species (or northward or southward-ranging
species) are those that, while extant, do not live at a particular locality at present, but are found either entirely or mostly to the
north (cool waters in this region) or to the south (warmer waters in this region). An example of a locality, dated to ~130 ka by
thermoluminescence (Grove et al., 1995), is the marine deposit in Tomales Bay, north of San Francisco, California. This
deposit contains many "neutral" species, i.e., those that still live in the area at present, but also host a large number of
extralimital southern and southward-ranging species (Fig. 13). In contrast, the Davenport terrace in the Santa Cruz, California
area, dated to ~80 ka by U-series methods on corals (Muhs et al., 2006), hosts only one southward-ranging species, but several
extralimital northern and northward-ranging species. Warmer waters off California during MIS 5e and cooler waters during
MIS 5a are consistent with the zoogeographic aspects of planktonic foraminiferal faunas found in deep-sea cores (Kennett and
Venz, 1995) and with sea surface temperatures (SST) derived from alkenones (Herbert et al., 2001; Yamamoto et al., 2007).
**4 Relative Sea Level indicators**
Relative sea level indicators from the Pacific Coast of North America for MIS 5e and all pertinent data related to them are
given in Table S1 and Figure 14 (U-series-dated coral-bearing localities) and Table S2 and Figure 15 (localities correlated to
MIS 5e using aminostratigraphy). In the sections that follow, the regions these localities are from are discussed with respect
to the nature of the sea level record, as this differs from region to region. Within the course of these discussions, previous
studies are examined and the basis for the age assignments is discussed critically. For simplicity, the review of the regions is
taken from north to south. In the text that follows, there is an indication near each site discussed of what the unique RSL
identification is, corresponding to the WALIS database.
**4.1 Southwestern Canada**
Records of marine deposits dating to MIS 5e are difficult to find on the coast of British Columbia. Erosion by repeated
advances of the Cordilleran ice sheet has likely removed much of the potential record. Furthermore, the sedimentary record
that does exist is highly complex, due to rapid sedimentation rates, active tectonics, and glacial isostatic adjustment (GIA)
effects. Mollusk-bearing glaciomarine sediments were deposited in lowland areas adjacent to coastal British Columbia or in
Puget Sound when isostatic depression of these areas allowed inflow of ocean waters. Thus, at least some of the marine record
that is now emergent is not strictly "interglacial," but likely occurred at the transition between a glacial period and the following
interglacial period.
In southwestern Canada, most investigators have hypothesized, from the stratigraphic sections that have been studied, that
the main record of MIS 5e is the Muir Point Formation (Hicock and Armstrong, 1983; Alley and Hickock, 1986; Hicock,
1990; Clague et al., 1992). On Vancouver Island in British Columbia (Fig. 16), the Muir Point Formation consists of gravel,
sand, and silt, with abundant peat and wood layers, suggesting a mostly terrestrial origin, but Hicock and Armstrong (1983)
hypothesize an alluvial fan-to-floodplain-to-coastal plain-to-delta sequence, based on the sediment facies. Indeed, Alley and
Hicock (1986) and Hicock (1990) report minor amounts of marine dinoflagellate cysts in a part of the Muir Point Formation,



implying tidal or estuarine conditions, and these investigators infer a paleo-sea level of at least +10 m, relative to present. A
last interglacial origin for the Muir Point Formation was hypothesized early in the study of this formation by its stratigraphic
position: it has organic materials that date to >40 ka and has normal polarity, but is underlain by older till and overlain by mid-
Wisconsin (MIS 3) Cowichan Head Formation sediments, in turn overlain by Vashon Till dating to the Fraser Glaciation (=late
Wisconsin, or MIS 2) (Alley and Hicock, 1986; Hicock, 1990). Vegetation evidence also suggests a climate at least as warm
as today's, based primarily on the abundance of thermophyllous *Pseudotsuga* (Douglas fir) pollen, implying interglacial
conditions (Hicock and Armstrong, 1983; Alley and Hicock, 1986; Hicock, 1990). An MIS 5e age is permitted by optically
stimulated luminescence (OSL) ages of $119 \pm 9$ ka and $112 \pm 11$ ka from the Muir Point Formation at and near its type section
(Lian et al., 1995). Because more study is needed for assessment of the age of the Muir Point Formation, no specific entry in
the WALIS database was attempted here.
**4.2 Washington, USA**
Only two fossil-bearing localities are candidates for MIS 5e deposits in the State of Washington, one in Puget Sound and
the other on the outer coast, at Willapa Bay (Fig. 16). Both have had a confusing and/or controversial history of study.
**4.2.1 Whidbey Island, Puget Sound**
As is the case with British Columbia, the southern Puget Sound area, within the boundaries of Washington State, has been
subjected to rapid sedimentation rates, active tectonics, and GIA effects, as well as removal of much of the geologic record,
due to advances and retreats of the Cordilleran ice sheet. Also similar to British Columbia, the main geologic unit that most
investigators agree records the last interglacial period (MIS 5e) is not primarily a marine deposit at all, but a terrestrial deposit
called the Whidbey Formation. Hansen and Mackin (1949) were among the first to study the formation, noting that it occurred
stratigraphically below deposits dating to the last glacial period (i.e., MIS 4 through MIS 2), and that it hosted pollen indicating
an interglacial vegetation similar to that of the present. Easterbrook et al. (1967) were the first investigators to apply the formal
name Whidbey Formation to the pollen-bearing unit studied by Hansen and Mackin (1949) and designated the type locality
on coastal bluffs of southwestern Whidbey Island (Fig. 16). At the type section, Easterbrook et al. (1967) and Easterbrook
(1968, 1969) noted that the Whidbey Formation is underlain by what is called Double Bluff Drift, consisting of till and
glaciomarine sediments. At this locality, the Whidbey Formation is overlain by glacial deposits of Possession (MIS 4?) and
Vashon (MIS 2) age. Easterbrook et al. (1967) conducted pollen analyses of Whidbey Formation sediments and concluded
that the vegetation implied an interglacial climate similar to the present. They also reported ages that showed the unit was
beyond the range of radiocarbon dating. More detailed pollen work was conducted by Heusser and Heusser (1981), who
reached the same conclusions about past climate conditions. Karrow et al. (1995) reported on nonmarine fossils in the Whidbey
Formation, including mollusks, ostracodes, insects, fish, vertebrates, and plant macrofossils. Their interpretations are similar
to those of Hansen and Mackin (1949), Easterbrook et al. (1967), and Heusser and Heusser (1981), that the deposit likely
represents an interglacial period with a degree of warmth similar to that of the present. It is important to note that in all of the
studies just cited, the Whidbey Formation is described as a *terrestrial* deposit, likely formed as floodplain sediments. None
of the studies cited here mention the presence of marine fossils within the deposit. Later studies have all confirmed a likely





MIS 5 age for the Whidbey Formation, based on thermoluminescence (TL) dating (151 ± 43 ka to 102 ± 38 ka; Berger and
Easterbrook, 1993), optically stimulated luminescence (OSL) dating (107 ± 8 ka; Lian et al., 1995), and $^{40}Ar/^{39}Ar$ dating of
plagioclase from pumice within the formation (128 ± 9 ka; Dethier et al., 2008).

With the advent of amino acid geochronology, several studies presented data on some of the marine-shell-bearing deposits

of Whidbey Island. Most of these studies focus on a shell-bearing deposit along Admiralty Bay, on the west coast of Whidbey
Island. This deposit is visible in an ~17 m thick coastal exposure (~2 m above sea level) of diamicton and/or glaciomarine
sediment composed of gravel, sand, and silt, with a layer of marine fossils, dominated by *Saxidomus gigantea*, in its uppermost
part (Polenz et al., 2009). The upper contact of this complex deposit is obscured by a recent landslide, but a short distance
inland and at higher (~40 m to ~60 m) elevations, glaciomarine deposits of the Everson Interstade, outwash of the Fraser
Glaciation, and till of the Vashon Stade (all of MIS 2 age) are mapped (Polenz et al., 2009) and likely overlie the shell-bearing
deposits exposed at lower elevation. Kvenvolden et al. (1980) reported that this fossiliferous deposit lies stratigraphically
between the Whidbey Formation and "middle Wisconsin sediments" and used amino acid ratios in *Saxidomus gigantea* to
estimate an age of ~80 ka. Their study, however, presents no stratigraphic evidence of the Whidbey Formation being exposed
at Admiralty Bay. Blunt (1982) analyzed shells from the same locality as Kvenvolden et al. (1980) and another locality ~150
m to the north. He used kinetic modeling to derive an age range of 77 ka to 99 ka for the locality studied by Kvenvolden et
al. (1980) and 75-110 ka for the newer locality. In a later study derived primarily from data in Blunt (1982), Blunt et al. (1987,
p. 331-332) described the Admiralty Bay deposit as belonging to the Possession Glaciation (which postdates the Whidbey
Formation), but later in the same paper (p. 340 and p. 346) said that the deposits are correlated with the Whidbey Formation.
These investigators also used *Saxidomus gigantea* and kinetic modeling to estimate ages of ~96 ka and ~107 ka for the deposit,
apparently pooling the two localities. Using these data, Easterbrook correlated the deposit at Admiralty Bay with the Whidbey
Formation. Kennedy et al. (1982) used the *Saxidomus gigantea* single-shell analysis in Kvenvolden et al. (1980) to estimate
an aminozone-derived, correlated age of 80 ka, and apparently used the pooled *Saxidomus gigantea* data in Blunt (1982) to
estimate an aminozone-derived, correlated age of 120 ka. In addition, Kennedy et al. (1982) reported that the 80 ka locality
hosts a cool-water fauna and the ~120 ka locality hosts a warm-water fauna. This is puzzling, because no faunal data are given
in Kvenvolden et al. (1980) or Kennedy et al. (1982), although Blunt (1982) reports a single extralimital northern or at least
northward-ranging species, *Mya truncata*, in the fossil deposit at Admiralty Bay. Furthermore, the two localities are only ~150
m apart and occur at roughly the same elevation according to Blunt (1982). Finally, Polenz et al. (2009) presented
sedimentological data indicating that the deposits at Admiralty Bay have a glaciomarine origin. These investigators correlated
the deposit either to the pre-last interglacial Double Bluff Glaciation (their favored option) or the post-last interglacial
Possession Glaciation. In my own examinations of the deposits at Admiralty Bay, I have seen no evidence for more than one
stratigraphic unit. I also agree with Polenz et al. (2009) that the shell-bearing deposit exposed there is likely glaciomarine
drift, dating to the transition between the penultimate glacial period (MIS 6), represented by the Double Bluff unit, and MIS
5e. It is likely an older equivalent of the shell-bearing glaciomarine drift of the late, last glacial Fraser glaciation, a unit called
the Everson glaciomarine deposits. For this reason, this locality has not been entered into the WALIS database.




### 4.2.2 Willapa Bay

The only other emergent, fossil-bearing locality that is a candidate for an MIS 5e deposit in Washington State is along the inner shores of Willapa Bay (**RSL ID 3684**) (Fig. 16). Near Bay Center, sea cliffs expose marine sediments that are richly fossiliferous (Fig. 17). Addicott (1966) reported the fossil fauna from this locality, which consists mostly of bivalves, and no taxa are extralimital, or even northward or southward ranging. Kvenvolden et al. (1979) provided the first published amino acid data from this area. They recognized four stratigraphic units (I, intertidal; II, subtidal; III, subaerial; and IV, subtidal, from oldest to youngest). Their unit IV is the thickest and apparently the most extensive deposit, interpreted to have an estuarine origin; the top of this unit defines a marine terrace surface, at an elevation of ~13 m. Almost all of the *Saxidomus gigantea* specimens they analyzed are from this youngest deposit. Using assumed calibration ages of ~68 ka and ~100 ka for the lowest terrace at Santa Cruz, California (Bradley and Addicott, 1968), which also hosts fossil *Saxidomus gigantea*, Kvenvolden et al. (1979) used linear kinetic modeling (taking temperature differences into account) to generate age estimates of 190 ± 40 ka for units I and II, and 120 ± 40 ka for unit IV, which they correlated to MIS 7 (I and II) and MIS 5 (IV). Their terrestrial unit III was interpreted to have formed when sea level lowered during MIS 6. It is now known that U-series ages on mollusks, including the ~68 ka and ~100 ka ages for Santa Cruz reported by Bradley and Addicott (1968), are not reliable (Kaufman et al., 1971). Nevertheless, reliable U-series ages on corals from the same terrace yielded ages in between these, averaging about 80 ka (Muhs et al., 2006). Thus, the newer ages, if used as calibration, would not change the original kinetic model ages for the Willapa Bay deposits. In any case, Wehmiller (1981) challenged Kvenvolden et al.'s (1979) age estimates, arguing that nonlinear kinetic modeling is more appropriate for numerical ages using amino acid data. Using nonlinear kinetic modeling, Wehmiller (1981) recalculated the ages of units I/II and IV at Willapa Bay to be 300 ± 50 ka and 70 ± 15 ka, respectively, suggesting correlation with MIS 9 and MIS 5a. Kvenvolden et al. (1981) countered that nonlinear kinetics could be applied to amino acid values within the ranges of what their samples yielded, and also noted that Wehmiller's (1981) age estimates would require a much more complex geologic history than their age estimates. Kennedy et al. (1982) reported new amino acid values in *Saxidomus* from unit IV and using a lateral correlation (aminozone) approach, considered that the unit IV deposits at Willapa Bay were of MIS 5a age, in agreement with Wehmiller (1981). They also reported that the fauna at the Bay Center locality of unit IV hosted cool-water forms, although Addicott (1966) reported no extralimital species or northward-ranging species. The cool-water aspect of the fauna at Bay Center is apparently based on the identification of *Mya japonica* in these deposits, reported by Kennedy (1978). Although *M. japonica* was once considered to range only in the Arctic seas, from Japan to Nome, Alaska (Abbott, 1974), Coan et al. (2000) consider that *M. japonica* does not have differences with *M. arenaria* that are sufficient to merit specific status. If so, then there are no extralimital northern species in the fauna of unit IV at Willapa Bay and the assemblage as a whole can be considered zoogeographically "neutral." Given all the uncertainties in what has been reported thus far for Willapa Bay, it seems likely that unit IV of Kvenvolden et al. (1979) could date to MIS 5e, but more geochronological information is needed to be certain of this.

### 4.3 Oregon, USA



Moving south from Washington, coastal Oregon is where the dominant geomorphic expression of MIS 5e shorelines as
erosional marine terraces begins. The coast of Oregon is within the Cascadia subduction zone (Fig. 2) and most of it can be
characterized as a high-wave-energy environment. Thus, erosional marine terraces are common landforms along a substantial
amount of the coast, particularly in the central and southern parts of Oregon (Fig. 16). A pioneering study by Griggs (1945)
involved the mapping and naming of the lowest three marine terraces in southern Oregon and the terrace names are still in use
today. More recently, detailed mapping of marine terraces along the Oregon coast has been conducted primarily by H.M.
Kelsey and his colleagues and students (Kelsey, 1990; McInelly and Kelsey, 1990; Bockheim et al., 1992; Kelsey and
Bockheim, 1994; Kelsey et al., 1996). Candidate landforms for some or all substages of MIS 5 are the lowest three terraces
found along much of the central and southern Oregon coast. Kennedy et al. (1982) inferred that the lowest of these, the Whisky
Run terrace near Coquille Point (Figs. 18, 19) likely correlated to MIS 5a because of a U-series age of ~72 ka on a coral from
its deposits, as well as relatively low D/L leucine values in *Saxidomus gigantea*, and a cool-water aspect to the terrace fauna
(Zullo, 1969; Kennedy, 1978). Later, both alpha-spectrometry and TIMS U-series ages of corals from the Whisky Run terrace
confirmed an age of ~80 ka, and a more extensive cool-water fauna was reported (Muhs et al., 1990, 2006). Higher terraces
are present in this area and north to Cape Arago, named the Pioneer, Seven Devils, and Metcalf terraces (lowest to highest),
mapped by McInelly and Kelsey (1990). Based on the ~80 ka age of the Whisky Run terrace, McInelly and Kelsey (1990)
inferred that MIS 5e is represented by the Seven Devils terrace, with the Pioneer terrace correlated to MIS 5c.
Farther south, the lowest terrace at Cape Blanco also hosts a cool-water fauna (Addicott, 1964a), but based on amino acid
values, Kennedy et al. (1982) interpreted this terrace to be of post-MIS 5 age, possibly as young as MIS 3. In a later study,
Kelsey (1990) remapped the terraces in this area and named this the Cape Blanco terrace (Fig. 20). He correlated this with the
Whisky Run terrace at Coquille Point, supported by new amino acid and oxygen isotope values in *Saxidomus gigantea* from
Cape Blanco (Muhs et al., 1990). Kelsey also mapped and named higher landforms above the Cape Blanco terrace, the Pioneer,
Silver Butte, and Indian Creek, from lowest to highest (Figs. 18, 19). He considered the Pioneer terrace to represent the MIS
5c high-sea stand and the Silver Butte terrace to represent the MIS 5e high stand. Amino acid data given by Muhs et al. (1990)
support the correlation of the Pioneer terrace to MIS 5c, but no fossils have yet been found on the Silver Butte terrace.
North of Coquille Point, near Newport, Oregon, Kennedy et al. (1982) reported amino acid values in *Saxidomus gigantea*
from a low marine terrace near Newport jetty and a higher terrace at Yaquina Bay. On the basis of these amino acid ratios and
a cool-water fauna (lower terrace) and a warm-water fauna (higher terrace), Kennedy et al. (1982) correlated these terraces
with MIS 5a and MIS 5e, respectively. Later mapping by Kelsey et al. (1996) identified these as the Newport (lower) and
Yachats (higher) terraces, respectively, with an intermediate-elevation landform they named the Wakonda terrace. They
correlated the Newport, Wakonda, and Yachats terraces with MIS 5a, 5c, and 5e, respectively.
Summarizing all these studies, U-series, amino acid, oxygen isotope, and faunal data all support a correlation of the lowest
marine terrace at Newport, Coquille Point, and Cape Blanco to MIS 5a (Fig. 19). At Cape Blanco, similar amino acid ratios
and oxygen isotope data correlate the Pioneer terrace with MIS 5c. At Newport-Yaquina Bay, the Yachats terrace (**RSL ID**
**3685**) is correlated to MIS 5e by amino acids and faunal data. Lack of fossils precludes correlation of intermediate and higher





terraces at all these localities. To address this problem, Kelsey and Bockheim (1994) used degree of soil development to
correlate undated terraces in all three areas, plus a fourth area in southernmost Oregon, near Cape Ferrelo (Fig. 16), where all
terraces lack fossils. With the generation of a soil development index that utilizes time-dependent soil properties (e.g., Bt
horizon thickness, color, texture, clay content), they identified, from north to south, the Yachats, Seven Devils, Silver Butte,
and Gowman terraces as the likely candidates for records of the MIS 5e high-sea stand.
**4.4 Northern California, USA**
**4.4.1 Crescent City coastal plain**
Surprisingly few studies of marine terraces have been undertaken in northern California, in part because fossil-bearing
occurrences that would permit dating are rare. Northernmost California is within the Cascadia subduction zone, similar to
coastal Oregon (Fig. 21). About 25 km south of the Oregon border, marine terraces have been studied for decades on the
Crescent City coastal plain. Maxon (1933) named all the marine terrace deposits in this area collectively the Battery Formation,
and he also noted the presence of fossil invertebrates in the deposits. Similarly, Delattre and Rosinski (2012) mapped deposits
of the entire Crescent City coastal plain as the Battery Formation. The first attempt at dating these deposits was by Kennedy
et al. (1982), who presented amino acid data from *Saxidomus gigantea* from low-elevation (~7 m) sea cliff exposures in
southern Crescent City. These investigators also reported a cool-water fauna from this low-elevation terrace and on the basis
of D/L leucine values, correlated the terrace with MIS 5a.
In contrast to Maxon (1933) and Delattre and Rosinski (2012), Polenz and Kelsey (1999) recognized three marine terraces
in this area (Qpm3, Qpm2, and Qpm1, from youngest to oldest), differentiated by subtle elevation changes and differing
degrees of soil development, following the approach used by Kelsey and Bockheim (1994) in southern Oregon. Polenz and
Kelsey (1999) correlated the three terraces they mapped (Qpm3, Qpm2, Qpm1) with MIS 5a, 5c, and 5e, respectively, although
they noted that Qpm1 could correlate with MIS 7. The localities studied by Kennedy et al. (1982) are situated on what Polenz
and Kelsey (1999) mapped as Qpm2, the terrace they correlated with MIS 5c. It should be noted, however, that it is
questionable whether amino acid ratios can distinguish ~80 ka deposits from ~100 ka deposits, and cool-water faunas are
expected from terraces of either age, based on alkenone paleotemperature data from a nearby deep-sea core (ODP 1020) studied
by Herbert et al. (2001). Thus, the best evidence for a possible MIS 5e shoreline in this area is the Qpm1 terrace mapped by
Polenz and Kelsey (1999), found mostly inland of the younger terraces. This terrace has maximum platform elevations of ~29
m to ~15 m.
**4.4.2 Trinidad Head area**
Marine terraces are scarce between Crescent City and along a coastal reach ~60 km to the south. However, in the Trinidad
Head area (Fig. 21), there are multiple marine terraces, well expressed geomorphically. This area, like the Crescent City
coastal plain, is also within the Cascadia subduction zone. Carver (1992) mapped seven terraces in this area, with additional
undifferentiated higher elevation terraces. Based on ages assigned from the oxygen isotope record and an untested assumption
of a constant uplift rate, Carver (1992) gave estimated terrace ages that were also followed by Delattre and Rosinski (2012).
The rationale for these age assignments is reported to be from degree of soil development and thermoluminescence (TL) ages





reported by Berger et al. (1991). However, the method by which ages from degree of soil development are derived is not
described and all of the TL ages reported by Berger et al. (1991) are either not consistent with Carver's (1992) mapping or date
younger deposits that overlie the marine terrace deposits. Although not mapped, McCrory (2000) also presented shore-parallel
terrace profiles for seven marine terraces in this area, with shoreline angle elevations ranging from ~15 m to ~255 m. No
numerical ages are available, but McCrory (2000) used a graphical method of estimating terrace ages, as described by Lajoie
(1986). Finally, Padgett et al. (2019) remapped the terraces in this area and assigned ages based on degree of soil development
and an assumption that the terrace with the most prominent inner edge (their "Surface 3") dates to MIS 5e. They further
assumed that the lower-elevation "Surface 1" and "Surface 2" terraces date to MIS 5a and 5c, respectively. Thus, the mapping
and age assignments of Carver (1992) and Delattre and Rosinski (2012) disagree with those of Padgett et al. (2019), but it is
important to emphasize that none of these studies have any supporting numerical ages. Interestingly, pre-MIS 5e ages are
given by Kennedy et al. (1982) for marine deposits in this area, based on D/L values in *Saxidomus gigantea*, but because no
geomorphic or stratigraphic data are given, it is not known how these aminostratigraphic data can be linked to the other studies.
Much more work needs to be done on dating the terraces in this area, and although it seems likely that an MIS 5e record is
present, it cannot be determined at this time which shoreline is representing it.
**4.4.3 Eureka-Cape Mendocino area**
The Eureka area is situated within the southernmost part of the Cascadia subduction zone, but Cape Mendocino is close to
the Mendocino triple junction, where the North America, Gorda, and Pacific plates intersect (Fig. 21). McLaughlin et al.
(2000) mapped the geology of the Eureka, California area, as well as the Cape Mendocino area to the south of Eureka. Within
the Eureka area itself, these investigators mapped a unit simply called "Qt," which is primarily nonmarine, fluvial terrace
deposits, but which also includes shallow marine deposits, including an informally named deposit called the "Hookton marine"
unit (Ogle, 1953). Wehmiller et al. (1977b) and Kennedy et al. (1982) reported leucine D/L values in *Saxidomus gigantea*
from two localities in the Eureka area (one of which is at an elevation of 15-17 m, **RSL ID 3686**). Both localities are within
the "Qt" unit of McLaughlin et al., 2000) that would permit correlation of the host deposits with MIS 5e. One of these localities
is reported to host a warm-water fauna and the other is reported to host a zoogeographically "neutral" fauna, but stratigraphic
and faunal details are not given. It seems likely that the Eureka area hosts marine deposits that correlate to MIS 5e, but more
detailed work is required to confirm this.
Between Cape Mendocino and Point Delgada, the region is within the influence of the tectonically active Mendocino triple
junction (Fig. 21). Along this rugged part of the coast, numerous terraces have been identified, including some dating to the
Holocene and even historic time, described later. For the Pleistocene, terrace elevation transects were reported by Merritts
and Bull (1989). No numerical ages are given, but terraces correlated to MIS 5e were reported at elevations of ~150 m (a short
distance north of Point Delgada) and at ~250 m (near Punta Gorda). These correlations were made using a graphical correlation
method described by Bull (1985). Bull's (1985) method was developed before much was known about the importance of GIA
effects, which have been shown to affect the California coast, and this issue is discussed later. Thus, whether the correlations



of Merritts and Bull (1989) of the terraces in the Punta Gorda and Point Delgada areas to MIS 5e are valid or not will have to
await independent dating.

### 4.4.4  Laguna Point to Point Arena

Between Laguna Point and Point Arena, marine terraces form the coastal plain area of this part of northern California. This
area is south of the Mendocino triple junction and is within the strike-slip tectonic region of the San Andreas Fault zone (Fig.
21). Although detailed mapping has not been conducted in this area, general terrace maps are available and a portion of
terraced coastline is shown here (Fig. 22). Merritts and Bull (1989) have reported elevations of terrace inner edges on shore-
normal transects in this area as well. These investigators report as many as six marine terraces in the Cabrillo Point area, from
~10 m to ~130 m above sea level. Unfortunately, only one locality, thus far, has yielded fossils in this reach of coastline. On
the lowest terrace (shoreline angle of ~10 m) at Laguna Point, within MacKerricher State Park (Fig. 14), Kennedy et al. (1982)
reported D/L leucine ratios on *Saxidomus gigantea* fragments, as well as a fauna with extralimital northern species. The amino
acid ratios on these shells plotted above their ~80 ka aminozone, creating a dilemma: the amino acid data suggested correlation
with MIS 5e but the fauna was typical of MIS 5a deposits. Dorothy J. Merritts of Franklin and Marshall College returned to
this locality and recovered two solitary corals (*Balanophyllia elegans*) that she submitted to laboratories at the U.S. Geological
Survey. For this review, she kindly allowed use of these previously unpublished data. One coral has a low U content and an
apparent age of ~156 ka, too old to be considered to be of MIS 5e age, but likely biased old because of U loss. However, the
other coral has a U content of 4.82 ±0.10 ppm, a $^{232}$Th content of 0.06 ppm, a $^{230}$Th/$^{232}$Th value of 180, a $^{234}$U/$^{238}$U value of
1.0976 ±0.0016, a $^{230}$Th/$^{238}$U value of 0.7771 ±0.0024, and an age of 130.4 ±0.9 ka. The back-calculated initial $^{234}$U/$^{238}$U
value of 1.1411 ±0.0022 is within the range of modern seawater, giving the age of ~130 ka a high degree of confidence. This
age is consistent with the age implied by the amino acid data reported by Kennedy et al. (1982). The issue of an MIS 5e terrace
such as this, containing a cool-water fauna, is contrary to the general model proposed by Kennedy et al. (1982). These
apparently contradictory observations can be reconciled, however, in areas of low uplift rate, by formation of an MIS 5e
terrace, followed by reoccupation (and fossil reworking) during the MIS 5c high-sea stand, a topic that is explored in more
detail later.
Marine terraces at Point Arena and south of it have been mapped by Muhs et al. (2003, 2006), who report three terraces in
this area. Corals from their "Qt1," the lowest terrace (~20-25 m), were dated by alpha-spectrometric methods to ~80 ka, or
MIS 5a (Muhs et al., 1990, 1994). Later dating of coral from this terrace by TIMS also gave an age of ~80 ka, and
paleontological studies yielded a fauna with extralimital northern forms (Muhs et al., 2006). All these results are in good
agreement with those of Kennedy et al. (1982), who correlated the terrace to MIS 5a on the basis of D/L leucine values in
*Saxidomus gigantea*. The two higher terraces mapped by Muhs et al. (2003) have elevations of ~40-45 m (Qt2) and ~60-65
m (Qt3). A reasonable working hypothesis is that Qt3 represents the last interglacial peak (MIS 5e) high-sea stand and that
the intermediate Qt2 terrace formed during the MIS 5c high-sea stand. Unfortunately, fossils on both of these terraces have
yet to be found, so testing of this hypothesis is not yet possible.


### 4.4.5 Tomales Bay

Tomales Bay (**RSL ID 3794**) has been of interest to geologists because of its unusual configuration, conditioned largely by the fact that the San Andreas Fault zone is situated within the bay, parallel to the bay's long axis (Fig. 23). At a few points on the eastern side of Tomales Bay, there are exposures of a marine deposit called the Millerton Formation, long considered to be of Pleistocene age. On the most recent geologic map of the area, the formation is simply included within what is mapped as "marine terrace deposits" (Graymer et al., 2006). At Toms Point, it is a fossil-rich bed (with abundant *Saxidomus* and *Chione* shells), ~0.5 m thick, overlying a bench cut on Franciscan rocks, and overlain by nonmarine terrestrial deposits. The shell-rich bed is ~8-9 m above sea level at Toms Point. At Millerton Point, the beds are gravelly, ~1.0 m thick, and are rich in *Ostrea* and *Leukomca* (formerly *Protothaca*) shells, all exposed just above modern beach level. Johnson (1962), who conducted the most thorough study of the fossils from the Millerton Formation, noted that several extralimital southern and southward-ranging species are present (Fig. 13), with no northern species, implying water temperatures much warmer than those at Tomales Bay today. Kennedy et al. (1982) noted the warm-water aspect of the fauna in the Millerton Formation, and presented D/L leucine data in *Saxidomus* that fall slightly above their ~120 ka, MIS 5e aminozone. These investigators did not specifically accept or reject a possible MIS 5e age for the formation. Grove et al. (1995) studying the tectonics of the area, reported a TL age of 134 ±12 ka for the Millerton Formation (analyzed by G.W. Berger). Unfortunately, no analytical data are given for further consideration of this TL age. However, Muhs and Groves (2018) presented D-alloisoleucine/L-isoleucine data for *Chione* from the Millerton Formation, collected at Toms Point. Their data fall into a last interglacial (MIS 5e) aminozone when compared with a south-to-north transect of similar data from *Chione* in Baja California and California. More dating of this important formation would be highly desirable, but the available information indicates that the Millerton Formation along the northeast shores of Tomales Bay represents deposits that can be correlated to MIS 5e.

## 4.5 Central California, USA

### 4.5.1 Point Año Nuevo-Santa Cruz area

South of San Francisco, marine terraces dominate many parts of the coast of central California. As noted earlier, it was in this area, east of Santa Cruz (Fig. 21), that Alexander (1953) formulated the modern concept of how marine terraces form on tectonically active coastlines, specifically as landforms cut during interglacial sea-level high stands superimposed on crustal blocks experiencing steady uplift. Just west of where Alexander (1953) worked, Bradley and Griggs (1976) mapped six prominent terraces in the Santa Cruz-Point Año Nuevo area (Fig. 24). The lowest of these six marine terraces, between Santa Cruz and extending to at least just north of Point Año Nuevo, is called the Santa Cruz terrace. Detailed seismic profiling and examination of outcrops carried out by Bradley and Griggs (1976) show that the Santa Cruz terrace actually consists of three distinct platforms cut on bedrock. However, the three wave-cut platforms are covered with marine and, importantly, nonmarine deposits that have smoothed over the subaerial surface topographically into a single, broad landform. Bradley and Griggs (1976) referred to the three buried platforms as the Greyhound level (~45 m), Highway 1 level (~35 m), and Davenport level (~20 m). These elevations are rough averages, as the shoreline angle elevations vary as a function of where they are situated





with respect to active geologic structures. Too few exposures allow for these terraces to be mapped separately, but isolated
outcrops where the Davenport platform can be identified are present along the coast between Santa Cruz and north of Point
Año Nuevo (Fig. 24). Above the Santa Cruz terrace complex, higher terraces are found at ~55-60 m (Cement terrace), ~90-
100 m (Western terrace), ~120-140 m (Wilder terrace), ~180-195 m (Black Rock terrace), and ~240-260 m (Quarry terrace).
For years, there has probably been no greater speculation about the age of a coastal landform in California than that of the
Davenport platform between Point Año Nuevo and Santa Cruz. Anderson and Menking (1994) reviewed many of the previous
age estimates made for this terrace. Bradley and Addicott (1968) reported U-series ages of mollusks from this terrace that
ranged from ~100 ka to ~60 ka, but it is now well known that U-series analyses of mollusks are not reliable (Kaufman et al.,
1971). In a later study, Bradley and Griggs (1976) recognized this problem and suggested instead that the Highway 1 platform,
present just above the Davenport platform, was cut during the ~120 ka (MIS 5e) high-sea stand. They interpreted the Davenport
platform, although at a lower elevation, to have been cut during a hypothesized lower, ~140 ka sea stand, such as that seen at
reef VIIa on New Guinea (Bloom et al., 1974). Based on amino acid ratios in fossil mollusks and the faunal zoogeographic
aspect, Kennedy et al. (1982) concluded that the fossils on the Davenport platform dated to the ~80 ka high stand of sea during
MIS 5a. Assuming an age of ~120,000 yr for the Highway 1 terrace, Lajoie et al. (1991) estimated an age of ~100 ka (MIS
5c) for the Davenport terrace. Perg et al. (2001) used cosmogenic isotopes to estimate an age of 65 ka for the Highway 1
terrace where it is found just northwest of Santa Cruz, which would correlate the terrace with MIS 3. These workers offered
cosmogenic isotope ages for the higher terraces as well, including ~92 ka (Western terrace), correlated to MIS 5a; ~137 ka
(Wilder terrace), correlated to MIS 5c; ~139 ka (Black Rock terrace), correlated to MIS 5e; and ~226 ka (Quarry terrace),
correlated to MIS 7. The age estimates of Perg et al. (2001) combined with the elevations given above would characterize this
reach of coastline as having one of the highest rates of uplift along the California coast, exceeded only by the coast near the
Mendocino triple junction and south of the "big bend" of the San Andreas Fault, discussed later.
Muhs et al. (2006) reported U-series ages corals on from Green Oaks Creek, Point Año Nuevo, and Point Santa Cruz, all
from deposits of the Davenport level of the Santa Cruz terrace (Fig. 24). Twelve corals gave ages ranging from ~84 ka to ~76
ka and all 11 corals collected near Green Oaks Creek have back-calculated initial $^{234}U/^{238}U$ values ranging from 1.154 to
1.1460. These values fall well within the range of modern seawater, giving a high degree of confidence that the corals have
experienced closed-system histories with respect to U-series isotopes. In addition, deposits of the Davenport terrace have
faunas containing a large number of extralimital northern or northward-ranging species of mollusks, consistent with dated,
~80 ka terraces elsewhere in Oregon and California (Addicott, 1966; Kennedy, 1978; Muhs et al., 2006). The U-series ages
of ~80 ka agree with the amino acid age estimate reported earlier by Kennedy et al. (1982).
Although Perg et al. (2001) did not analyze sediments from the Davenport platform directly, their ages for the higher
terraces imply that the Davenport terrace should correlate with one of the interstadial high stands of sea, recorded as uplifted,
coral-reef terraces on New Guinea. These terraces date to ~50,000, ~40,000 or ~30,000 yr B.P. (Chappell et al., 1996; Cutler
et al., 2003). However, based on the U-series ages for corals from the Davenport terrace, it is very likely that the cosmogenic
ages for the older Santa Cruz terraces are underestimates. A reasonable explanation is that the ages reported by Perg et al.



(2001) reflect the ages of alluvium that overlies the marine deposits. The terrestrial sedimentary cover in this area is typically
much thicker than the marine cover and marine sediments are rarely, if ever, exposed at the ground surface. For example, at
Point Año Nuevo, the sea cliff exposes the Davenport platform at ~7.8 m above sea level, overlain by ~0.5 to ~0.2 m of marine
deposits with fossils. However, above the marine deposits are ~9.8 m of alluvial sands and gravels, interbedded with silts and
clays. A well-developed soil, with an A/E/Bt [Bts]/C profile developed in these nonmarine deposits, indicates a substantial
age for this alluvium.

In light of the ~80 ka (MIS 5a) age for the Davenport terrace, a reasonable working hypothesis is that the other platforms

in the Santa Cruz terrace complex date to earlier high-sea stands of MIS 5. Thus, the Highway 1 platform could date to MIS
5c and the Greyhound platform could date to MIS 5e. If this correlation is correct, then the much of the MIS 5e shoreline
(represented hypothetically by the Greyhound platform) has been eroded away. Based on the shore-parallel elevation profiles
of Bradley and Griggs (1976), less than ~4 km of the shore-parallel extent of this terrace still exists along ~32 km of coastline
that they mapped. In contrast, the Highway 1 platform occurs nearly continuously from the city of Santa Cruz northwest to
Point Año Nuevo. If this platform was cut during MIS 5c, then much of the MIS 5e (Greyhound) terrace must have been
removed before much uplift could take place, contrary to the high uplift rates implied by Perg et al. (2001). The challenge
here, as in many places, is to devise a method of dating terraces that lack fossils.
**4.5.2. San Luis Obispo County, California**

In northern San Luis Obispo County, five marine terraces have been mapped by Hanson et al. (1994) in the area around

San Simeon (Fig. 25). These terraces are, in ascending order, the Point (7-8 m), San Simeon (4-23 m), Tripod (23-38 m), Oso
(29-47), and La Cruz (53-79 m) terraces. Terrace elevations vary as a function of proximity to the northwest-trending San
Simeon Fault zone. Unfortunately, fossils are apparently lacking in this area, so there is little age control for any of these
terraces. Hanson et al. (1994) correlated the Tripod terrace to MIS 5e, based on a simple lateral correlation to the low-elevation
marine terrace exposed near Cayucos, to the south (see discussion below). However, this correlation is currently only a
working hypothesis, as the Cayucos terrace is ~35 km distant, and it is not certain that such a long, shore-parallel correlation
can be justified.

West of the town of Cayucos (**RSL ID 3688**) a broad marine terrace extends along the coast for several kilometers, as

discussed earlier (Fig. 6a). This terrace and its deposits are well known, in part because the terrace sediments host a rich
molluscan fauna (Valentine, 1958), but also because it is the first terrace in California where a coral was dated by U-series
(Veeh and Valentine, 1969). The latter workers reported an age of ~130 ka for this coral, which is recalculated here to ~122
ka, using more recent estimates of the half-lives of U-series nuclides. With this correlation to the MIS 5e, the Cayucos terrace
has been an important calibration point for many aminostratigraphic studies (Wehmiller et al., 1977a; Kennedy et al., 1982;
Wehmiller, 1982; Muhs et al., 2014b). The terrace is broad, with a shore-normal extent of up to ~600 m. Although it has
been reported that the shoreline angle of the terrace, at around 7-8 m above sea level, is exposed near the town of Cayucos
itself (Stein et al., 1991), this measurement is actually of the wave-cut bench behind a paleo-sea stack. However, the
measurement is not greatly different from that made by extending a shore-normal topographic profile of the wave-cut bench





landward and finding its possible intersection with an extension of the paleo-sea cliff topographic profile downward (Fig. 6b), which yields a possible shoreline angle elevation of ~8 m (**RSL IDs 3776, 3801**).

Using TIMS U-series methods, Stein et al. (1991) analyzed 12 corals from near LACMIP loc. 10731 (**RSL ID 3688**), near the town of Cayucos (Figs. 6a, 25).  Eleven corals from this locality gave ages ranging from ~125 ka to ~113 ka, with somewhat elevated initial $^{234}U/^{238}U$ values, indicating that the ages are probably biased old to some degree, but still likely correlative to MIS 5e.  However, one coral gave an apparent age of ~101 ka, with an initial $^{234}U/^{238}U$ value that was only slightly elevated, indicating minimal age bias.  Muhs et al. (2002a) revisited the same locality and analyzed seven corals, with four yielding ages of ~123 ka to ~116 ka, in broad agreement with Stein et al. (1991), and with three giving ages of ~110 ka to ~108 ka.  Muhs et al. (2002a) interpreted these data to indicate the possibility that during  MIS 5c at ~100 ka, the high-sea stand overtook at least the outer part of the terrace created during MIS 5e at ~120 ka, with the result that fossils of two ages were mixed together. This interpretation is supported by a reexamination of the fossil fauna reported by Valentine (1958), using updated modern zoogeography.  Although the fauna contains several extralimital southern species of mollusks, as well as some southward-ranging mollusks, it also contains some extralimital northern and northward-ranging species.  This mix of warm-water (~120 ka?) and cool-water (~100 ka?) molluscan forms was interpreted by Muhs et al. (2002a) to be consistent with the apparent mix of ~120 ka and ~100 ka corals.

South of Cayucos, in the Diablo Canyon (**RSL ID 3808**)-Point San Luis (**RSL ID 3777**)-Shell Beach (**RSL ID 3807**) area (Fig. 25), Hanson et al. (1994) mapped a sequence of at least 12 marine terraces, up to an elevation of at least ~200 m.  The lowest two terraces, Q1 (12-4 m) and Q2 (34-12 m) were correlated with MIS 5a and 5e, respectively, using a variety of dating methods.  A U-series age, recalculated here using updated half-lives, on corals from the Q2 at Point San Luis (done by alpha spectrometry) is ~118 ka and supports this correlation (Muhs et al., 1994), as do ages (by TIMS) on corals from Shell Beach that range from ~127 ka to ~122 ka (Stein et al., 1991).  However, at other localities of the Q2 terrace, Hanson et al. (1994) noted inconsistencies between amino acid age estimates of the terrace mollusks and zoogeographic aspects of the faunas.  For example, at some localities, amino acid ratios would imply correlation to MIS 5a, but faunas have a warm-water aspect, implying a correlation to MIS 5e.  At other localities, such as Shell Beach, both amino acid data and faunal data would indicate an MIS 5a age (Hanson et al., 1994), but U-series data by Stein et al. (1991) imply an MIS 5e age.  Some of these issues were also discussed by Wehmiller (1992) and Kennedy (2000), but details of the faunas and amino acid data do not allow for complete resolution of the problem.  At one locality on the Q2 terrace, near Diablo Canyon, Muhs et al. (1994) reported a U-series age on coral (by alpha spectrometry) of ~108 ka, implying correlation with MIS 5c.  If that age is correct, then it is possible that, like Cayucos, the Q2 terrace in this area contains fossils of both MIS 5c and 5e age, which would explain the occurrence of both cool-water (~100 ka) and warm-water (~120 ka) forms on the same terrace.  More work is needed at localities on both the Q1 and Q2 terraces to resolve this problem.

**4.6  Southern California, USA**

Coastal southern California is defined here as that part of the coast that is east or south of Point Conception (Fig. 25).  This point is a major geographic feature, because here the California coast changes from a north-south orientation to an east-west



one. The change in coastal orientation is structurally controlled by the orientation of the San Andreas Fault, which has a major
restraining bend (informally referred to as the "Big Bend") inland of, and to the northeast of Point Conception. Point
Conception also marks a major faunal boundary in the marine invertebrate communities of the Pacific Coast of North America,
long recognized by marine zoogeographers (Valentine, 1966). This has relevance to the interpretation of marine terrace faunas,
discussed in more detail below.

### 4.6.1 Point Conception to Arroyo Hondo

Marine terraces form the coastal plain between Point Conception and Arroyo Hondo (Fig. 25). Upson (1951) was the first
to study these terraces and map them. He also noted the presence of marine fossils in some of the deposits and reported
indentifications of the taxa, based on an examination of his collections by W.P. Woodring. Preliminary age assignments for
low-elevation terraces in this area, based on amino acid ratios, were made by Kennedy et al. (1982). Later, however, Rockwell
et al. (1992) mapped five terraces (I, II, III, IV, and V, from lowest to highest) in the area and Kennedy et al. (1992) provided
new amino acid data and faunal lists from terraces I, II, and III. Terrace I, also called the Cojo terrace has shoreline angle
elevations varying from ~17 m to ~10 m, depending on proximity to local structures. Two bone specimens from deposits
overlying this terrace gave concordant $^{230}$Th/$^{238}$U and $^{231}$Pa/$^{235}$U ages of ~70 ka and ~87 ka, leading Rockwell et al. (1992) to
conclude that these were close minimum-limiting ages and permitted correlation with MIS 5a. Two localities on terrace I have
D/L leucine and valine values in *Saxidomus* that are significantly lower than those of this genus from deposits from terrace III
(~40-30 m), which Rockwell et al. (1992) correlated with MIS 5e (**RSL ID 3687**). The intermediate-elevation terrace II is
correlated with MIS 5c. Faunas in deposits of both terraces I and II have a cool-water aspect, whereas the fauna of terrace III
has a warm-water aspect, supporting these age assignments (Kennedy et al., 1992). Thus far, no corals have been reported in
terrace deposits in this area.

### 4.6.2 Malibu

Marine terrace deposits along the Malibu coast of Los Angeles County have been studied since the landmark paper of Davis
(1933), who named the two most prominent landforms the Malibu (higher) and Dume (lower) terraces. These two terraces,
plus an intermediate one called the Corral terrace, were mapped in detail by Birkeland (1972), who also measured the shoreline
angle elevations. These elevations vary in a shore-parallel sense, ranging from 76-61 m (Malibu), 54-46 m (Corral), and 40-
15 m (Dume). Szabo and Rosholt (1969) analyzed mollusks from the Corral (called "Terrace C" in their paper) and Dume
terraces for U-series isotopes, including $^{231}$Pa and $^{235}$U. They recognized that mollusks are open systems with regard to U-
series isotopes but devised an open-system model of age determination. From this model, they proposed ages of ~154 ka to
~115 ka (average of ~131 ka) for the Corral terrace and ~112 ka to ~95 ka (average of ~104 ka) for the Dume terrace. Szabo
and Rosholt (1969) correlated the Corral terrace to the ~120 ka Rendezvous Hill ("Barbados III") terrace of Barbados and the
Dume terrace to the ~105 ka Ventnor ("Barbados II") terrace of the same island. The open-system model received considerable
criticism from Kaufman et al. (1971), who concluded that mollusks were not suitable for U-series dating, using either closed-
system or open-system approaches. Interestingly, Simms et al. (2016) nevertheless accepted the ~131 ka and ~104 ka mollusk
ages and correlated the Corral and Dume terraces with MIS 5e and 5c, respectively. Kennedy et al. (1982), however, reported





amino acid data from the Dume terrace that suggested correlation with MIS 5e, supported by the presence of a warm-water
fauna, studied earlier by Addicott (1964b). Although more work needs to be conducted on these terraces, it seems likely that
the lowest, Dume terrace (**RSL IDs 3689, 3690**) is the most probable representative of MIS 5e.

**4.6.3 Palos Verdes Hills-San Pedro**

The Palos Verdes Hills, also in Los Angeles County, is an uplifted crustal block with at least a dozen marine terraces (Fig.
26). The crustal block is bounded by faults on its southeast and northern sides. Based on mapping of terraces in a now-classic
study by Woodring et al. (1946), the Palos Verdes Hills was likely an island during some point or points in its history, most
recently during the last interglacial period, or MIS 5e. Woodring et al. (1946) numbered the terraces, from "1" (the lowest
terrace within the city of San Pedro) to "13" (the highest in the Palos Verdes Hills). The marine deposits overlying terrace 1
in San Pedro were referred to as the Palos Verdes Sand (**RSL IDs 3691, 3772, 3773, 3795**) and this unit was regarded as being
of the same age throughout its mapped extent, although it was recognized that there are substantial differences in the faunal
character from place to place (Woodring et al., 1946). To the west, on the Palos Verdes Hills, deposits of all the terraces were
considered by Woodring et al. (1946) to be older than the Palos Verdes Sand, so the lowest elevation terrace there was referred
to as terrace 2. Aminostratigraphic work by Muhs et al. (1992) showed that the Palos Verdes Sand in northern San Pedro is
correlative with terrace 4 (~72 m) on the Palos Verdes Hills and the Palos Verdes Sand in southern San Pedro is correlative
with terrace 2 (~47 m) on the Palos Verdes Hills (Fig. 26). These investigators correlated terrace 4 with MIS 5e and terrace 2
with MIS 5a, based on both aminostratigraphy and terrace faunas (warm-water faunas on terrace 4; cool-water faunas on
terrace 2). Because of these correlations, Muhs et al. (2006) considered that the terrace numbering system of Woodring et al.
(1946) was misleading and instead named terrace 2 the "Paseo del Mar" terrace and terrace 4 the "Gaffey" terrace (Figs. 26,
27). These correlations remained untested until some years later, when corals were recovered from both terraces 2 (Paseo del
Mar terrace) and 4 (Gaffey terrace, **RSL IDs 3771, 3784**). U-series ages by TIMS gave ages of ~80 ka (MIS 5a) from three
localities on terrace 2, and ages of ~119 ka to ~113 ka (MIS 5e) from a single locality on terrace 4 (Muhs et al., 2006). It is
likely that intermediate-elevation terrace 3, found only on the west side of the Palos Verdes Hills, correlates with MIS 5c, but
fossil corals or mollusks from this terrace have not yet been found.

**4.6.4 Newport Bay area**

The Newport Bay area of Orange County, California, south of Los Angeles, has long been known for its highly fossiliferous
marine terrace deposits (**RSL IDs 3692, 3693, 3796, 3820, 3821**). Terraces were mapped by Vedder et al. (1957) and then
remapped by Vedder et al. (1975). The most extensive of these is the area locally referred to as Newport Mesa (Fig. 28).
Grant et al. (1999) measured the shoreline angle elevations of eight terraces in this area and Newport Mesa corresponds to
their "Terrace 2." The shoreline angle elevations of this terrace range from ~32-36 m. A lower-elevation surface (their
"Terrace 1") has shoreline angle elevations ranging from ~19-22 m. From a fossil locality in the eastern part of Terrace 2,
Kanakoff and Emerson (1959) reported what is likely the most abundant marine invertebrate fauna of Pleistocene age on the
Pacific Coast of North America, with at least 500 species of mollusks, corals, bryozoans, brachiopods, echinoids, crabs,
barnacles, and worms. When the suitability of U-series dating of mollusks was still in a stage of assessment, Szabo and Vedder





(1971) attempted dating fossils by this method from the lowest three terraces in the area. As is usually the case with mollusks,
virtually all the specimens analyzed had evidence of open-system histories. Wehmiller et al. (1977a) and later Kennedy et al.
(1982) analyzed mollusks from the area for amino acid geochronology, primarily using the genera *Saxidomus* and *Leukoma*.
They showed that mollusks from half a dozen localities on Terrace 2 likely date to MIS 5e, but at least three localities have
evidence of older, pre-MIS 5e fossils, one of which is the main locality studied by Kanakoff and Emerson (1959). Grant et al.
(1999) conducted TIMS U-series analyses of corals from both Terrace 2 (two localities) and Terrace 1 (one locality). One of
their localities on Terrace 2 is close to the main locality studied by Kanakoff and Emerson (1959) and three analyses of one
*Paracyathus pedroensis* coral colony gave ages of ~124-120 ka, with minimal likely age bias, permitting correlation to MIS
5e. Three *Balanophyllia elegans* samples from the same terrace at another locality gave older apparent ages, but with clear
evidence of an open system history. A single *Paracyathus pedroensis* coral colony from Terrace 1 gave an apparent age of
~106 ka, allowing correlation to MIS 5c. At least two of the localities correlated to MIS 5e by Wehmiller et al. (1977a) are
on what Grant et al. (1999) later mapped as Terrace 1 and correlated to MIS 5c, based on their U-series age from this terrace,
highlighting the need for additional study of these terraces.
**4.6.5 San Diego County**
In the San Diego area (**RSL IDs 3694 to 3701, 3785, 3822**), multiple marine terraces have been documented (Kern and
Rockwell, 1992). The lowest two terraces, mapped by Kern (1977), have received the most attention. These are the Bird Rock
terrace (shoreline angle elevation of ~8 m) and the Nestor terrace (shoreline angle elevation of ~23 m), best exposed along the
west coast of Point Loma (Fig. 29). Both terraces host deposits and fossils that are thought to represent high-energy, rocky
intertidal environments (Kern, 1977). A more quiet-water, "bay" fauna characterizes what is called the Bay Point Formation
at somewhat more protected localities. This formation is considered to be correlative to deposits of the Nestor terrace
(Valentine, 1959; Kern, 1971). Ku and Kern (1974) reported three alpha-spectrometric U-series ages of corals from the Nestor
terrace (~109 ka, ~131 ka, and ~124 ka). Of these, the age of ~109 ka is the only analysis that yielded an initial $^{234}U/^{238}U$
value within the range of modern seawater. Nevertheless, an "average" age of ~121 ka (correlated to MIS 5e) for the Nestor
terrace has been assumed by many subsequent investigators who have used this terrace as a calibration point for amino acid
geochronology (Wehmiller et al., 1977a; Wehmiller and Belknap, 1978; Wehmiller and Emerson, 1980; Emerson et al., 1981;
Kennedy et al., 1982; Wehmiller, 1982; Keenan et al., 1987). Stein et al. (1991) reported somewhat older U-series ages of
individual corals, analyzed by TIMS, ranging from ~145 ka to ~133 ka, and offered the possibility that the Nestor terrace was
not cut during the MIS 5e high-sea stand. The same investigators also reported an age of ~97 ka for the Bird Rock terrace.
Muhs et al. (1994) redated corals from both the Nestor and Bird Rock terraces using alpha-spectrometric U-series analyses,
and reported ages of 126 ±6 ka and 85 ±4 ka, respectively.
In some amino acid studies that have used the Nestor terrace fossils as calibration points (Wehmiller et al., 1977a; Kennedy
et al., 1982), the fauna has been reported to be one characterized by warm-water forms, although no taxa are specifically
mentioned. Using the detailed fauna presented by Valentine and Meade (1961) and Kern (1977), however, Muhs et al. (2002a)
challenged the idea that the Nestor terrace hosts predominantly warm-water species. Although some extralimital southern



forms are present, there are a larger number of northward-ranging species. Muhs et al. (2002a) also reported new TIMS U-series analyses of individual corals from deposits of the Nestor terrace. Nine of these have ages ranging from ~128 ka to ~113 ka, but three corals have ages ranging from ~109 ka to ~98 ka, similar to what was reported for Cayucos, California, discussed earlier. These investigators interpreted the results from both localities to indicate that the deposits at Cayucos and on the Nestor terrace contain fossils representing MIS 5e (with warm-water mollusks) and MIS 5c (with cool-water mollusks).

Elsewhere in the San Diego area, at Torrey Pines State Park (Fig. 25), and near the Mexican border, Wehmiller et al. (1977a) and Kennedy et al. (1982) correlated low-elevation marine terrace deposits to the Nestor terrace using amino acid geochronology. These investigators correlated the Torrey Pines and "border locality" deposits with MIS 5e based on both amino acid ratios and reports of faunas with warm-water aspects (Emerson and Addicott, 1953; Valentine, 1960). There have been, however, no U-series or amino acid studies of the quiet-water Bay Point Formation fossils, so assumed correlations to MIS 5e for these deposits remain hypothetical.

**4.6.6 Channel Islands**

The eight islands off the coast of southern California are called the Channel Islands, because of the proximity of the northern chain of four islands to Santa Barbara Channel (Fig. 25). Of the eight islands, all but Santa Catalina Island are characterized by geomorphically well expressed marine terraces. In addition, some of the most fossiliferous and best-preserved terraces along the entire coast of North America are found on the Channel Islands. Five islands are preserved either in Channel Islands National Park or by The Nature Conservancy and two (San Nicolas Island and San Clemente Island) are owned by the U.S. Navy. Thus, the urban development that has obscured much of the marine terrace geomorphology on mainland California is absent on the islands. Fourteen terraces have been mapped on San Nicolas Island (Vedder and Norris, 1963), to an elevation of ~240 m, and San Clemente Island hosts at least 20 terraces, the highest at an elevation of almost 600 m. Even tiny Santa Barbara Island, which has an area of less than 3 km$^2$, hosts at least five marine terraces (Muhs and Groves, 2018).

The majority of work on last interglacial marine terrace records has been done on the southern islands. Muhs et al. (1994) reported alpha-spectrometric ages of the lowest marine terraces on San Clemente Island (**RSL IDs 3755 and 3800**) and San Nicolas Island (**RSL IDs 3775 and 3813 to 3817**). Their study showed that the 2nd emergent terraces on both islands have deposits hosting fossils that likely date to MIS 5e and the 1st terrace on San Nicolas Island dates to ~80 ka, or MIS 5a. Later, higher precision TIMS U-series analyses confirmed these ages for both San Clemente Island and San Nicolas Island (Muhs et al., 2002a, 2006). More detailed work on San Nicolas Island, however, with both new terrace mapping and new TIMS U-series ages (Muhs et al., 2012), showed that although the 1st terrace is a single landform with deposits dating to ~80 ka, the 2nd terrace is a composite feature, with a broad, lower elevation surface (terrace "2b") and a narrow, higher elevation surface (terrace "2a"). Fossils from terrace 2a date only to MIS 5e and do not contain cool-water mollusks, but fossils from terrace 2b date to both MIS 5e (~120 ka) and MIS 5c (~100 ka) and contain a mix of mollusks with both warm-water and cool-water aspects, similar to what was reported for Cayucos and the Nestor terrace at Point Loma. Muhs et al. (2012) interpreted these results to indicate that the MIS 5c high-sea stand was high enough and the uplift rate on San Nicolas Island was low enough that much of the MIS 5e terrace (2a) was removed by sea cliff retreat at ~100 ka, and fossils from both the ~120 ka and ~100





ka sea stands were mixed into the deposits of terrace 2b. With these new findings in mind, Muhs et al. (2014a) examined the faunal record of the MIS 5e terrace on San Clemente Island, which is also a composite landform (i.e., two platforms, 2a and 2b, as on San Nicolas Island). This investigation showed that the "MIS 5e" terrace deposits on this island also contain a mix of both warm-water and cool-water fossils, chiefly mollusks, that also imply a mix of MIS 5e (warm) and MIS 5c (cool) taxa. This interpretation also explains a previously enigmatic molluscan oxygen isotope record, implying cooler waters in what had been thought to be solely ~120 ka deposits (Muhs and Kyser, 1987).

The largest of the northern Channel Islands (San Miguel, Santa Rosa, and Santa Cruz) also have marine terraces that date to MIS 5e. TIMS U-series analyses give ages of ~120 ka for the 2nd emergent terraces (shoreline angle elevations of ~20-24 m) on San Miguel (**RSL ID 3778**) and Santa Rosa (**RSL IDs 3779 to 3782**) Islands (Muhs et al., 2014b). As is the case on San Nicolas and San Clemente Islands, the fossil faunas from the deposits of these terraces host both warm-water and cool-water taxa (Orr, 1960; Muhs et al., 2014b), although currently there is only sparse geomorphic evidence of two high sea stands (i.e., both MIS 5e and 5c). At one locality on Santa Rosa Island, there is a marine terrace with an outer edge at ~7 m above sea level with an uncertain shoreline angle elevation, as the inner part of the terrace is covered by eolian sand. Apparent TIMS U-series ages of corals from this terrace range from ~113 ka to ~110 ka and all have slightly elevated initial $^{234}U/^{238}U$ values, indicating at least some bias to older ages (Muhs et al., 2015). Further, mollusks from this terrace (SRI-1 on Fig. 12) have lower amino acid ratios than those in the 24-m-high terrace (SRI-5F on Fig. 12) dated to ~120 ka elsewhere on the island. Thus, it is possible that this isolated terrace fragment represents an MIS 5c record, but more work is needed to confirm this. On both islands, there is a lower elevation terrace with a shoreline angle elevation of ~3 m (San Miguel Island) and ~7 m (Santa Rosa Island). TIMS U-series analyses give an age of ~80 ka for this terrace on Santa Rosa Island and amino acid ratios in mollusks indicate a similar age for the 3-m-high terrace on San Miguel Island (Muhs et al., 2015, 2018). The lowest-elevation terrace on Santa Cruz Island (**RSL IDs 3789, 3790, 3811, 3812**) has shoreline angle elevations ranging from ~6 m to ~17 m and both U-series ages of corals and amino acid ratios in mollusks indicate that it dates to MIS 5e (Pinter et al., 1998; Muhs and Groves, 2018). Thus far, there is no evidence of terraces dating to MIS 5c or 5a on Santa Cruz Island, suggesting that the long-term uplift rate on this island is relatively low.

The two smallest of the Channel Islands, Anacapa Island (**RSL ID 3791**) and Santa Barbara Island (**RSL IDs 3792, 3793**), both have low-elevation terraces, with shoreline angle elevations of ~10-11 m above sea level. No coral ages are yet available for these terraces, but amino acid ratios indicate a mixed population of mollusk ages, correlated to MIS 5e and either MIS 5c or MIS 5a (Fig 12). The terrace fauna on Santa Barbara Island is diverse, with a large number of warm-water forms and a smaller number of cool-water forms (Lipps et al., 1968; Muhs and Groves, 2018). The much more sparse fauna on Anacapa Island hosts some warm-water forms, but only one northward-ranging species.

Santa Catalina Island's marine terrace record, or to put it more accurately, its apparent lack of a record, has been an enigma for decades (see Smith, 1933, for one of the earliest discussions). The island is situated between crustal blocks to the north (Palos Verdes Hills) and south (San Clemente Island) that both host abundant marine terraces. Bedrock is not a limiting factor, because the Catalina Schist that characterizes much of the island is similar to that of the Franciscan rocks that host marine



terraces elsewhere in central and northern California (e.g., Cayucos, Fig. 6). In support of this, Emery (1958) showed that submarine terraces are found off Santa Catalina Island. At a minimum, even with no uplift, with the likelihood of a higher than present sea level during MIS 5e, one should expect to see some evidence of a terrace that dates to ~120 ka. One hypothesis that has been offered is that the island is subsiding and that the terrace record is largely submerged (Castillo et al., 2018). However, there is no a priori reason to suppose that in a tectonic setting similar to those elsewhere in southern California that Santa Catalina Island should be subsiding when all adjacent areas are uplifting. Schumann et al. (2012) presented evidence that in fact the opposite is true, i.e., Santa Catalina Island is experiencing uplift, possibly at a high enough rate that fluvial erosion has removed most evidence of any terraces. This remains the most viable explanation to date, but more work is needed to confirm this.

**4.7 Baja California, Pacific Coast**

South of the city of Ensenada, Baja California, there is a prominent peninsula called Punta Banda (Fig. 30). Rockwell et al. (1989) mapped several marine terraces on this peninsula and provided alpha-spectrometric U-series ages of corals and hydrocorals from the lowest terraces. The 3rd terrace, called the Sea Cave terrace (**RSL IDs 3786, 3802**), has a shoreline angle elevation that varies between ~34 m and ~40 m (Fig. 31). The 1st terrace, called the Lighthouse terrace, has a shoreline angle elevation that varies between ~15 m and ~18 m. An intermediate, unnamed terrace at ~22 m elevation occurs only in a small area on the outer part of the peninsula. U-series ages indicate that the Sea Cave terrace is ~120 ka (MIS 5e) and that the Lighthouse terrace is ~80 ka (MIS 5a). The intermediate terrace at ~22 m could represent MIS 5c, but no corals were found on the terrace. Muhs et al. (2002a) reported new, TIMS U-series analyses of corals from both the Sea Cave and Lighthouse terraces that confirm the earlier ages generated by alpha spectrometry. Many of the corals from Punta Banda show minimal age bias, based on back-calculated initial $^{234}$U/$^{238}$U values (Fig. 10c). Corals from the Sea Cave terrace that show mostly closed-system histories have ages ranging from ~124 ka to ~118 ka, and those from the Lighthouse terrace range from ~83 ka to ~80 ka.

Isla Guadalupe is situated ~260 km southwest of the Pacific coast of Baja California (Fig. 2). The island is attractive as a reference locality for estimating paleo-sea level during the last interglacial period because it is one of the few localities adjacent to North America that can be considered, *a priori*, to be tectonically stable. The island is distant from any plate boundary, has no active faults nearby, has no active volcanoes on it or near it, is bounded on its eastern side by a seafloor with undisturbed marine sediment, and has no history of recent earthquakes (Gonzalez-Garcia et al., 2003). Lindberg et al. (1980) reported that emergent marine deposits are found on the southern and eastern of the coasts of the island. These deposits have elevations of ~1 m to ~8 m above sea level, with most localities described as ~1 m to ~6 m above sea level (**RSL IDs 3803 to 3805**). From these deposits, Muhs et al. (2002a) reported ages of ~123 ka to ~118 ka for *Pocillopora* corals and most show closed-system histories. An extensive faunal list by Lindberg et al. (1980) indicates that the ~120 ka deposits host a large number of extralimital species of mollusks. In addition, Isla Guadalupe marks, thus far, the northernmost occurrence of hermatypic corals along the Pacific Coast of North America during MIS 5e (Durham, 1980).





South of Punta Banda, marine terraces are prominent landforms all along the coast of Baja California.  In a coastal reach
from Punta Banda south for at least 300 km, multiple marine terraces are present, mapped by Orme (1980) (Fig. 30).  Many
low-elevation marine terrace deposits along this reach of coast are highly fossiliferous (Emerson, 1956, 1960; Emerson and
Addicott, 1958; Addicott and Emerson, 1959; Valentine, 1960a, 1961) and are candidates for records as MIS 5e shorelines.
Unfortunately, little work has been done on age determinations for most of these terraces.  What geochronologic work has
been done along the Pacific coast of both Baja California and Baja California Sur is aminostratigraphic correlation, using U-
series-dated localities, such as the Nestor terrace to the north, and a single locality to the south, Bahía Magdalena (see
Wehmiller and Emerson, 1980).  Thus, before reviewing the results of the aminostratigraphic correlations, the work done at
Bahía Magdalena is discussed first.
Bahía Magdalena (**RSL IDs 3707, 3798**) is situated on the Pacific side of Baja California Sur (Fig. 32) and marine deposits
there have long been famous for their extensive Pleistocene fauna (Jordan, 1936).  Near the village of Puerto Magdalena on
the peninsula, one of Jordan's (1936) fossil sites (California Academy of Sciences [CAS] locality 754) contains fragments of
the colonial coral *Porites californica* (now considered to be *P. panamensis*).  These corals occur in marine terrace deposits
that have a *maximum* elevation of ~6 m and have been dated to the last interglacial (LIG) period (~118 ka to ~116 ka) by
alpha-spectrometric uranium-series methods (Omura et al., 1979).  When the data of Omura et al. (1979) are recalculated using
the more recent estimates of half-lives (Cheng et al., 2013), the coral ages from Bahía Magdalena are ~114.8 ka to ~114.0 ka.
The northernmost locality on the Pacific coast of Baja California that has been examined for paleontology and
aminostratigraphic correlation is a low-elevation (outer edge elevation of ~5 m to ~10 m) terrace at Camalú (**RSL ID 3702**,
Fig. 30), studied by Valentine (1980).  Amino acid ratios in *Leukoma* shells from the deposits of this terrace led Valentine
(1980) to conclude that the terrace dates to MIS 5e.  It is important to note here that although Valentine (1980) did not report
actual amino acid ratios for Camalú, his interpretation is supported by data presented graphically by Keenan et al. (1987).
Faunal data make an MIS 5e interpretation for the terrace at Camalú complicated, however, as there are several northward-
ranging species of mollusks and only one southward-ranging species (Valentine, 1980).  Approximately 300 km south of
Camalú, at Punta Santa Rosalíllíta (**RSL ID 3962**), Woods (1980) mapped three emergent marine terraces, named Tomatal
(shoreline angle of ~7 m), Andres (~25-30 m), and Aeropuerto (~50-60 m).  Although there are no U-series ages on corals
from these terraces, Woods (1980) reported amino acid data from fossil mollusks that correlate the Tomatal terrace with the
peak of the LIG at ~120 ka.  As is the case at Camalú, data presented graphically by Keenan et al. (1987) support this
interpretation.
Farther south on the Pacific coast of northern Baja California Sur, at Bahía Tortugas and Bahía Asunción (Fig. 33), Emerson
et al. (1981) and Keenan et al. (1987) reported amino acid data for low elevation terraces that they correlated to MIS 5e.  Two
terraces are present at Bahía Tortugas (**RSL IDs 3703, 3704**), one at ~27-24 m and the other at ~12 m, although it is not clear
if these elevations refer to shoreline angles or simply the fossil localities that were studied.  In any case, amino acid ratios
clearly distinguish the two terrace deposits, with the higher elevation terrace attributed to MIS 5e, based on aminostratigraphic
correlation between Bahía Magdalena to the south and the Nestor terrace to the north.  The lower terrace, with lower ratios, is





considered to be "~95 ka," but correlation to either MIS 5c or MIS 5a is possible.  Deposits of both terraces contain *both* warm-
water and cool-water species, but the upper terrace contains substantially more warm-water forms than the lower terrace.
At Bahía Asunción (**RSL IDs 3705, 3706**), Keenan et al. (1987) identified three age groups of marine terrace deposits
based on amino acid ratios.  The youngest of these is correlated with MIS 5e, based on aminostratigraphic correlation to Bahía
Magdalena.  The deposits correlated to MIS 5e are situated ~6 m above sea level at one locality and ~11-12 m above sea level
at another locality.  However, deposits hosting what these investigators considered to be older deposits, based on amino acid
ratios, have elevations that fall within the same general range as those correlated to MIS 5e, so additional work in this area is
warranted to clarify the age-elevation relations.
**4.8 Golfo de California coasts of Baja California, Baja California Sur, and Sonora, Mexico**
As discussed above, waters in the Golfo de California are distinctly warmer than those in the Pacific Ocean along the west
coast of Baja California and Baja California Sur (Mitchell et al., 2002).  Thus, the potential for finding coral-bearing marine
deposits or even true coral reefs is greater in this region than on the outer coast of Baja California and Baja California Sur.
Cabo Pulmo is located in the southernmost part of Baja California Sur (Fig. 32), adjacent to the Golfo de California.  Ortlieb
(1987) mapped emergent marine terraces near Cabo Pulmo, as well as to the southwest, towards Cabo San Lucas, and to the
north.  Squires (1959) described a coral-bearing marine terrace deposit near Cabo Pulmo (**RSL ID 3806**), which Ortlieb (1987)
reported as having a shoreline angle elevation of ~6 m.  Muhs et al. (2002a) reported three TIMS U-series analyses of *Porites*
and *Pocillopora* corals from this deposit.   The *Pocillopora* colony gave an apparent age of ~140 ka, but is clearly biased old,
based on an elevated initial $^{234}U/^{238}U$ value.  On the other hand, *Porites* corals from these deposits gave ages of ~127 ka and
~120 ka and have only slightly elevated initial $^{234}U/^{238}U$ values.  Thus, it is clear that this deposit represents MIS 5e.
North of Cabo Pulmo, Isla Cerralvo (**RSL ID 3832**) is situated off the eastern coast of Baja California Sur.  Tierney and
Johnson (2012) studied a section 8.7 m thick on southwestern tip of the island, composed of alternating layers of growth-
position corals (Fig. 32) and cobbles, the latter interpreted to be from storm transport.  Five coral-cobble cycles are represented
by the layers in this section, all interpreted to represent a single interglacial period.  Tierney and Johnson (2012) report a U-
series age of ~126 ka from one of the *Porites* coral colonies found in the section.  The highest growth-position reef layer is at
an elevation of ~3.9 m, overlain by sands interpreted to be from a prograding beach, up to an elevation of 7.1 m.  Coral-bearing
sediments (interpreted here to be from storm deposits) occur as high as ~8.7 m above sea level.
Marine terrace deposits are exposed on both sides of Punta Coyote (**RSL IDs 3828 to 3831**), Baja California Sur (Fig. 32).
U-series analyses, done by alpha spectrometry, have been conducted on both *Porites* and *Pocillopora* corals recovered in low-
elevation terrace deposits here, reported by Sirkin et al. (1990) and Szabo et al. (1990).  Apparent ages of corals from each of
five localities would allow correlation of the deposits to MIS 5e, but one coral has a back-calculated initial $^{234}U/^{238}U$ value that
is higher than modern seawater and three have initial $^{234}U/^{238}U$ values that are *lower* than modern seawater, an unusual
situation, and how this affects apparent ages is not known.  Based on what general information is given, it appears that these
deposits have inner edge elevations that may be on the order of ~8 to ~10 m above sea level.  Farther north, at Bahía Coyote
(**RSL IDs 3826 and 3827**), DeDiego-Forbis et al. (2004) reported U-series ages of *Porites* corals from terraces exposed along



the coast. The highest elevations of what appear to be growth-position *Porites* colonies are estimated to be ~18 m to ~22 m above modern sea level. At least four of the corals analyzed appear to have experienced gain of bulk U, but two corals analyzed have acceptable U contents and apparent ages of ~138 ka. Because both of these corals have initial $^{234}$U/$^{238}$U values that are higher than modern seawater, both are biased old by some amount, but permit correlation of the terraces to MIS 5e.

Studies by Johnson (2002) and Johnson et al. (2007) provide data on last interglacial coral reefs at two localities in Baja California Sur, Isla Coronado, and Punta Chivato (Fig. 32a, b, c). TIMS U-series analyses were conducted on these corals in laboratories of the U.S. Geological Survey, and complete analytical data are given in Muhs et al. (2014b). At Isla Coronado (**RSL ID 3818**), Johnson et al. (2007) report that a *Porites panamensis* colony formed one of the largest fossil structures yet reported in the Golfo de California. The top of the coral reef surface is ~12 m above sea level. Analysis of coral from this reef gave an age of ~127 ka, with an initial $^{234}$U/$^{238}$U value higher than modern seawater. Thus, the age is likely biased old by some amount, but still allows correlation to MIS 5e. At Punta Chivato (**RSL ID 3819**), a *Porites panamensis* colony ~15 cm high, in growth position (Fig. 32), is situated on Pleistocene river gravels at a present elevation of 7.5 m to 10 m above sea level (Johnson, 2002; Johnson et al., 2007). U-series analysis of a *Porites* sample from this colony gave an age of 117.7 ka, with an initial $^{234}$U/$^{238}$U value indistinguishable from modern seawater, giving a high degree of confidence that this deposit correlates with MIS 5e. Just south of Punta Chivato, a low-elevation marine terrace is present along the coast near Mulegé (**RSL IDs 3823 and 3824**, Fig. 32). Rather than a constructional coral reef, this landform appears to be a California-style marine terrace, with a wave-cut platform and overlying deposits that contain corals. The terrace has a shoreline angle elevation of ~12 m above sea level and two alpha-spectrometric U-series analyses gave ages of ~146 and ~124 ka, with initial $^{234}$U/$^{238}$U values only slightly higher than modern seawater (Ashby et al., 1987). Both corals have somewhat lower than optimum $^{230}$Th/$^{232}$Th values, suggesting the possibility of some inherited $^{230}$Th, which would bias the apparent ages older. Nevertheless, it is likely that the terrace correlates with MIS 5e, as concluded by Ashby et al. (1987). Between Punta Chivato and Mulegé, ~15 km north of the latter locality, Libbey and Johnson (1997) listed an extensive (>40 species) fossil molluscan fauna from a terrace that they report can be traced to, or nearly to Punta Chivato. Many of these taxa have modern ranges that extend from the upper part of the Golfo de California to southern Mexico, Panama, Ecuador, or Peru, indicating a likely marine paleotemperature range at least as warm as that of the present.

North of Punta Chivato, coral-bearing marine terrace deposits have not been reported on either coast of the Golfo de California. Nevertheless, molluscan-rich terrace deposits are common and permit the possibility of amino acid geochronology. By far the most extensive studies of these deposits are those by Ortlieb (1987, 1991). Because of the relatively high mean annual air temperatures in the Golfo de California (~20°C in the north ranging to ~23°C in the south), even amino acid geochronology becomes problematic in identifying deposits that correlate to MIS 5e. The reason for this is that many species of mollusks will have reached, or be close to, racemic equilibrium for most amino acids after ~120 ka. Ortlieb's (1987, 1991) approach to this problem was to consider that shells (the bivalves *Chione* and *Dosinia*) that were beyond radiocarbon range, but yielded amino acid ratios not yet at equilibrium (but close to it) could be interpreted to be of MIS 5e age. This method is supported by his analyses of shells from both Bahía Magdalena, on the Pacific side of Baja California Sur and at Bahía San





Nicolas (**RSL IDs 3741, 3742**), on the Golfo de California side of Baja California (Figs. 32, 33), where U-series ages on corals
have been obtained (Omura et al., 1979; Ortlieb, 1987). His interpretations are supported by more recent amino acid data on
*Chione* reported by Umhoefer et al. (2014), calibrated to U-series data reported by DeDiego-Forbis et al. (2004) from Bahía
Coyote (Fig. 34). Also shown in this figure are D-alloisoleucine/L-isoleucine values in radiocarbon-dated *Chione* shells of
late Holocene age, from Cholla Bay, Sonora (Martin et al., 1996) and a blue-shaded band that defines the equilibrium range
for D-alloisoleucine/L-isoleucine (1.25-1.35; Miller and Mangerud, 1985). When all data are considered, it is apparent that
shells falling within a range of ~0.70 to ~1.00 can be correlated to U-series-dated 120 ka localities. This range of values is
substantially lower than the equilibrium range of 1.25-1.35, but considerably higher than the range of values in Holocene
shells, ~0.02 to ~0.12. Ortlieb's (1987, 1991) results, along with those by Umhoefer et al. (2014) show, therefore, that an MIS
5e shoreline can be traced from the uppermost Golfo de California, south along the coasts of Baja California and Sonora, for
at least ~850 km. Most of the MIS 5e shorelines studied by Ortlieb (1987, 1991) have relatively low elevations, ranging from
~2 m to ~8 m above sea level (Table S2).
**4.9  Pacific Coast of southern Mexico**
Although corals presently flourish along the Pacific coast of southern Mexico, south of the Golfo de California (López-
Pérez, 1998), there are only scattered reports of fossil corals as reefs or in emergent terrace deposits along this reach of
coastline. Emergent marine terraces or fossil coral reefs have been reported from both mainland Mexico, south of Oaxaca
(Palmer, 1928a,b; Squires, 1959) and offshore Tres Marias Islands (Fig. 7) (Hertlein and Emerson, 1959; Foose, 1962). There
do not appear to have been any recent studies of these deposits, nor are any geochronological data available. Given the
elevations that are described in these studies, however, as well as the fossil records, they are candidates as MIS5e shoreline
records and deserve further study.
**4.10  Central America**
The tectonic setting of the Pacific coast of Central America differs from that of Mexico to the north. Here, both the Cocos
and Nazca plates are being actively subducted under either the North America plate or the Caribbean plate (Figs. 1, 35a).
Furthermore, the Panama triple junction is situated offshore, just south of the Costa Rica-Panama border (Fig. 35a), making
this area tectonically and structurally complex.
Ocean temperatures are, in principle, warm enough to support hermatypic coral reef growth off the Pacific coast of Central
America from Guatemala to Panama. Toth et al. (2017) point out, however, that between Mexico and Nicaragua, there are
very few if any true coral reefs on the Pacific coast, in what is referred to as the "Central American faunal gap" (Fig. 7). These
investigators hypothesize that the lack of modern coral reefs along this reach of coast may be a function of a lack of hard
substrate available for larval settlement, although severe and prolonged upwelling has also been offered as a contributing factor
(Glynn et al., 2017). In all likelihood, this also limits the potential for finding emergent coral reefs that date to MIS 5e as well.
Farther south, along the coasts of Costa Rica and Panama, modern coral reefs are much more common and many developed
as early as ~7,000 yr ago (see summary in Toth et al., 2017). This is also an area where subduction of the Cocos plate beneath
the Caribbean plate is currently active and uplift rates are high (Gardner et al., 1992; Marshall and Anderson, 1995). Much of





the work on emergent marine terraces here has been focused on the Nicoya Peninsula and Osa Peninsula of Costa Rica, and
the Burica Peninsula of Panama (Fig. 35a). Gardner et al. (1992) reported marine terraces of Holocene age (~7000 yr to 980
yr) at elevations of ~3 to ~9 m on the Osa Peninsula and Marshall and Anderson (1995) report Holocene marine terraces with
ages of ~4700 yr to ~500 yr at elevations of ~4 m to ~16 m on the Nicoya Peninsula, demonstrating that the uplift rate in this
region of active subduction is relatively high. Fisher et al. (1998), Gardner et al. (2001), and Sak et al. (2004) all pointed out
that uplift in this region is controlled primarily by roughness of the subducting plate: forearc uplift on the Caribbean plate
corresponds to the position of migrating seamounts on the northeastward-moving Cocos plate.
With such high uplift rates, based on the elevations of Holocene marine terraces, any MIS 5e marine terraces on the Pacific
coast of Costa Rica would have to be at relatively high elevations now. On the Nicoya Peninsula, Marshall and Anderson
(1995) recognized two marine terraces. The younger of these is the suite of Holocene marine deposits, called the "Cabuya"
terrace. The higher terrace, called the "Cobano" terrace, is a broad coastal mesa, situated at an average elevation of ~180 m,
and is hypothesized to have formed during MIS 5e, although no geochronologic data are presented in support of this (Marshall
and Anderson, 1995). Using Holocene uplift and rotation rates, Gardner et al. (2001) estimated the Cobano terrace to have
formed between ~200 ka and ~100 ka, also permitting an interpretation of an MIS 5e age.
On the Osa Peninsula of Costa Rica, Gardner et al. (2013) mapped Quaternary marine deposits of three ages, from youngest
to oldest, the Jiménez (Holocene), Tigre (MIS 3?), and Rincón (MIS 5?) members of what they called the Marenco Formation
(Fig. 35b). They reported an OSL age of 109 ±28 ka for deposits of the Rincón member and correlated this unit to MIS 5e.
Based on marine terrace shell radiocarbon ages, the Jiménez member dates to the Holocene. Gardner et al. (2013) also dated
shells from the Tigre member, which, when calibrated, range from ~31 ka to ~48 ka with a few samples yielding apparently
infinite ages. They correlated the Tigre member with MIS 3.
Gardner et al.'s (2013) correlation of the Tigre member to MIS 3, based on their radiocarbon ages, requires some scrutiny.
A critical examination applies to similar radiocarbon ages reported by Gardner et al. (1992) and Sak et al. (2004), also on the
Osa Peninsula, as do ages reported by Morell et al. (2011) for terrace shells on the Burica Peninsula of Panama (Fig. 35a).
Indeed, some of the marine terrace radiocarbon ages reported by Morrell et al. (2011) date not only to MIS 3, but actually give
apparent ages dating to the late last glacial period (MIS 2), at a time when sea level was several tens of meters below present.
Emergent marine deposit shells giving apparent radiocarbon ages of MIS 3 age have been reported on coastlines in various
parts of the globe for decades, with some investigators claiming that such ages require a paleo-sea level close to, or even above
present sea level during this interstadial period. Periodically, there have been critiques of such claims (Thom, 1973; Bloom,
1983; Colman et al., 1989), but many investigators continue to regard shell radiocarbon ages of ~30 ka to ~45 ka as truly finite
and accurate. The problem is that modern carbon is nearly everywhere and has considerable mobility. This means that old
shells are notorious for incorporating at least small amounts of modern carbon. Thus, even very small amounts of modern
carbon can make an infinitely old shell yield an apparently "finite" radiocarbon age (Pigati et al., 2007). An 80-ka sample, for
example, with a very small amount of modern carbon, can easily yield an apparent radiocarbon age of ~40 ka to ~45 ka. More



work needs to be done on the ages of the older marine terrace fossils of both the Osa Peninsula of Costa Rica and the Burica
Peninsula before any inferences about terraces of either MIS 3 or MIS 5 age can be made.
**5 Last Interglacial sea level fluctuations**
One of the issues that has been actively debated in the past few decades is whether MIS 5e was characterized by a single
sea-level high stand or multiple high stands. Some of the original evidence for more than one high stand came from the Huon
Peninsula of New Guinea, where reefs VIIa and VIIb were interpreted to represent early and later high-sea stands of MIS 5e,
respectively (Bloom et al., 1974; Chappell, 1974). U-series dating of corals from these two terraces using TIMS methods
confirmed that both terraces likely date to MIS 5e (Stein et al., 1993). Such a record on New Guinea does not, however,
require that this was a global phenomenon, because coseismic uplift has been well documented for this coast, with as many as
six coral reefs emerging in the Holocene alone (Ota et al., 1993). On Barbados, the uplift rate is much lower than that on New
Guinea, but more than one high stand of sea during MIS 5e has been proposed here as well (Schellmann and Radtke, 2004;
Thompson and Goldstein, 2005). Unlike New Guinea, however, multiple Holocene terraces have not been reported on
Barbados, and coseismic uplift is a less likely explanation for possible multiple LIG terraces.
In addition to tectonically active coastlines, there have been claims of multiple sea stands during MIS 5e from deep-sea
records and reefs on tectonically stable coastlines. Rohling et al. (2008), studying the oxygen isotope record in planktonic
foraminifera recovered from Red Sea sediment cores, suggested that there could have been as many as four separate high
stands of sea during MIS 5e. Thompson et al. (2011) reported TIMS U-series ages of corals from San Salvador Island and
Great Inagua Island in the Bahamas, proposing at least two high stands during MIS 5e, and possibly as many as four high
stands, similar to the Red Sea record of Rohling et al. (2008).
Modeling efforts have also addressed the question of a dual high-sea stand during the LIG. Kopp et al. (2009) conducted
a statistical analysis of a database generated from many reported MIS 5e deposits worldwide, from both tectonically active
and stable coastlines. These investigators concluded that early within MIS 5e there was a sea-level high, followed by a drop
of ~4 m, succeeded by another sea-level high. Unfortunately, some of the hypothesized MIS 5e sites used by Kopp et al.
(2009) are either poorly dated or not dated at all, rendering this reconstruction uncertain. In a more recent review of both field
and modeling evidence, Barlow et al. (2018) concluded that there is no evidence of more than one high-sea stand during MIS
5e. Along the Pacific Coast of North America, there has been, thus far, no evidence of more than one high-sea stand during
MIS 5e, along either tectonically stable or uplifting coasts.
**6 Other interglacials**
**6.1 Interglacials prior to MIS 5e**
Because of ongoing tectonic processes during the Quaternary, multiple marine terraces are recorded along much of the
Pacific Coast of North America, from southern Oregon to Baja California. As noted earlier, Woodring et al. (1946) mapped
13 marine terraces in the Palos Verdes Hills (Fig. 26), the highest of which is at an elevation of ~400 m. Vedder et al. (1957,





1975) and Grant et al. (1999) documented at least six marine terraces above the Newport Mesa terrace, correlated to MIS 5e,
in the Newport Beach area.  Vedder and Norris (1963) mapped 14 marine terraces on San Nicolas Island (Fig. 36a), with the
highest at an elevation of ~270 m.  Fossils are found in deposits of all 14 terraces.  On this island, amino acid ratios in fossil
*Tegula* specimens show a steady increase with terrace elevation (Fig. 36b).  By the time the 8th and 10th terraces are reached,
D-alloisoleucine/L-isoleucine values in *Tegula* are at equilibrium values of ~1.25, indicating considerable antiquity.  San
Clemente Island hosts more than 20 marine terraces, and these landforms show superb geomorphic preservation (Fig. 37).
Fossil-bearing marine terrace deposits are found as high as ~265 m (Cockerell, 1939), similar to San Nicolas Island, and the
highest marine terrace is found at an elevation of nearly 600 m.  If the late Quaternary uplift rate has been steady over the
history of the island (Muhs et al., 2014a), the highest terrace on San Clemente Island could be ~3 Ma.  Even some of the
smallest islands off the California coast host a long-term history of interglacial high-sea stands superimposed on steady uplift.
Santa Barbara Island has an area of only ~2.6 km$^2$, yet it hosts at least five marine terraces, up to an elevation of ~100 m (Fig.
38).  On many of the California islands, pre-MIS 5e marine terraces are distinguished from younger terraces by the presence
of the extinct fossil gastropod *Pusio fortis* (formerly *Calicantharus fortis*).  For example, on Santa Barbara Island, this taxon
is found in deposits of the 2nd, 3rd, and 4th terraces (Fig. 38), but is not found in deposits of the 1st terrace, which appears to
contain a mix of fossils dating to MIS 5e and MIS 5c (Muhs and Groves, 2018).  Multiple marine terraces are found along the
Pacific coast of Mexico as well.  Rockwell et al. (1989) recognized 14 marine terraces on Punta Banda, in northern Baja
California, with the highest at an elevation of ~347 m.  Farther south, Orme (1980) mapped multiple marine terraces, with the
highest between Cabo San Quintin and Punta Baja (Fig. 30), at an elevation of ~300 to ~357 m.

Unfortunately, there are few data on the possible ages of pre-MIS 5e terraces on the Pacific Coast of North America.

Indeed, numerical ages of terraces dating from MIS 7, 9, and 11 have yet to be confirmed for any part of the Pacific coast of
the continent, although it is likely that marine terraces representing these high-sea stands are preserved.  Although corals are
present in deposits of several higher elevation terraces, open-system histories have likely prevailed in many of these fossils.
For example, Muhs et al. (2004) presented U-series data for corals from the 10th terrace (elevation ~236 m) on San Nicolas
Island, indicating possible ages of ~600 ka to ~450 ka.  These apparent ages are, however, not consistent with the late
Quaternary uplift rate, nor are they consistent with amino acid ratios at equilibrium values in fossil mollusks from this terrace
(Fig. 37b).  A more promising isotopic method of age determination for fossils of pre-MIS 5e terraces on the Pacific Coast is
Sr isotope stratigraphy, a calibrated method of geochronology.  Early experiments with this method in California showed
promise (Ludwig et al., 1992) and since that time, better calibration curves have been developed (Howarth and McArthur,

1997).

Latitudinal, north-south-trending aminozones parallel or subparallel to MIS 5a and MIS 5e aminozones, show the potential

for at least lateral correlation of older, pre-MIS 5e marine terraces.  Wehmiller (1982) used such an approach on the Pacific




Coast of North America, from southern Baja California Sur to Oregon. His data showed the possibility for marine records
prior to MIS 5e, including high-sea stands associated with MIS 7, 9, 11, 13, and 15.
Another possibility for dating older terraces is the use of kinetic modeling with amino acid ratios. In this approach, a
theoretical kinetic pathway is used with a calibrated amino acid ratio for shells from a deposit that is independently dated (such
as by U-series on coral). Clarke and Murray-Wallace (2006) review the various mathematical expressions for different kinetic
pathways. One of the most widely used method is the parabolic kinetic model, derived from heating experiments that simulate
long periods of geologic time (Mitterer and Kriausakul, 1989). When this method is applied to the terrace sequence on San
Nicolas Island (Fig. 37b), using the ~120 ka age for terrace 2a and its D-alloisoleucine/L-isoleucine value of 0.52, the higher
amino acid ratios for the older terraces yield apparent ages of ~375 ka (terrace 4), ~480 ka (terrace 5), ~510 ka (terrace 6), and
≥680 ka for terraces 8 and 10. If these ages are correct, they would permit correlation of terrace 4 with MIS 11 and terraces 5
and 6 with MIS 13. Along the terrace transect shown in Figure 37b, this would also imply that terraces that formed during
MIS 7 and MIS 9 were likely removed by erosion during MIS 5e. Interestingly, the ages and uplift rates derived from older
terraces in this exercise are similar to the uplift rate derived from the MIS 5e terrace. While all of these implied results seem
reasonable geologically, it is important to remember that kinetic modeling of amino acid racemization and epimerization is
still theoretical and age estimates derived from such an approach are simply possibilities for additional testing.
Still another method to address the question of ages of older marine terrace deposits is the use of cosmogenic isotopes. In
the San Diego region, one of the oldest marine terrace deposits is called the Clairemont terrace, part of a larger complex of
marine terrace and beach ridge deposits called the Lindvista terrace sequence. Based on data in Lajoie et al. (1991), as many
as 13 terraces occur above the MIS 5e terrace, each with wave-cut benches and prominent beach ridges. The Clairemont
terrace is found at an elevation of ~96 m above sea level, and Simms et al. (2020) used cosmogenic nuclides at two localities
to estimate an age of ~1.48 Ma for this terrace.

### 1218 6.2 High-sea stands after MIS 5e

Whereas ages of pre-MIS 5e marine terraces on the Pacific Coast of North America are rare, there are several marine terrace
ages that postdate the peak of the last interglacial period, mostly for the relatively high-sea stands of MIS 5c (~100 ka) and
MIS 5a (~80 ka). With regard to MIS 5c, TIMS U-series ages of corals dating to this high-sea stand have been confirmed, but
mixed with MIS 5e deposits at Cayucos, Point Loma, and San Nicolas Island, as discussed above (Stein et al., 1993; Muhs et
al., 2002a, 2012). In two other areas, both of which have somewhat higher late Quaternary uplift rates, there are terraces that
are good candidates for MIS 5c records, although both are as yet undated. On the Palos Verdes Hills, what Woodring et al.
(1946) mapped as the "2nd" and "4th" terraces have been dated to ~80 ka and ~120 ka, respectively (Muhs et al., 2006), as
noted earlier. To avoid confusion with terrace numbering that is inconsistent from this area to nearby San Pedro, Muhs et al.
(2006) named these (informally) as the Paseo del Mar (2nd) and Gaffey (4th) terraces. In the western part of the Palos Verdes





Hills (Fig. 26), there is an intermediate-elevation terrace that Woodring et al. (1946) mapped as the "3rd" terrace. Because it occupies a morphostratigraphic position between the ~80 ka (2nd) Paseo del Mar and ~120 (4th) Gaffey terraces, it is very likely that this terrace records the MIS 5c high-sea stand. Woodring et al. (1946) did not report any fossil localities on this terrace, and 30 years of periodic searches by the present author have not resulted in any either, so the terrace remains undated.

The other locality that provides morphostratigraphic evidence of a possible MIS 5c record is Punta Banda, in northern Baja California. At this locality (Fig. 31), Rockwell et al. (1989) reported a small terrace fragment at ~22 m above sea level above the Lighthouse (1st) terrace at ~15 m and the Sea Cave (3rd) terrace at ~34 m. Similar to the Palos Verdes Hills, the Lighthouse and Sea Cave terraces are dated to ~80 ka and ~120 ka, respectively, by both alpha-spectrometric U-series (Rockwell et al., 1989) and TIMS U-series methods (Muhs et al., 2002a). Unfortunately, as with the Palos Verdes Hills, no corals have yet been found on the 2nd, ~22 m terrace on Punta Banda.

While marine terraces dated to MIS 5c are rare on the Pacific Coast of North America, terraces dated to MIS 5a are abundant (Fig. 39 and Table S3). Corals have been acquired and dated by TIMS U-series methods at Coquille Point (Oregon), Point Arena (northern California), three localities between Point Año Nuevo and Santa Cruz (central California), Santa Rosa and San Nicolas Islands, the Palos Verdes Hills, and Point Loma (all in southern California), and Punta Banda (northern Baja California). Analytical and faunal data for these terraces are given in Addicott (1966), Zullo (1969), Kern (1977), Kennedy (1978), Rockwell et al. (1989), and Muhs et al. (2002a, 2006, 2012). In all cases, the faunas are characterized by cool-water forms, with several extralimital northern and northward-ranging species.

In addition to U-series-dated localities, a large number of localities lack corals, but have mollusks that permit aminostratigraphic correlation to MIS 5a, following the approach pioneered by Wehmiller et al. (1977a) and Kennedy et al. (1982). These localities can be found from near Newport, Oregon, into northern California, and to the Channel Islands of southern California (Fig. 39). Like their U-series-dated counterparts, these terraces host faunas with extralimital northern or northward-ranging species of mollusks.

It is very likely that there are terraces dating to MIS 5a and/or MIS 5c in Baja California, Baja California Sur, and Sonora as well, based on amino acid and faunal studies by Emerson et al. (1981) and Ortlieb (1987). In addition, along the Pacific coast of northern Baja California, numerous fossil localities, shown earlier in Figure 30, contain mixes of warm-water and cool-water molluscan faunas. Although none of these terraces have either U-series or aminostratigraphic data for age control (with the exception of Camalú, as noted earlier), their low elevations allow for the possibility that they record some part of MIS 5. The mixes of cool-water and warm-water mollusks invite comparison to similar mixes of faunas with contrasting



thermal aspects, found at Cayucos, Point Loma, and San Nicolas Island, that have U-series ages on corals that include both
MIS 5e and 5c.
Two localities in southern California have U-series and amino acid evidence for emergent marine terraces dating to MIS 3.
Both localities are south of the "big bend" in the San Andreas Fault (Figs. 1, 2) where this large constraining bend brings about
a shift from fault lateral movement to predominantly crustal compression between the Pacific and North America plates. The
result is unusually high rates of uplift, such that terraces formed when sea level was substantially lower than present are now
emergent. Isla Vista, a university community (Fig. 25), is built on a marine terrace whose outer edge is at an elevation of ~7
m. Based on amino acid ratios in *Saxidomus* valves, Wehmiller et al. (1977a), Wehmiller (1982), and Kennedy et al. (1982)
thought that this terrace likely predated MIS 5a. This conclusion was also based on the fact that the fauna within the deposits
of this terrace contains a large number of extralimital northern species (Wright, 1972), consistent with very cold waters off the
California coast at this time, expected for the time period postdating MIS 5a, based on independent evidence (Kennett and
Venz, 1995). Gurrola et al. (2014) reported U-series ages of ~49 ka and ~47 ka for corals from this terrace, which support the
original age interpretations (see Muhs et al., 2014b, for isotopic data for one of these specimens). All these investigators
correlated this terrace with MIS 3.
Also in southern California, there is a terrace that has been correlated to MIS 3 near a locality simply called "Sea Cliff,"
northwest of Ventura (Fig. 25). Along an ~6 km reach of coastline here, there are two marine terraces, a low-elevation surface
dated to the Holocene (see discussion below) and a higher elevation terrace of Pleistocene age above it. The Pleistocene
terrace has a variable elevation in a shore-parallel sense, from just over ~100 m to just over ~200 m above sea level (Wehmiller
et al., 1978). Although no corals have yet been found in the deposits of the Pleistocene terrace, amino acid ratios indicate that
it is likely ~50 ka, similar to the terrace at Isla Vista (Wehmiller et al., 1978; Kennedy et al., 1982; Wehmiller, 1982). The
elevation of this terrace, along with its young age and formation at a time of relatively low sea level indicates that this reach
of coastline has experienced an extremely high rate of uplift.
In northern California, near the Mendocino triple junction of the Gorda, Pacific, and North America plates, there is a third
locality with a marine terrace correlated to MIS 3 (McLaughlin et al., 1983a, b). This terrace is found near Point Delgada (Fig.
21) and has a maximum elevation of ~7 m above sea level. Correlation of this terrace to MIS 3 is based on a radiocarbon age
of ~45 ka from fossil wood found in terrestrial deposits that overlie the marine terrace deposits. Although radiocarbon ages
on wood are usually reliable, this apparent age is near or at the limit of the method and in addition is found within overlying
deposits, not the marine terrace deposits themselves. Thus, the cautions discussed earlier with regard to modern carbon



contamination would apply here as well. Although the age from Point Delgada is interpreted to be a close, minimum-limiting
age, it is in fact just a minimum-limiting age and the terrace itself could be older.
Based on early results of amino acid geochronology in Kennedy et al. (1982), it was originally thought that marine terrace
deposits at a fourth locality, Cape Blanco, Oregon (Figs. 18-20), could correlate with MIS 3. Later amino acid studies, linked
with a nearby U-series-dated, coral-bearing locality (Coquille Point, Oregon), showed that the low terrace at Cape Blanco
likely dates to MIS 5a (Muhs et al., 1990). The fauna at Cape Blanco, with its cool-water species, is similar to that at Coquille
Point (Muhs et al., 2006). Furthermore, oxygen isotope ratios in fossil *Saxidomus gigantea* and *Mya truncata* collected from
the two localities do not have significant differences (Muhs et al., 1990).

### 6.3 Holocene sea level indicators

Emergent Holocene marine deposits are found at several localities along the Pacific Coast of North America. Within the
southern Puget Sound area of Washington State (Fig. 16), emergent marine terraces or peat-covered tidal flats are found at
five localities, as much as ~7 m above sea level, with ages ranging between ~1,000 and 1,100 yr B.P. (Bucknam et al., 1992).
It has long been recognized (e.g., Kelsey, 1990) that there is the potential for coseismic uplift along the zone where the Juan
de Fuca plate is being subducted beneath the North America plate (Fig. 2). What is interesting about the Holocene terraces in
the Puget Sound area, however, is that they are some distance inland from this subduction zone. Bucknam et al. (1992) attribute
Holocene uplift here to reverse slip along an inferred fault within the crust of the North America plate.
Near the Mendocino triple junction area of northern California (Figs. 2, 21), Holocene marine terraces have also been
documented (Lajoie et al., 1991). The most recent of these produced 1.4 m of uplift associated with the $M_S$ 7.1 earthquake at
Cape Mendocino in 1992 (Carver et al., 1994). Merritts (1996) reported that earlier Holocene, coseismic uplift events had
occurred prior to the A.D. 1992 earthquake. Based primarily on radiocarbon ages of marine shells, at least four such events
occurred between ~7 ka and ~0.6 ka.
South of the "big bend" area of the San Andreas Fault (Figs. 2, 25), crustal compression is the dominant tectonic style.
Thus, in the area to the south, uplift rates are very high. Sarna-Wojcicki et al. (1987) mapped two marine terraces in this area,
between Ventura and Santa Barbara. The higher of the two terraces ranges in elevation from ~120 to ~210 m, and amino acid
data in mollusks reported by Wehmiller et al. (1978) indicate that it is likely ~45 ka (MIS 3), as discussed above. The lower
of the two terraces has elevations that range from ~6 to ~35 m. Radiocarbon ages of marine mollusks from this terrace range
from ~5 ka to ~1.8 ka (Sarna-Wojcicki et al., 1987). More recent detailed work by Rockwell et al. (2016) identified four
Holocene terraces in this area, with radiocarbon ages of ~6.7 ka, ~4.4 ka, ~2.1 ka, and ~0.95 ka. Each terrace represents a
separate coseismic uplift event.
Still farther south, the coast of Central America is adjacent to the subduction zone, where the northeast-moving Cocos plate
is being subducted beneath the Caribbean plate (Fig. 35). In addition, the Panama triple junction is situated just south of the
Costa Rica-Panama border, where the Cocos, Caribbean, and Nazca plates intersect. On the Nicoya Peninsula of Costa Rica,



Marshall and Anderson (1995) reported marine terraces at elevations of ~4 to ~16 m above sea level, with radiocarbon ages
of ~4.1 ka to ~0.4 ka. Gardner et al. (2001), working the same general area, reported similar elevations and ages for two
terraces, in agreement with, but adding detail to the study of Marshall and Anderson (1995). Fisher et al. (1998) ascribed uplift
in this region to subduction of seamount chains on the Cocos plate. Marine terraces of Holocene age have also been reported
for the nearby Osa Peninsula of Costa Rica (Gardner et al., 2013) and the Burica Peninsula of Panama (Fig. 35) by Morell et
al. (2011).

**6.4 Implications for paleozoogeography**

There has been considerable interest in MIS 5e not only for its implications for future sea-level rise, but also for warming
of the oceans. Indeed, sea-level rise is linked to ocean warming due to the possibility of thermal expansion of the world's
oceans. In addition, however, possible ocean warming during MIS 5e has importance for understanding how modern marine
ecosystems might respond to future warming.
Global-scale studies of MIS 5e have been carried out using proxy paleoclimate data from deep-sea cores with the goal of
estimating sea surface temperatures (SST). Results of these investigations have not been entirely consistent. CLIMAP Project
Members (1984) concluded that overall, the last interglacial ocean was not significantly different from the modern ocean. It
is important to note, however, that for many regions of the world, including much of the ocean around Australia, the
Mediterranean Basin, the Bering Sea, the central Pacific Ocean, and the eastern Pacific Ocean off North America, there were
few cores available. Using a larger dataset, Turney and Jones (2010) concluded that MIS 5e global temperatures were on
average ~1.5°C warmer than present, although part of this conclusion is based on ice and terrestrial records. In yet another
compilation, McKay et al. (2011) concluded that on a global scale, SST during MIS 5e was not significantly different from
the present. From this, these investigators inferred that thermal expansion likely played only a minor role, if any, in the higher
than present sea level during MIS 5e. However, as the same investigators also pointed out, some regions are exceptions to this
generalization.
In some regions where core data are sparse, shallow invertebrate marine fossil faunas serve as an important record of SST.
For example, extralimital species of mollusks and corals, indicating warmer SST during MIS 5e, have been documented in the
Indian Ocean along the western coast of Australia (Kendrick et al., 1991), around New Zealand and the southern coast of
Australia (Murray-Wallace et al., 2000), along the Bering Sea and Arctic Ocean coasts of Alaska (Brigham-Grette and
Hopkins, 1995), in the eastern Atlantic Ocean off Africa and the Mediterranean Basin (Cuerda, 1975, 1987, 1989; Cuerda and
Sacarès, 1992; Hearty et al., 1986; Meco et al., 2002, 2006; Muhs et al., 2014c), in the western Atlantic Ocean around Bermuda
(Richards et al., 1969; Muhs et al., 2002b), and in the Pacific Ocean along the shores of Oahu in the Hawaiian Islands (Kosuge,
1969; Muhs et al., 2002b; Groves, 2011). The sites studied in Australia, the central Pacific, the western Atlantic Ocean, the
eastern Atlantic Ocean off Africa, the Mediterranean, and the central Pacific are all localities anchored by reliable MIS 5e U-
series ages on corals.
In the context of both deep-sea core proxy climate data and shallow-water marine invertebrate records from around the
globe, it is interesting to consider what the SST off the Pacific Coast of North America was during MIS 5e. Herbert et al.





(1998) showed that alkenone unsaturation indices, derived from modern core-top samples, correlate in a linear fashion with
modern SST. Using this relation, Herbert et al. (2001) generated both oxygen isotope values in foraminifera (to identify MIS
5e) and alkenone unsaturation indices to estimate SST in five cores, taken off northern California to south of Cabo San Lucas,
Baja California Sur. In all cases, SST during MIS 5e is substantially higher than at present or during earlier parts of the
Holocene. One of the cores examined (Ocean Drilling Project, or ODP 893) is from Santa Barbara Basin, the same locality
studied for temperature-sensitive foraminiferal species by Kennett and Venz (1995). The latter workers found that MIS 5e
was the only time, other than the Holocene, when warm-water foraminifera were present in Santa Barbara Basin, in good
agreement with the alkenone unsaturation index data. Two other cores, one off central California and one off southern
California, studied by Yamamoto et al. (2007), also gave alkenone-based SST indicating substantially warmer waters off the
Pacific Coast during MIS 5e compared to present.
Given these findings, it is pertinent to evaluate the shallow-water marine terrace records of mollusks and other invertebrates
of MIS 5e age from the Pacific Coast of North America. As noted earlier, pioneering amino acid studies by Wehmiller et al.
(1977a) and Kennedy et al. (1982) considered that marine terrace deposits correlated to MIS 5a (~80 ka) had cool-water faunas
and those correlated to MIS 5e (~120 ka) had warm-water faunas. While the cool-water forms that are so prominent in terraces
correlated to MIS 5a by amino acid geochronology (Kennedy et al., 1982) have been largely confirmed to indeed be ~80 ka,
based on TIMS U-series ages of corals (Muhs et al., 2002a, 2006), the GIA-related fossil mixing of ~120 ka and ~100 ka (MIS
5c) fossils into single terrace deposits complicates matters. Nevertheless, using those localities where there is good evidence
for MIS 5e age fossils and assuming that the cool-water forms represent MIS 5c, the shallow-water marine invertebrate record
still allows some inferences about SST during the peak of the last interglacial period. Here, examples of bivalves, gastropods
and corals are examined from dated deposits to illustrate what can be inferred about ocean temperatures during MIS 5e.
Two species of bivalves that live dominantly in tropical waters off the Pacific Coast of North America are *Chione undatella*
and *Dosinia ponderosa*. *C. undatella* is one of two species of *Chione* (*C. californiensis* is the other) that presently live only
south of the Point Conception area (Fig. 40 a). Although *C. undatella* is found only as far north as Goleta (near Santa Barbara),
California, it ranges south along the coast of Mexico, including the Golfo de California, to Peru, and is also found on the
Galapagos Islands (Coan and Valentich-Scott, 2012). In fossil form, *C. undatella* is found in several marine terrace deposits
either directly dated to MIS 5e or correlated to it on the basis of amino acids, from Bahía Magdalena, Baja California Sur,
north to Potrero Canyon near Los Angeles, California (Fig. 40a). In addition, however, it has also been reported from a terrace
correlated to MIS 5e from near San Luis Obispo Bay by Kennedy (2000) and also has been found at Tomales Bay, California.
The presence of *C. undatella* at Tomales Bay is particularly significant, because this locality is ~500 km northwest of its
modern northern limit. *Dosinia ponderosa* at present ranges only as far north as Laguna Ojo de Liebre, on the Pacific coast
of Baja California Sur, just east of Punta Eugenia (Fig. 40b). Like *C. undatella*, *D. ponderosa* ranges south along the coast of
Baja California Sur, including the Golfo de California, all the way to Peru and including the Galapagos Islands (Coan and
Valentich-Scott, 2012). In fossil form, it is found at several localities dated or correlated to MIS 5e in Baja California Sur and



Sonora, all within its present range, but also as far north as Potrero Canyon near Los Angeles, California, and at Newport Bay (Fig. 40b). These California localities are ~750 km northwest of the modern northern limit of *D. ponderosa*.

The gastropod fossil records from MIS 5e terrace deposits also show that what are now southern species lived farther north during the last interglacial period. *Mexacanthina lugubris* is a gastropod only rarely found as far north as San Diego. Bertsch and Aguilar Rosas (2016) report that on the Pacific coast, *M. lugubris* presently lives from San Diego to Cabo San Lucas and on the eastern Golfo de California coast, the species is found from Bahía Kino, Sonora to Mazatlán, Sinaloa. Fossil occurrences of *Mexacanthina lugubris* in deposits dated to ~120 ka are found all along this taxon's modern Pacific Coast distribution, from Bahía de Magdalena, Baja California Sur, to Point Loma, near San Diego. However, there are also some occurrences reported in MIS 5e deposits, well north of the modern range endpoint for *Mexacanthina lugubris* (Fig. 41a). Although warmer waters during the last interglacial period allowed *Chione undatella* to migrate north of its modern range by several hundred kilometers, Point Conception was apparently a barrier to northward migration of *Mexacanthina lugubris* beyond the Santa Barbara region. Another gastropod, *Stramonita biserialis*, presently lives from Cedros Island, just north of Punta Eugenia, south along Baja California Sur, throughout the Golfo de California, and all the way to Chile, as well as being on the Galapagos Islands (Keen, 1971). In MIS 5e deposits, it is found at localities in the Golfo de California and along the coast of Baja California Sur, all within its modern range (Fig. 41b). However, it is also found in MIS 5e deposits on Isla Guadalupe, along the northwestern coast of Baja California, and in some southern California localities as far north as San Pedro, California, near Los Angeles (Fig. 41b). The occurrence of *S. biserialis* in San Pedro is a northward extension of its modern range by nearly 700 km.

One particularly interesting locality, with U-series ages of ~120 ka on corals, is the low-elevation marine deposit on Isla Guadalupe, off the Pacific coast of Baja California (Figs. 42, 43). Isla Guadalupe is interesting zoogeographically even at present, because its modern marine invertebrate fauna has elements of both the Californian and Panamanian faunal provinces (Lindberg et al., 1980), making it a particularly sensitive area. The MIS 5e marine deposits here have a fauna that has been reported by Lindberg et al. (1980) and Durham (1980). Muhs et al. (2002a) summarized the modern geographic ranges of this fauna, showing that it has a substantial number of extralimital southern species of taxa, along with what was then thought to be two northward-ranging species. With new data on ranges of species that have been published since that time (Coan and Valentich-Scott, 2012; Berschauer and Clark, 2018), that paleozoogeographic analysis has been redone here. Results indicate that the fauna contains no northward-ranging species, but hosts 13 extralimital southern species (Fig. 44). All but three of these taxa have southern range endpoints south of the equator, and six species have northern range endpoints no farther north than Bahía Magdalena, which is over 700 km southeast of Isla Guadalupe.

In addition to the extralimital southern species of bivalves and mollusks within the MIS 5e fauna of Isla Guadalupe, there are two other taxa which merit additional discussion for the paleozoogeographic significance. The MIS 5e deposits of Isla Guadalupe host the North American Pacific Coast's northernmost MIS 5e occurrence of a hermtypic colonial coral, *Pocillopora guadalupensis* (Durham, 1980). This species is not known to be living in the eastern Pacific at present (Reyes-Bonilla and López-Pérez, 1998), nor has it been found in other Pleistocene or Pliocene marine deposits in the region (López Pérez, 2008). The eastern Pacific region at present hosts five species of *Pocillopora*, with one species (*P. verrucosa*) found as far north as





Isla San Marcos, on the eastern coast of Baja California Sur, and four species currently living offshore near Cabo San Lucas
(Reyes-Bonilla and López-Pérez, 1998). However, Isla Guadalupe is ~1000 km northwest of Cabo San Lucas (Fig. 42).
Durham (1980) pointed out that *P. guadalupensis* more closely resembles *Pocillopora* species of the central and western
Pacific Ocean (such as *P. ligulata*) than it does to species of this genus found in the southeastern Pacific, a relationship he
described as "strange." However, it is now known that living *P. ligulata* is found not only in the central Pacific Ocean
(including the Hawaiian Islands) but is also found off the coasts of Colombia and Ecuador (Glynn et al., 2017). Thus, one
could hypothesize that perhaps there is a last interglacial evolutionary link between *P. ligulata* from tropical waters of
northwestern South America and *P. guadalupensis* of Isla Guadalupe.
The other particularly noteworthy fossil reported by Lindberg et al. (1980) from the MIS 5e fauna of Isla Guadalupe is the
cowry *Cypraea* (*Erosaria*) *cernica*, now called *Naria cernica*. This species is not known to occur anywhere along the Pacific
coasts of the Americas, and currently lives in the tropical waters of the Indo-West Pacific province (Burgess, 1985). The
closest living populations of this species to Isla Guadalupe are in the Hawaiian Islands (Severns, 2011), although interestingly,
the species has not yet been reported as a fossil within MIS 5e (or older) deposits on the Hawaiian Islands (Groves, 2011). In
any interpretation, however, Isla Guadalupe is thousands of kilometers away from any present location where *Naria cernica*
can be found, making its presence on this island during MIS 5e a remarkable find.
Collectively, the MIS 5e fossil record for bivalves (*Chione undatella*, *Dosinia ponderosa*), gastropods (*Mexacanthina*
*lugubris*, *Stramonita biserialis*, *Naria cernica*), and coral (*Pocillopora guadalupensis*) from several localities, from Baja
California to northern California, indicates that water temperatures off the Pacific Coast of North America were substantially
warmer than present. It does not appear that there were wholesale shifts of entire faunal provinces, such as the present-day
Californian province being replaced entirely by Panamic species (Fig. 43). Lindberg et al. (1980) point out that of the modern
fauna of Isla Guadalupe, ~75% are from the Californian province, ~6% are Panamic, and ~19% are biprovincial. In contrast,
the MIS 5e fauna of Isla Guadalupe consists of ~39% Californian species, 32% Panamic species, and ~29% biprovincial
species. Thus, while it is clear that greater numbers of warm-water species lived in more northerly locations than is the case
today, certain species existed within each faunal province as they do today. Furthermore, it appears that some of the physical
geographic barriers that define provincial boundaries also served as barriers during MIS 5e, despite northward migrations. A
good example of this is the migration of *Mexacanthina lugubris* north of where its present northern limit is situated, but
apparently Point Conception prevented this taxon from migration farther north, into what is now the Oregonian province (Fig.
43). Still, while it may be difficult to quantify the degree of ocean warming off the Pacific Coast during MIS 5e, the presence
of numerous extralimital species in terrace deposits at many localities is consistent with the alkenone and foraminiferal data
from deep sea cores that eastern Pacific Ocean SST were higher than present, from northern California to southern Baja
California.
**6.5 Controversies**



In some of the earliest studies of late Quaternary sea level history, supported by what was then the relatively new U-series
dating method, there was general agreement that sea level stands during MIS 5c (~100 ka) and MIS 5a (~80 ka) were
substantially below modern sea level, by as much as 10 to 20 m (Broecker et al., 1968; Mesolella et al., 1969; Veeh and
Chappell, 1970; Bloom et al., 1974; Chappell, 1974).  These early studies were on Barbados and New Guinea, two areas far
apart from one another, unrelated tectonically, and having quite different long-term uplift rates.  The broad agreement in
paleo-sea level estimates for MIS 5c and 5a at both localities seemed to provide support for a "global" eustatic sea-level
history for the late Quaternary.  Later, a third locality with emergent reef terraces, the northwest coast of Haiti, showed
general agreement with Barbados and New Guinea for paleo-sea levels at MIS 5c and 5a (Dodge et al., 1983), which
reinforced the concept of a global eustatic sea-level curve.  For New Guinea, the original paleo-sea level estimates were
refined by Chappell and Shackleton (1986).
Because marine terraces on many coastlines lack materials suitable for dating, a number of graphical methods emerged in
an attempt to compare the elevation spacing of a suite of undated marine terraces with a "global" sea level curve.  Most of
these schemes assumed that the detailed paleo-sea level record of the Huon Peninsula of New Guinea (Chappell and
Shackleton, 1986) is a faithful representation of global, eustatic sea level change.  Lajoie (1986) even ventured the opinion
that dating a suite of marine terraces was simply a matter of correlating the undated landforms with the appropriate peaks on
the New Guinea sea level curve.  Similarly, Bull (1985) proposed that dating of an entire suite of otherwise undated marine
terraces could be accomplished solely by graphical means.  In Bull's (1985) method, a given terrace was assigned an age and
paleo-sea level corresponding to a possible correlative terrace on New Guinea.  The resultant uplift rate, along with the New
Guinea sea level curve, was used to plot inferred amounts of uplift for other terraces.  The process was repeated for different
assumed ages of the original terrace chosen and different uplift rates.  Whichever of the resultant plots yielded the best-fit
linear array of points on an inferred uplift vs. age plot was interpreted to be the correct correlation and uplift rate.  In this
method, once the "correct" uplift rate was identified, all terraces in the suite were dated simultaneously.  The technique and
variations of it have been applied to undated or partially dated terrace sequences in New Zealand (Bull and Cooper, 1986),
northern California (Merritts and Bull, 1989; McCrory, 2000), central California (Hanson et al., 1994), southern California
(Trecker et al., 1998), Mexico (Mayer and Vincent, 1999), and Italy (Calanchi et al., 2002).
Despite the apparent agreement for a global eustatic sea level curve, there were always localities with marine terrace
elevations that did not seem to fit the Barbados-New Guinea sea level curve for the late Quaternary.  On the Atlantic Coastal
Plain of the USA, emergent marine deposits, a few meters above sea level, gave U-series ages on coral of ~80 ka (Cronin et
al., 1981), unexpected on a passive continental margin, given the sea level estimates at this time from Barbados and New
Guinea.  Similar results were obtained on tectonically stable Bermuda, where the marine facies of the Southampton
Formation, at 1-2 m above sea level, yielded U-series ages on coral averaging ~80 ka (Harmon et al., 1983).  Later studies on
both Bermuda and the Atlantic Coast Plain, with more elevation measurements and precise TIMS U-series dating, gave the
same results as these early studies (Muhs et al., 2002b; Wehmiller et al., 2004).  On the tectonically active Ryukyu Islands of



Japan, where reef terraces dating to ~120 ka, ~100 ka, and ~80 ka are all present, elevations yield paleo-sea levels at MIS 5c
and 5a that are close to present (Ota and Omura, 1992). U-series ages and terrace elevations from the Pacific Coast of North
America (California and Mexico) also give paleo-sea level estimates for MIS 5c and 5a that are much closer to present sea
level than what would be expected from the Barbados-New Guinea records (Muhs et al., 1994). Thus, despite the
attractiveness of being able to date, using graphical techniques, an entire suite of marine terraces with no independent age
control, it is in fact a hazardous practice.

The explanation for the disagreement between some paleo-sea level estimates and the Barbados-New Guinea sea-level

history is likely due to glacial isostatic adjustment (GIA) processes. Indeed, GIA effects can and should be expected in high-
latitude and mid-latitude regions of the Northern Hemisphere where large, continental ice sheets were found during glacial
periods. Thus, using the Barbados-New Guinea sea level curve for terrace correlations via elevation spacing in such regions
will likely yield spurious results. In some far-field regions, distant from the Laurentide, Cordilleran and Fennoscandian ice
sheets, the elevation-spacing method of terrace correlation might be applicable, but in virtually all mid-latitude and high-
latitude regions, the approach is untenable. Creveling et al. (2015) modeled apparent sea levels around the world, assuming
a true "eustatic" high-sea level of +6 m. These investigators showed that relative sea level, at the end of MIS 5e, could have
varied from ~5.3-5.7 m above present (in far-field regions such as Australia and South Africa) to as much as ~9-11 m above
present (on coastlines and islands of North America or around it). Further efforts along these lines by Dendy et al. (2017)
confirm these differences and provide additional insights on how ice sheet configuration during the penultimate glaciation
(MIS 6) influenced sea levels during MIS 5e. A combined field and modeling study on San Nicolas Island, California
showed that simulations of GIA processes over the period since MIS 5e yielded a sea-level history that matched the
elevation spacing of marine terraces dating to MIS 5c and 5a (Muhs et al., 2012). Later modeling by Creveling et al. (2017)
refined paleo-sea level estimates for both of these time periods and extended the concept of differing sea level histories to
much of the globe. Simms et al. (2016, 2020) conducted GIA modeling specifically on the Pacific Coast of North America
and also confirmed MIS 5c and 5a paleo-sea levels higher than what would be predicted by the Barbados-New Guinea
terrace records.

GIA processes may also help explain what in the past had been an enigmatic observation about MIS 5e marine terrace

faunas and controversy about their origins. It was noted above that there is no persuasive field evidence of more than one
high-sea stand during MIS 5e on the Pacific Coast of North America. What *has* been documented, however, is evidence that
in areas of low uplift rate, marine terraces that formed during MIS 5e were reoccupied by the high-sea stand that followed it,
MIS 5c (~100 ka). The evidence of this sequence of events has actually been in existence for more than a century, with the
recognition of "thermally anomalous" faunas, i.e., those fossil faunas with *both* extralimital northern and extralimital southern
species of mollusks within the same deposit. Many hypotheses have been proposed to account for this (see review in Muhs
and Groves, 2018), but TIMS U-series dating finally demonstrated that corals of both MIS 5e and MIS 5c age exist in the same
marine terrace deposits at localities in central and southern California (see data in Table S1). At Cayucos (central California)





and Point Loma (southern California), both ~120 ka and ~100 ka corals exist within the same terrace deposits and terrace
deposits at both localities also host a mix of warm-water (~120 ka?) and cool-water (~100 ka?) mollusks (Muhs et al., 2002a).
On San Nicolas Island, the same mix of ~120 ka and ~100 ka corals and warm and cool mollusks is present in what Muhs et
al. (2012) called terrace 2b. Remnants of a slightly higher elevation terrace ("2a") have only ~120 ka corals and no cool-water
mollusks. Where corals are lacking from other Channel Islands marine terrace deposits, amino acid data show the likelihood
of two ages of shells, along with a mix of warm-water and cool-water mollusks (see Muhs and Groves, 2018, for examples).
Muhs et al. (2012) showed that the likely explanation for these observations is a low uplift rate combined with GIA processes
that resulted in a higher sea level during MIS 5c (Fig. 45).

Despite the general agreement between GIA models and field evidence from the Pacific Coast, there is an unresolved issue.

As noted, GIA modeling for MIS 5a and MIS 5c conducted by Muhs et al. (2012), Creveling et al. (2017) and Simms et al.
(2016) fit the elevation differences seen in terraces of these ages in California. A problem that remains, however, is the relative
elevation of the MIS 5e sea level on the Pacific Coast. Simms et al. (2016) modeled relative sea level at ~119 ka, including a
correction for the eustatic component of sea-level rise (taken to be 6 ± 3 m, relative to present), from Washington State to
southernmost Baja California Sur. Their results indicate a paleo-sea level as high as +13 m relative to present along much of
this coast from Washington to the southern Channel Islands, decreasing to +12 m in northern Baja California, and ultimately
decreasing to +10 m in southernmost Baja California Sur. Along with their modeled paleo-sea levels for MIS 5a and 5c, there
is a good match to the elevational spacing of a number of terrace sequences along the coast. In addition, there is good
agreement with the Simms et al. (2016) modeling for MIS 5e and similar modeling done for selected sites on the Pacific Coast
by Creveling et al. (2015). Despite these promising results, Muhs et al. (2021) pointed out that there are several localities,
from central California to southernmost Baja California Sur, where elevations of MIS 5e terraces do not agree with the GIA
model results. These sites include Cayucos, Point San Luis, Santa Cruz Island, Anacapa Island, and Santa Barbara Island,
California, as well as Isla Guadalupe, Baja California. Older, higher elevation terraces at many of these localities preclude an
explanation of subsidence, as these higher terraces indicate a trend of steady, long-term uplift in the Quaternary. The reason
for the differences between the field data and these well conceptualized GIA models is not understood and needs more study.

It is interesting to note that GIA processes and their effect on relative sea levels may not be limited to MIS 5 paleo-sea

levels. Returning to San Nicolas Island, it was noted that there are 14 terraces on this island (Fig. 37). Vedder and Norris
(1963) reported faunal data from most of these terraces. Deposits of the 5th, 8th, and 10th terraces all contain mixes of both
warm-water and cool-water species of mollusks, suggesting a similar sequence of events as that described above for MIS 5e
and MIS 5c. Higher, older terraces elsewhere in California have not yet been investigated for this same kind of record but
would be a worthwhile effort.

A controversy that exists for MIS 5e along the Pacific Coast of North America is the amount of sea surface warming during

the last interglacial period derived from the fossil record compared to that from climate modeling. As noted earlier, faunal
evidence from a variety of marine terrace localities, from southern Baja California Sur to north of San Francisco Bay, indicates
substantial warming during MIS 5e, relative to present. Northward migration of what are now subtropical or tropical species





into mid-latitudes is documented at several localities (Figs. 13, 40, 41, 42, 44). These observations from the marine terrace
record are mirrored in the foraminiferal and alkenone records found in deep-sea cores off the Pacific Coast of North America
(Kennett and Venz, 1995; Herbert et al., 2001; Yamamoto et al., 2007), as summarized earlier. Modeling of SST (as well as
land surface temperatures) by Otto-Bliesner et al. (2013) indicates, however, that there was little annual surface temperature
change during MIS 5e, compared to pre-industrial modern time, on a global basis. In addition, for the Pacific Coast of North
America specifically, their model results mirror that of the global simulation, i.e., very little difference in MIS 5e time
compared to pre-industrial modern time. Otto-Bliesner et al. (2013) noted that their modeling was not able to reproduce many
proxy paleoclimate records that indicate greater warmth during MIS 5e. The reason for the disagreement between the model
results and the marine terrace faunal records (as well as those from deep-sea cores) is not understood at present and needs
more investigation.
**7 Future research directions**
In examining the work done to date on the Pacific Coast of North America, several topics that could merit additional work
have been mentioned. However, some specific needs that would be particularly useful are described here. Terrace mapping
can certainly be improved in many areas that have not received much attention. Particularly needed are good maps of marine
terraces in certain parts of northern California, central California, Baja California, and Sonora. More U-series ages on corals
are needed, particularly in Oregon, northern California, and northern Baja California. Preliminary field studies that I have
conducted indicate that the solitary coral *Balanophyllia elegans* is present in marine terrace deposits in some parts of northern
California where geochronology has yet to be conducted. Similarly, around the Golfo de California, colonial corals are likely
present in many marine terrace deposits that have not yet been studied, both on the Baja California peninsula, and on the coast
of mainland Mexico. More characterization of fossil faunas is needed. Very little work has yet been done on the coast of the
Golfo de California but could be carried out in concert with new U-series dating of colonial corals. Continued refinement of
GIA models and testing of those models is needed, particularly in view of the growing appreciation that paleo-sea levels around
the world during MIS 5e, 5c, and 5a are going to differ from coast to coast. Higher relative sea levels during MIS 5c and 5a,
with faunal mixing, is confirmed at only three localities at present in coastal California, but should be investigated at other
localities, particularly to see if there is a GIA gradient, which some models suggest should have existed (Simms et al., 2016).
Examination of fossil faunas and dating, perhaps with Sr isotopes, of pre-MIS 5e interglacials is an endeavor that could be
very usefully pursued. Records of older interglacials are found on the Palos Verdes Hills (Woodring et al., 1946), in the
Newport Bay area (Vedder et al., 1957, and on San Nicolas Island (Vedder and Norris, 1963). Such studies could test the
degree to which GIA effects were active during the middle and early Quaternary. Development of new dating methods for
marine terrace deposits that lack corals or even mollusks is encouraged. Cosmogenic and luminescence methods have promise,
but need to be investigated more thoroughly, particularly in those areas where there is independent geochronologic control
using U-series methods on corals. Finally, it would be useful to continue exploration of why climate modelling has shown
very little evidence for warming of the eastern Pacific Ocean off North America during MIS 5e, whereas the fossil records



(terrace faunas, foraminifera, alkenones) all point to substantial warming during this period. Because of the expectation of
future warming of the eastern Pacific Ocean, better agreement between models and geologic records is a worthy goal.
**8 Data availability**
Data from this study are open access and available at the following link: https://doi.org/10.5281/zenodo.5557355. Data
were exported from the WALIS database on 14 April 2021 and database descriptions can be found at the following link:
https://doi.org/10.5281/zenodo.3961543 [Rovere et al., 2020]. Further information about the database can be examined here:
https://warmcoasts.eu/world-atlas.html (last access: 14 April 2021).
**9 Author contribution**
The manuscript was written and all figures drawn by D.R. Muhs
**10 Competing interests**
There are no competing interests of which the author is aware.
**11 Acknowledgments**
I thank the Climate Research and Development Program of the U.S. Geological Survey for supporting my research on sea
level history through the "Geologic Records of High Sea Levels" project. Much appreciation goes to Alessio Rovere, who
kindly invited me to contribute this review and provided an excellent review and much editorial guidance. Deidre Ryan ably
assisted with the details of compiling amino acid data. A big thank-you goes to John Wehmiller (University of Delaware),
who helped me sort through amino acid data from nearly 40 years ago. Many thanks go to Markes Johnson (Williams College)
and Lauren Toth (U.S. Geological Survey) for contributing beautiful photos of fossil and modern corals, respectively. Laura
Brothers and Janet Slate (U.S. Geological Survey) and Jessica (JC) Creveling (Oregon State University) read an earlier version
of the paper and made many helpful comments for its improvement, which I appreciate very much.

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



**Figure 1: Tectonic setting of North America showing lithospheric plates, plate boundaries, and features referred to in the text. Redrawn in simplified form from Simkin et al. (2006). PTJ, Panaman Triple Junction.**

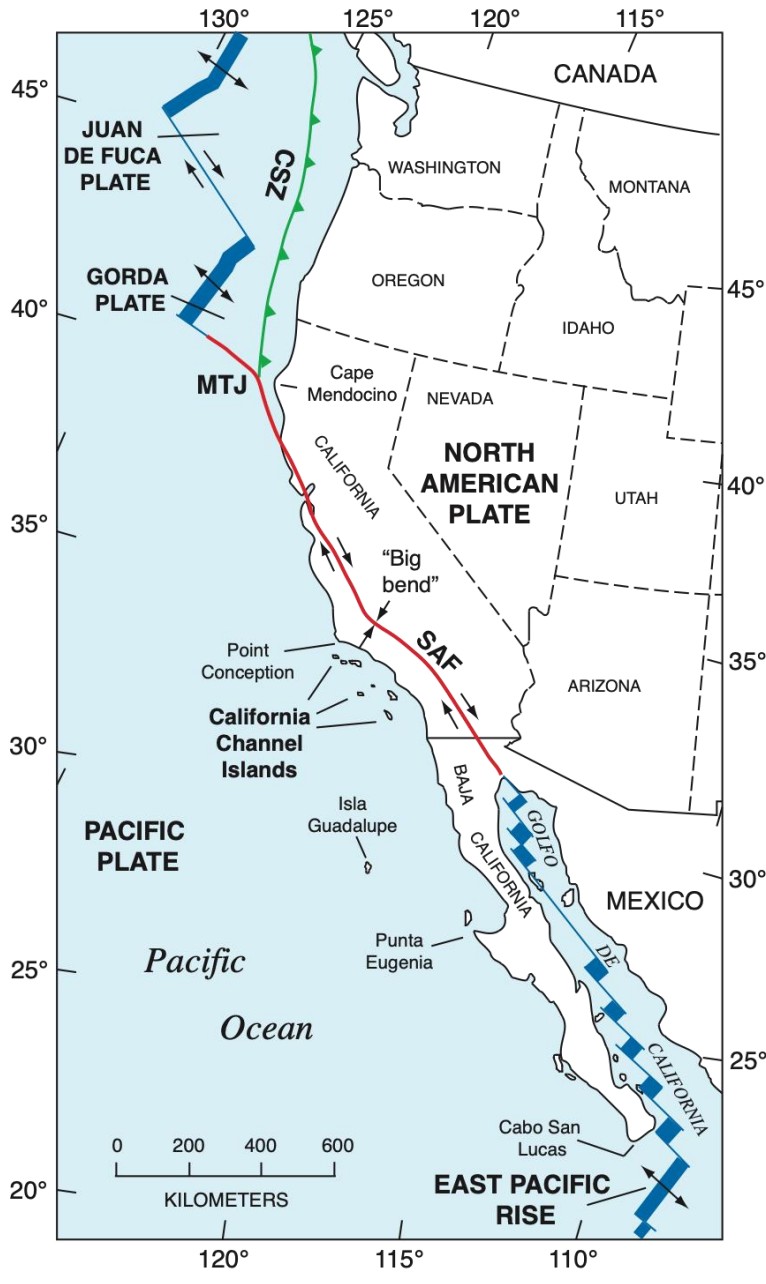

Figure 2: Tectonic setting of the Pacific Coast of North America, from southern Canada to southern Baja California Sur, Mexico, showing plates, plate boundaries, structures, and localities referred to in the text. Redrawn in simplified form from Simkin et al. (2006). CSZ, Cascadia Subduction Zone; MTJ, Mendocino Triple Junction; SAF, San Andreas Fault.



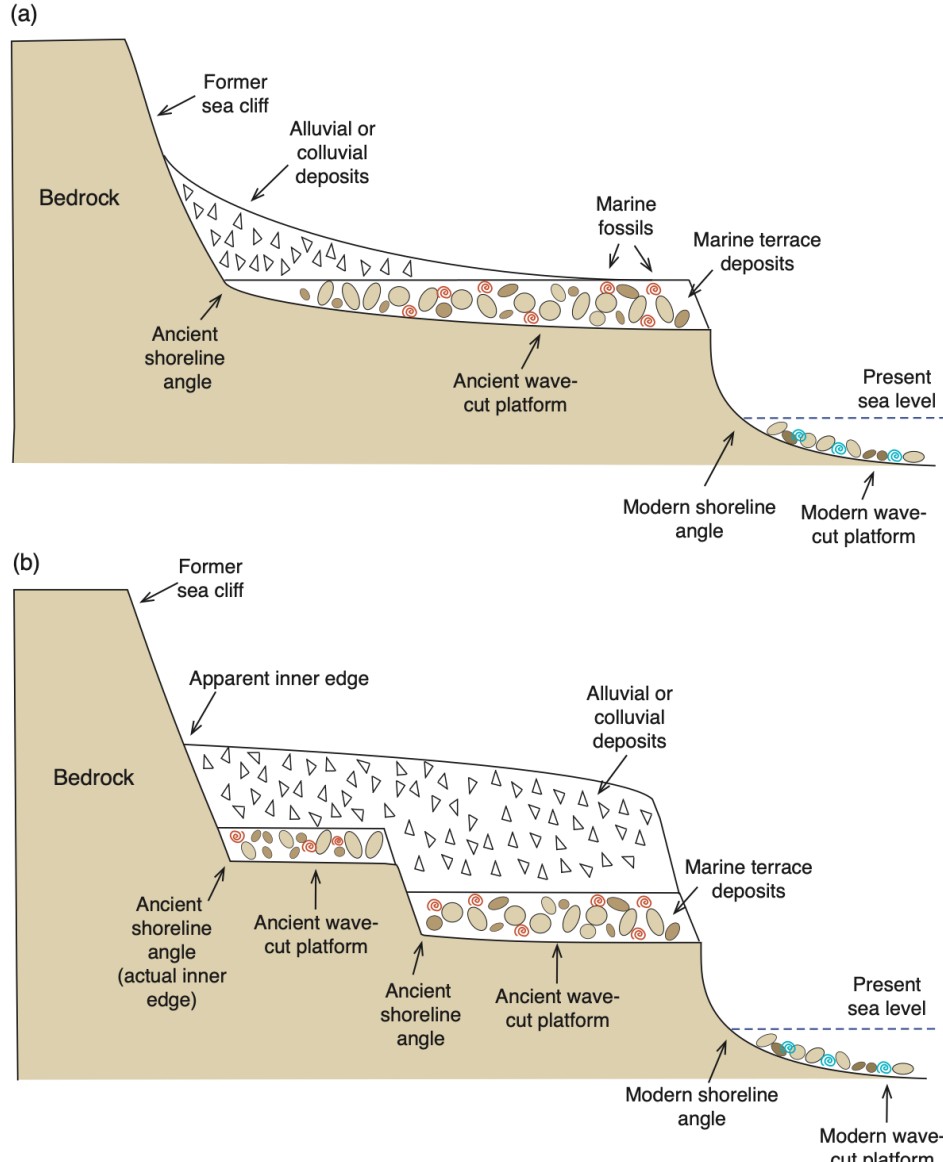

Figure 3. Diagrams showing the terminology used for marine terraces: (a) simple case of a modern wave-cut bench in the surf zone with marine gravels and modern shells (blue symbols), shoreline angle, and single emergent marine terrace above it, with a colluvial cover masking most of the marine terrace deposits with their fossils (red symbols); (b) more complex case with the features described above, but an additional terrace above the lower one. Note that in (b), colluvial deposits cover both of the emergent terraces, making them appear as one landform, with a single inner edge that is at a higher elevation than the shoreline angles of both emergent terraces.





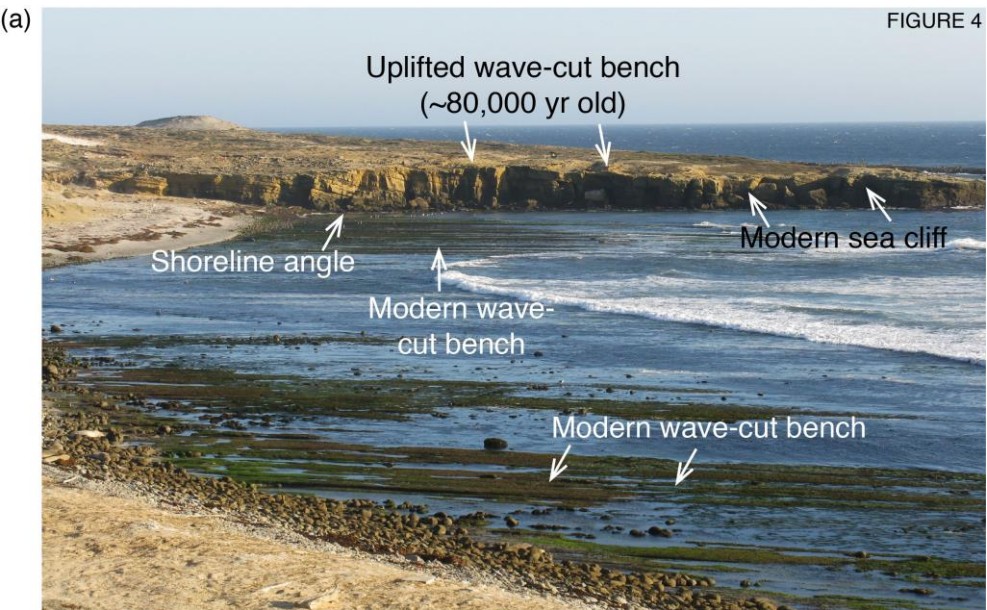

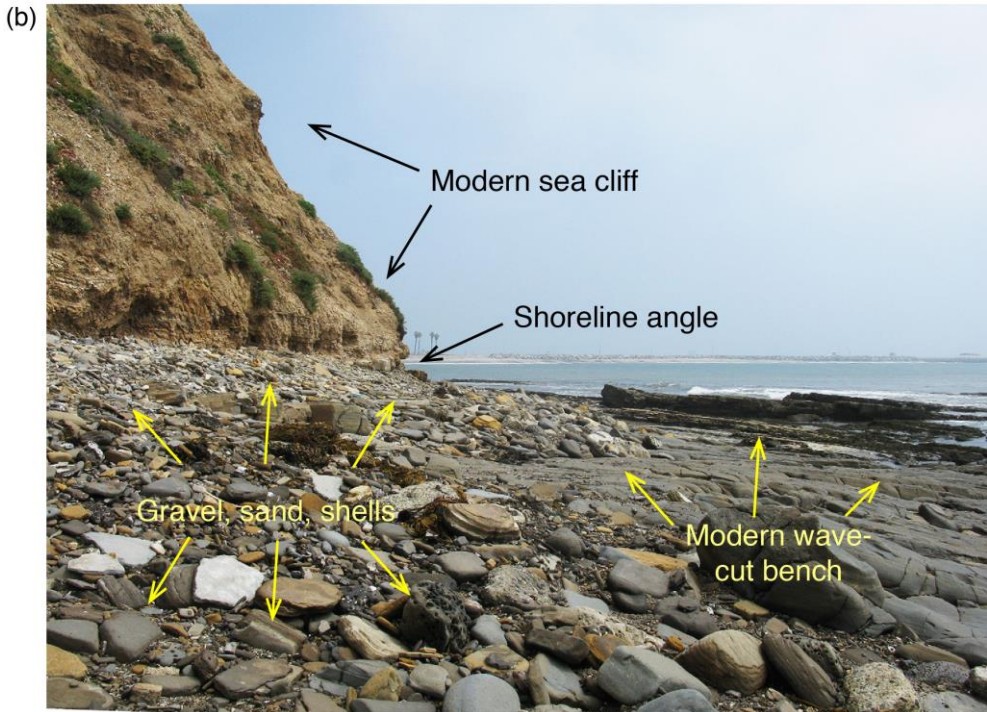

**Figure 4: (a) Modern wave-cut bench exposed at low tide, shoreline angle, sea cliff, and wave-cut bench of emergent, ~80 ka marine terrace, Cormorant Rock, San Nicolas Island, California, USA. (b) Modern wave-cut bench**





**exposed at low tide, overlying marine gravels, shoreline angle, and sea cliff, San Pedro, California, USA.**

**Photographs by D.R. Muhs.**



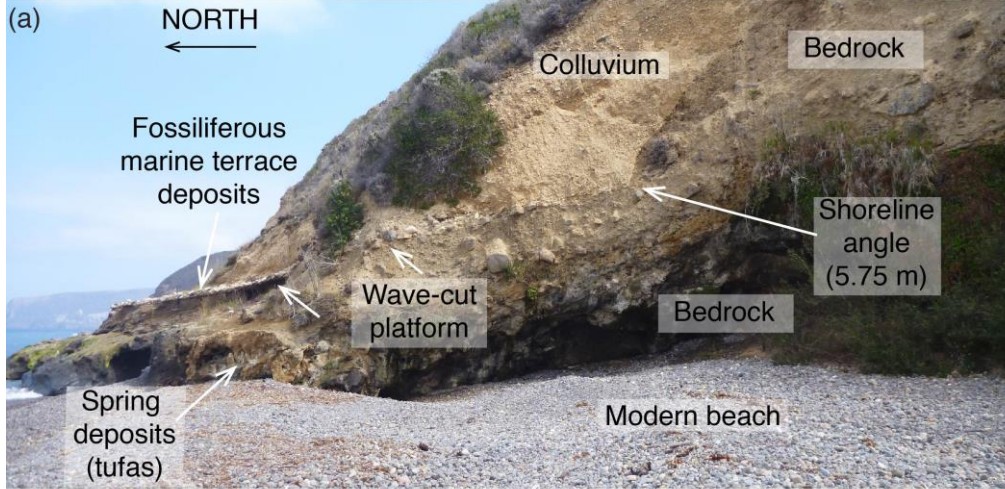

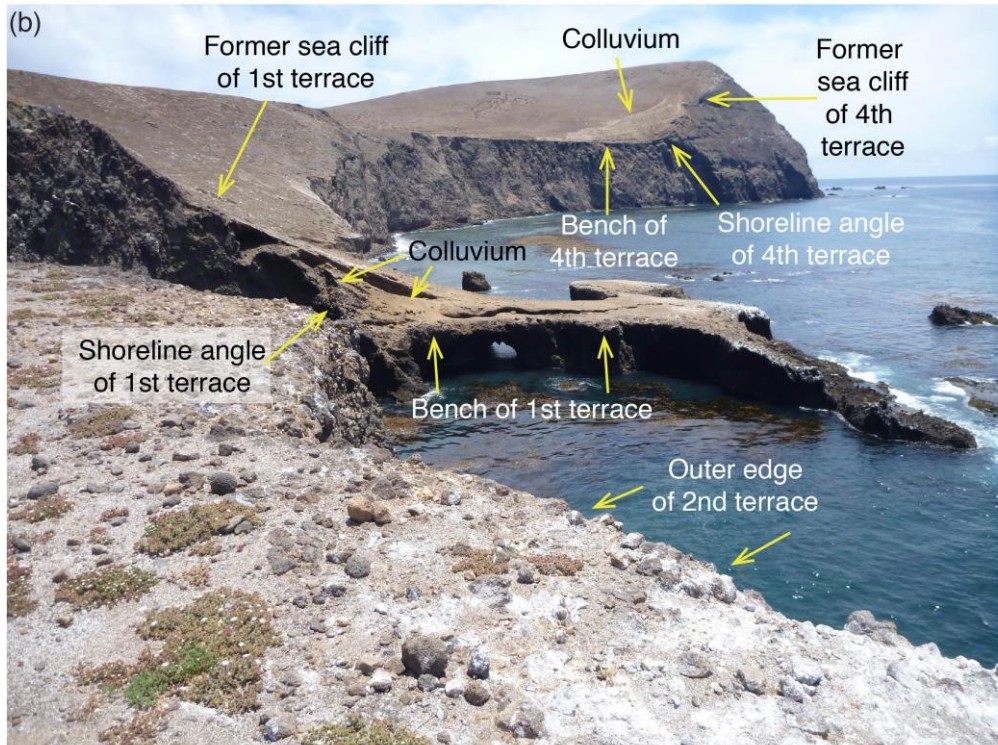

**Figure 5: Examples of exposures of ancient shoreline angles: (a) north coast of Santa Cruz Island, California, just east of Prisoners Harbor; (b) west side of Santa Barbara Island, California, showing benches and shoreline angles of three of the four lowest marine terraces. Photographs by D.R. Muhs.**



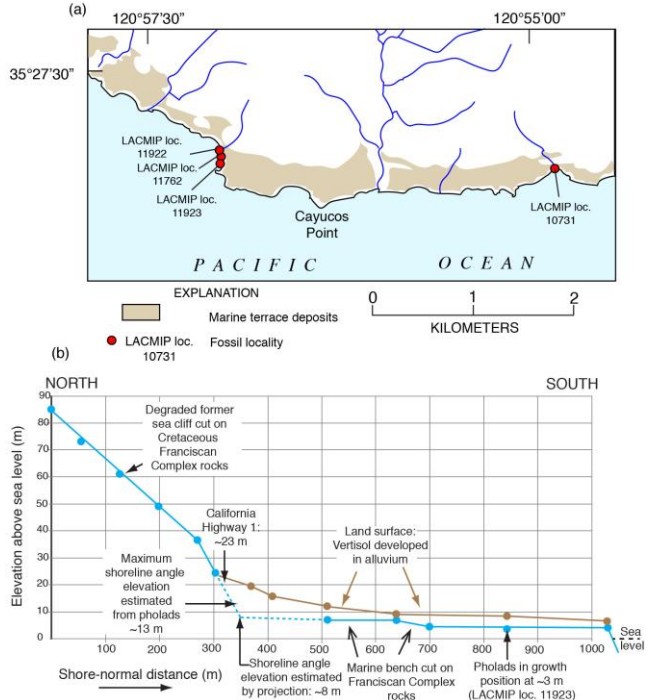

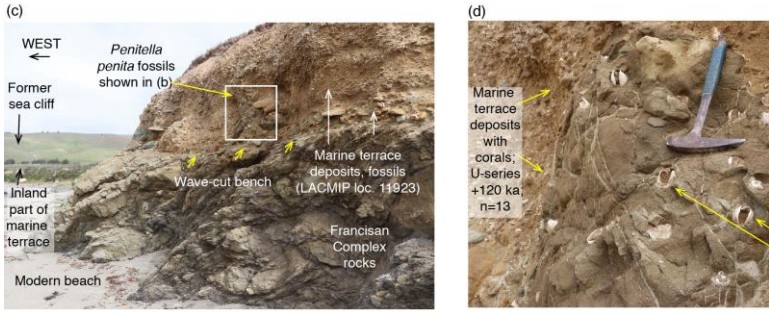

**Figure 6: Two methods of estimating paleo-sea level when shoreline angles are not exposed, Cayucos, California. (a) Map showing extent of marine terrace dated to ~120 ka and fossil localities from Muhs et al. (2002a). (b) Topographic profile of a shore-normal transect in the vicinity of fossil localities LACMIP. 11923, 11762, and 11922, showing measured bench elevations and paleo-sea cliff elevations, where they are exposed; intersection of extrapolated wave-cut bench slope landward and paleo-sea cliff slope downward yields an estimated shoreline angle elevation of ~8 m. (c) Photograph of outer edge of terrace at fossil locality LACMIP 11923, showing wave-cut bench 3 m above sea level with *Penitella penita* fossils (rock-boring bivalves) in growth position (see enlargement in (d)). *P. penita* lives in waters 10 m deep or shallower, so**



(b) *Porites*, Granitos del Oro reef,
Gulf of Chiriqui

(c) *Pocillopora*, Contadora reef,
Gulf of Panama

bench elevation (3 m) plus maximum depth of growth (10 m) yields a *maximum-limiting* paleo-sea level of ~13
m above present.  Photographs by D.R. Muhs.

Figure 7.  (a) Map showing the distribution of living hermatypic corals and coral reefs along the Golfo de California

coasts of Mexico, the Pacific coast of Mexico, and the Pacific coast of Central America (compiled from Reyes-

Bonilla and López-Pérez, 2009; Alvarado et al., 2010; Glynn et al., 2017).  (b), (c) Examples of modern



**hermatypic corals along the Pacific coast of Central America (photographs courtesy of Lauren Toth, U.S.**
**Geological Survey).**







Figure 8. **Examples of fossil marine organisms used for geochronology of marine terrace deposits on the Pacific coast of North America: (a) solitary corals *Balanophyllia elegans* (fossil), San Nicolas Island, California (U-series dating); (b) *Porites panamensis* (fossil), Isla Carmen, Baja California Sur (U-series dating); (c) *Saxidomus* (fossil), San Nicolas Island, California (amino acid geochronology); (d) *Tegula* (fossil), San Clemente Island, California (amino acid geochronology); (e) *Chione* (modern, upper row; fossil, lower row), Cholla Bay, Sonora (amino acid geochronology). All photographs by D.R. Muhs.**

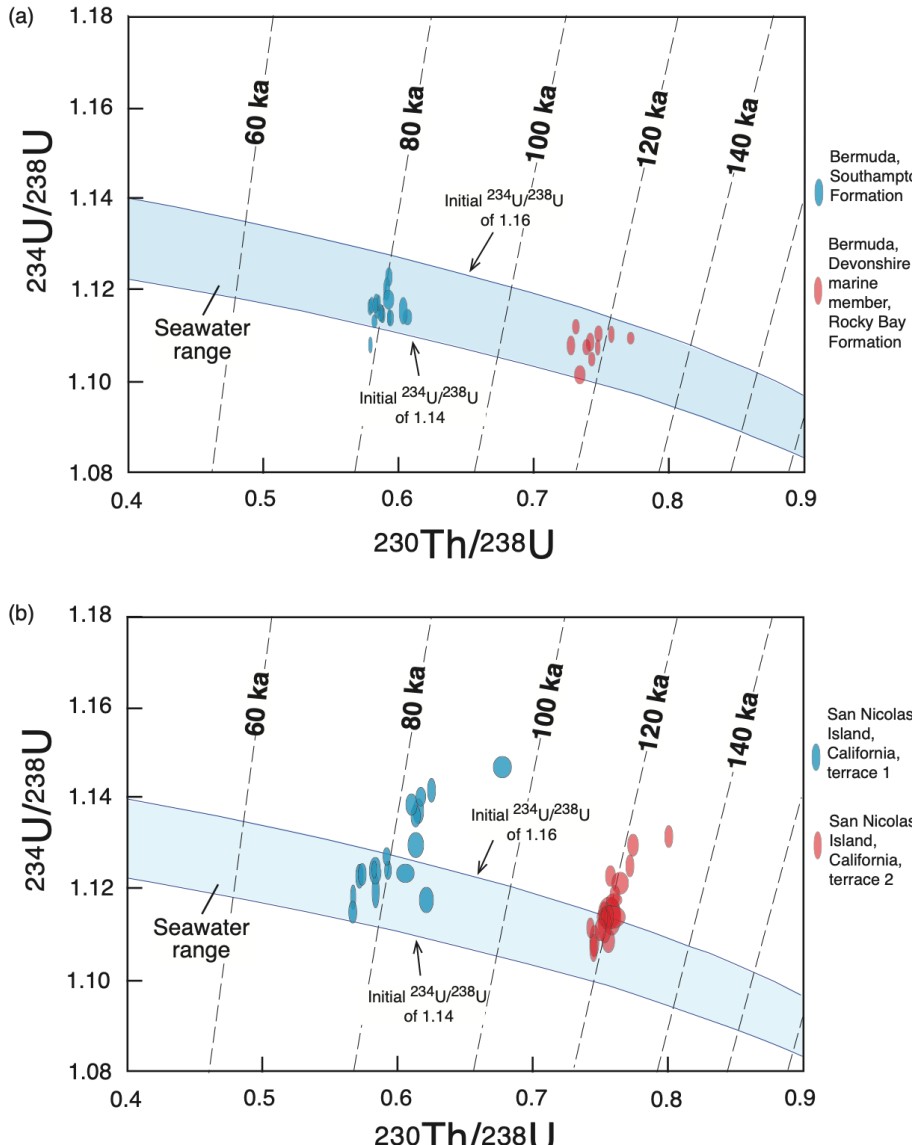

**Figure 9.** **Isotopic evolution diagrams for (a) colonial corals from the Southampton Formation (~80 ka) and Devonshire marine member of the Rocky Bay Formation (~120 ka) of Bermuda and (b) solitary corals from terrace 1 (~80 ka) and terrace 2 (~120 ka) of San Nicolas Island, California. Bermuda data are from Muhs et al. (2002a), but do not include two samples that have evidence of U loss. San Nicolas Island data are from Muhs et al. (2006). Blue bands define range of show isotopic evolution pathways for corals having mostly closed-system history and initial $^{234}$U/$^{238}$U activity values of 1.16 to 1.14, which *bracket* measured values in modern seawater (Chen et al., 1986; Delanghe et al., 2002) and modern corals (Muhs et al., 2002b).**





**Figure 10.** Plots of apparent $^{230}$Th/$^{238}$U ages vs. back-calculated initial $^{234}$U/$^{238}$U values in solitary corals from (a) Eel Point terrace, San Clemente Island, California, (b) Terrace 2, west end of San Nicolas Island, California [same as in Fig. 9b, but with different scales], (c) Sea Cave terrace, Punta Banda, Baja California, and (d) colonial corals (mostly *Acropora palmata*) from the Rendezvous Hill terrace, north end of Barbados, West Indies. San Clemente Island and Punta Banda data are from Muhs et al. (2002b), San Nicolas Island data are from Muhs et al. (2006), and Barbados data are from Muhs and Simmons (2017). Also shown (blue bands) is the range of $^{234}$U/$^{238}$U activity values in modern seawater (Chen et al., 1986; Delanghe et al., 2002). Note that in both solitary corals and colonial corals, samples plotting above seawater values tend to be biased to older apparent ages, but the degree of bias varies from locality to locality.



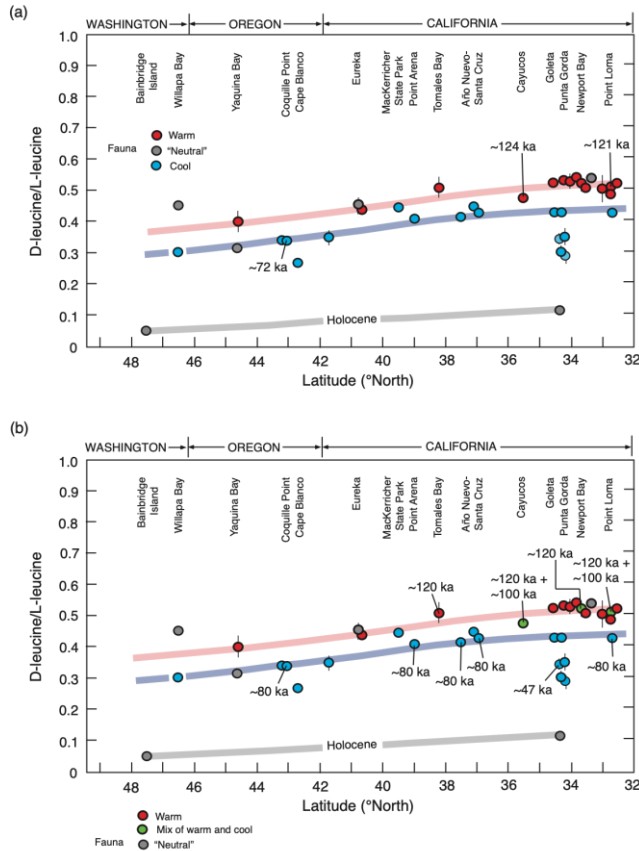

Figure 11. (a) Plot showing D-leucine/L-leucine in fossil *Saxidomus* shells (or equivalent values converted from *Leukoma staminea* shells; see Lajoie et al., 1980) from marine terrace deposits of the Pacific Coast of the USA, from Kennedy et al. (1982). Localities are arranged from north (left) to south (right), parallel to latitudinal trend of mean annual air temperatures increasing to the south. Samples plotting along the pink line are correlated with MIS 5e (~120 ka) based on calibration to U-series-dated corals from Cayucos and Point Loma; samples plotting along the blue line are correlated to MIS 5a (~80 ka), based on U-series-dated corals from Coquille Point, Oregon. Calibration points used are the only ones that were available at the time of the original study. Samples plotting below these lines are correlated with MIS 3 or to Holocene-dated deposits (gray line). Not included from the original study are data points from Whidbey Island, Washington, which are interpreted to be from glaciomarine deposits (Polenz et al., 2009). Colors of circles indicate molluscan fauna thermal aspects (see discussion of Fig. 13). (b) Plot of same data as in (a), except new U-series ages of corals, generated since 1982, have been added and thermal aspects of some faunas have been modified (Muhs et al., 2002b, 2006, 2014b).


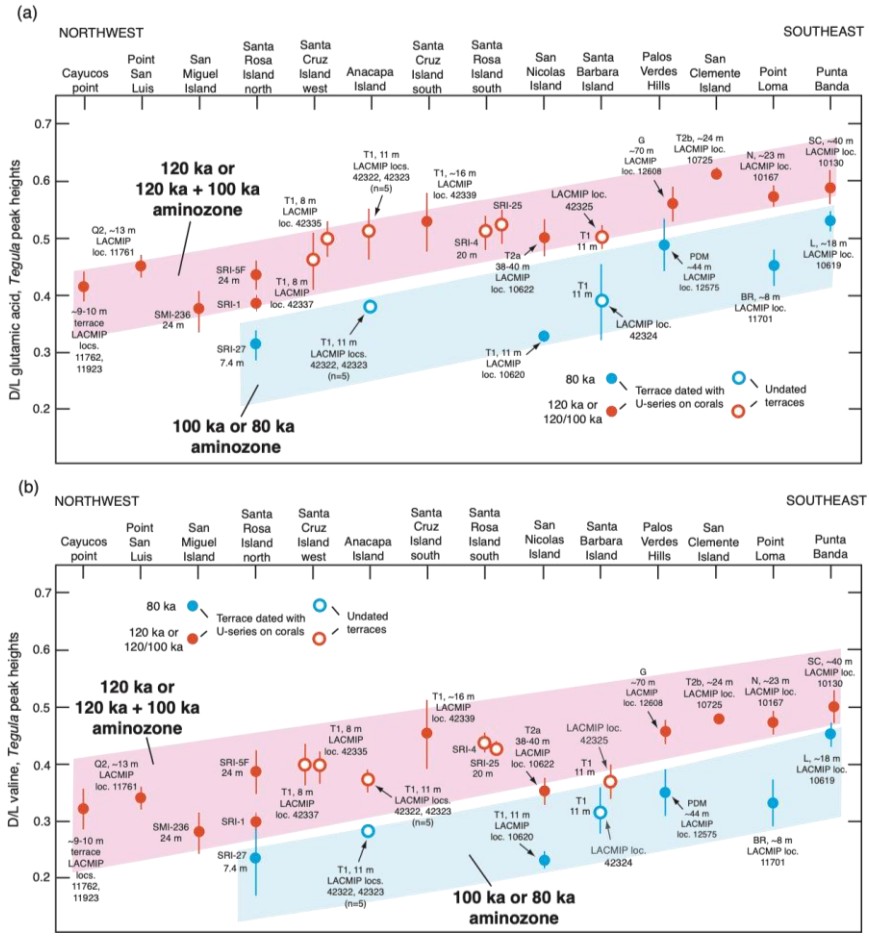

**Figure 12.** **(a) Plot of mean D/L values in glutamic acid (vertical axis) in fossil *Tegula* from dated (filled circles) and undated (open circles) marine terraces on the California and Baja California coast, shown as a function of latitude (horizontal axis) as a proxy for long-term temperature history, cooler in the northwest, warmer in the southeast. Error bars are ± 1 standard deviation, based on D/L values in 3 to 6 individual shells from the same deposit. Colored bands ("aminozones") indicate correlation between fossil localities of the same age, anchored by U-series dating of corals. Terrace name abbreviations: SMI, San Miguel Island, SRI, Santa Rosa Island; SCRZI, Santa Cruz Island; N, Nestor; BR, Bird Rock; PDM, Paseo del Mar; G, Gaffey; SC, Sea Cave; L, Lighthouse; see Muhs et al. (1994, 2002b, 2006, 2014b, 2015) for terrace stratigraphic names and U-series ages. Data from Santa Cruz Island-West, Santa Cruz Island-South, Santa Barbara Island, and Anacapa Island are from Muhs and Groves (2018); all other data are from Muhs et al. (2014b). (b) Same as in (a), but for mean D/L values in valine.**

**Figure 13.** **Modern geographic ranges of extralimital and northward or southward-ranging fossil mollusks found in ~80,000 yr B.P. marine terrace deposits at Green Oaks Creek, Point Año Nuevo and Santa Cruz, California and the ~130,000 yr B.P. Millerton Formation at Toms Point, Tomales Bay, California. Fossil data for the Millerton Formation are from Johnson (1962); Davenport terrace fossil data are from Addicott (1966) and Muhs et al. (2006). Modern species names and geographic ranges updated by the author from Abbott and Haderlie (1980), O'Clair and O'Clair (1998), and Coan et al. (2000).**

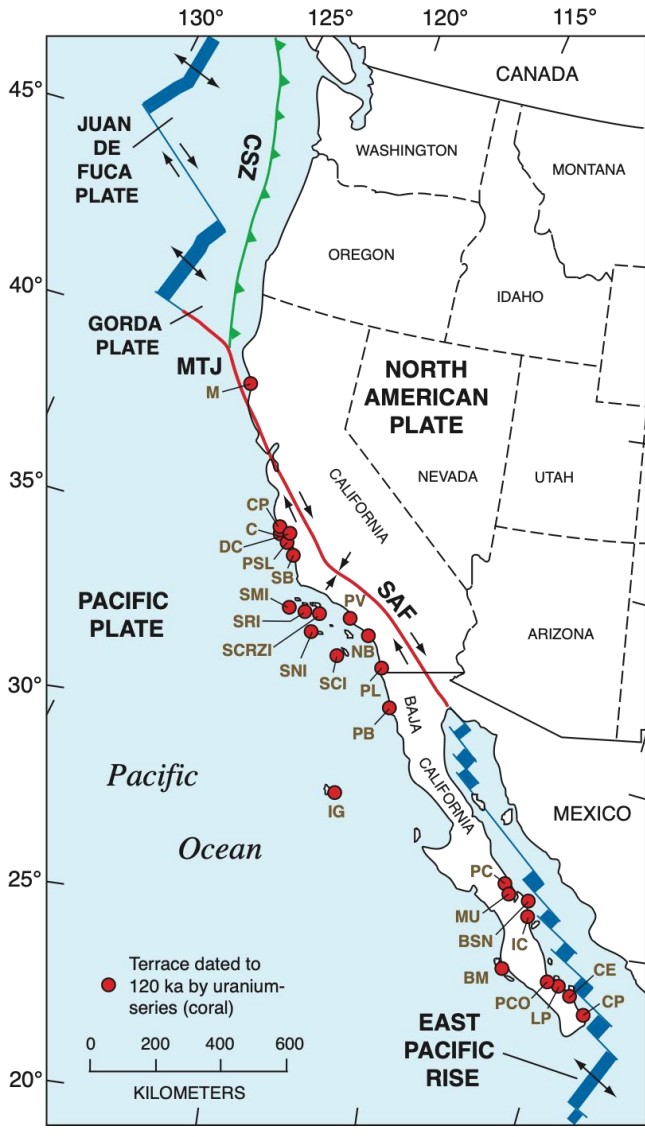

Figure 14. **Map of the Pacific Coast of North America with structural features as shown in Figure 2, but also plotted are localities (filled red circles) where U-series ages of corals dating to MIS 5e (~120 ka) have been reported. Abbreviations are keyed to Table S1 and are as follows: CP, Cayucos Point; C, Cayucos; DC; Diablo Canyon; PSL, Point San Luis; SB, Shell Beach; SMI, San Miguel Island; SRI, Santa Rosa Island; SCRZI, Santa Cruz Island; SNI, San Nicolas Island; PV, Palos Verdes Hills; NB, Newport Beach; SCI, San Clemente Island; PL, Point Loma; PB, Punta Banda; IG, Isla Guadalupe; BM, Bahía Magdalena; CP, Cabo Pulmo; CE, Isla Cerralvo; LP, La Paz; PCO, Punta Coyote; IC, Isla Coronado; BSN, Bahía San Nicolas; MU, Mulegé; PC, Punta Chivato.**

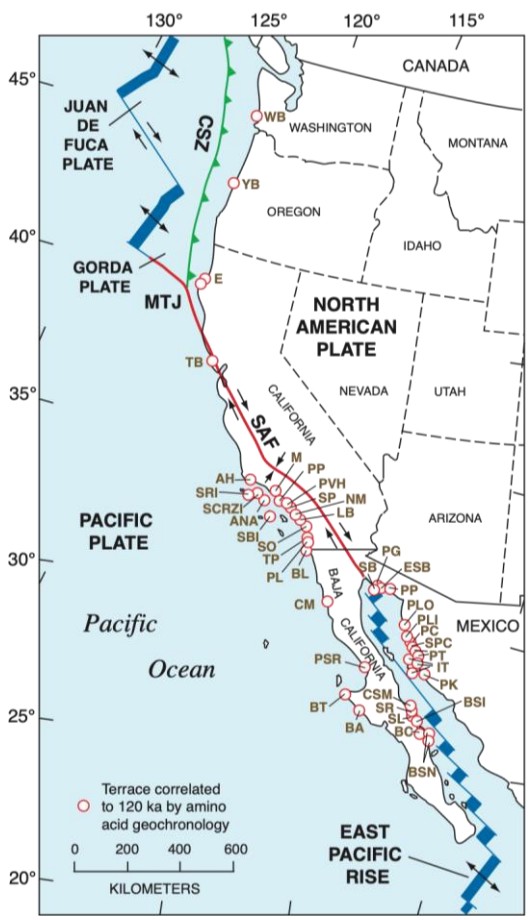

**Figure 15.** Map of the Pacific Coast of North America with structural features as shown in Figure 2, but also plotted are localities (open red circles) where amino acid geochronology has permitted correlation of marine deposits to MIS 5e (~120 ka). Abbreviations are keyed to Table S2 and are as follows: WB, Willapa Bay; YB, Yaquina Bay; E, Eureka; TB, Tomales Bay; AH, Arroyo Hondo; SRI, Santa Rosa Island; SCRZI, Santa Cruz Island; ANA, Anacapa Island; SBI, Santa Barbara Island; M, Malibu; PP, Pacific Palisades; PVH, Palos Verdes Hills; SP, San Pedro; NM, Newport Mesa; LB, Laguna Beach; SO, San Onofre; TP, Torrey Pines; PL, Point Loma; BL, Border locality; CM, Camalú; PSR, Punta Santa Rosalíllíta; BT, Bahía Tortuga; BA, Bahía Asunción; BSN, Bahía San Nicolas North; BC, Bahía Concepción; BSI, Bahía Santa Inés; SL, San Lucas; SR, Santa Rosalia; CSM, Caleta Santa Maria; SB, Salina la Borrascosa; PG, Punta Gorda; ESB, East of Salina la Borrascosa; PLO, Puerto Lobos; PLI, Puerto Libertad; PC, Punta Cuevas; SPC, Southeast of Punta Cuevas; PT, Punta Tepopa; IT, Isla Tiburón; PK, Punta Kino; CSZ, Cascadia Subduction Zone; MTJ, Mendocino Triple Junction; SAF, San Andreas Fault.



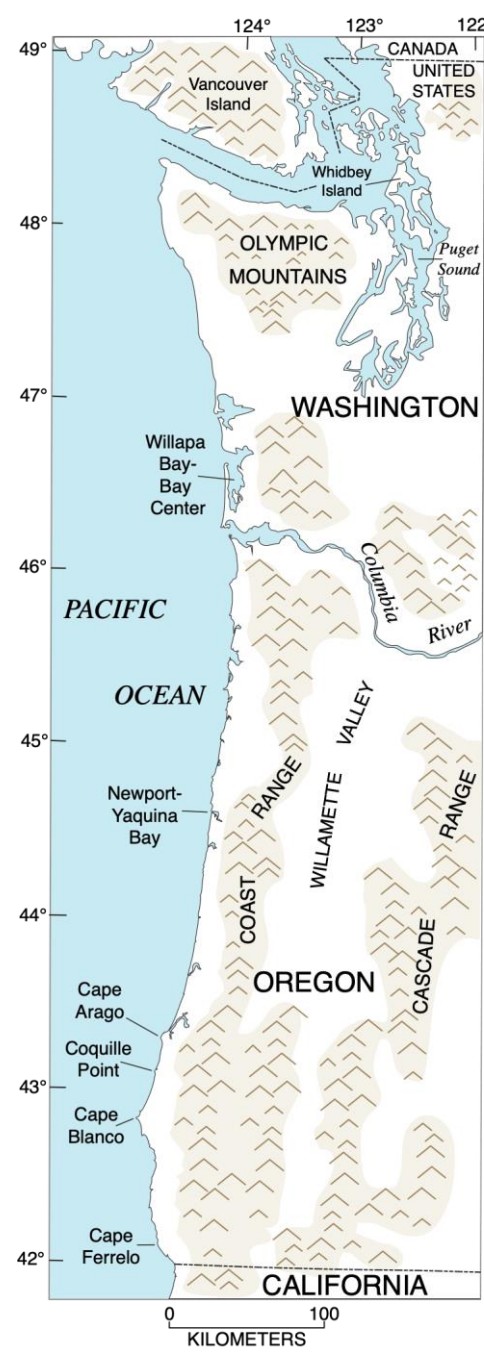

**Figure 16: Map of southwestern Canada, Washington, and Oregon, showing localities referred to in the text.**



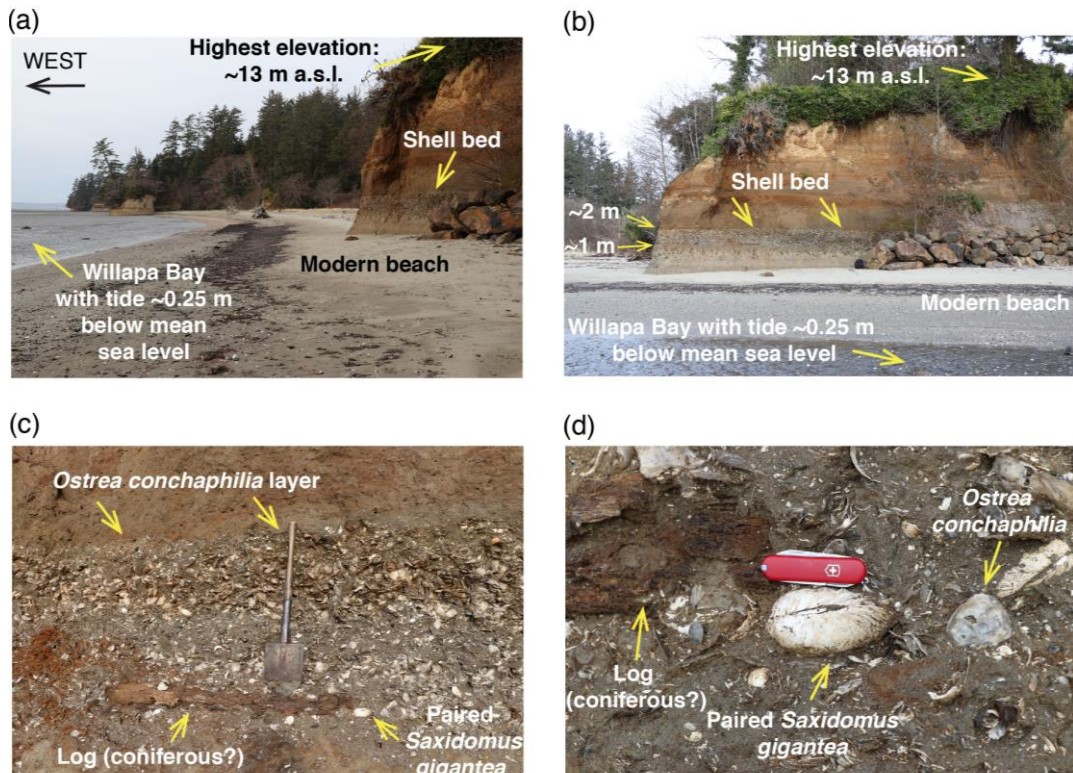

**Figure 17: Photographs of possible MIS 5e marine deposits in the Willapa Bay area, near Bay Center, Washington:**
**(a), (b) location of shell-bearing layer relative to modern sea level; (c) (d) closeup views showing shell bed with**
***Ostrea conchaphilia*, *Saxidomus gigantea*, and coniferous wood fragments. All photographs by D.R. Muhs.**

**Figure 18: Maps of marine terraces in the Coquille Point (a) and Cape Blanco (b) areas of southwestern Oregon.** (a) Qwr, Whisky Run terrace deposits; Qp, Pioneer terrace deposits; Qsd, Seven Devils terrace deposits (correlated to MIS 5e); Qm, Metcalf terrace deposits. (b) Qcb, Cape Blanco terrace deposits; Qp, Pioneer terrace deposits; Qsb, Silver Butte terrace deposits (correlated to MIS 5e); Qic, Indian Creek terrace deposits; Qpr, Poverty Ridge terrace deposits. Redrawn from terrace maps in McInelly and Kelsey (1990) for (a) and Kelsey (1990) for (b).

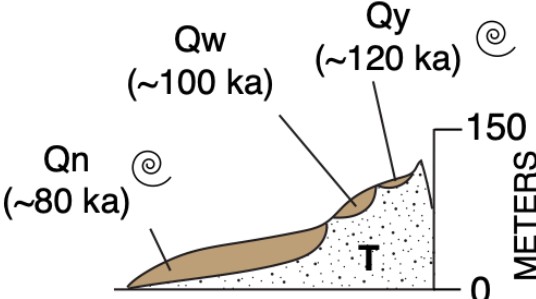

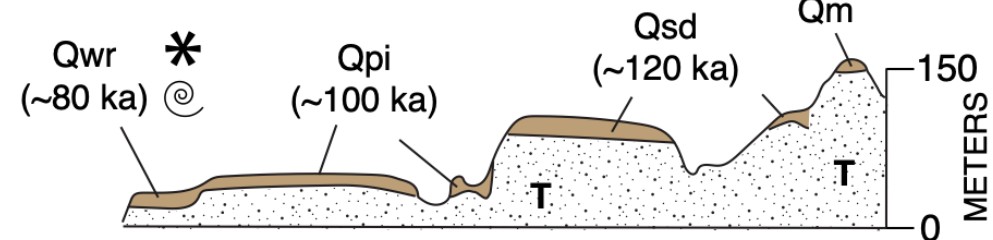

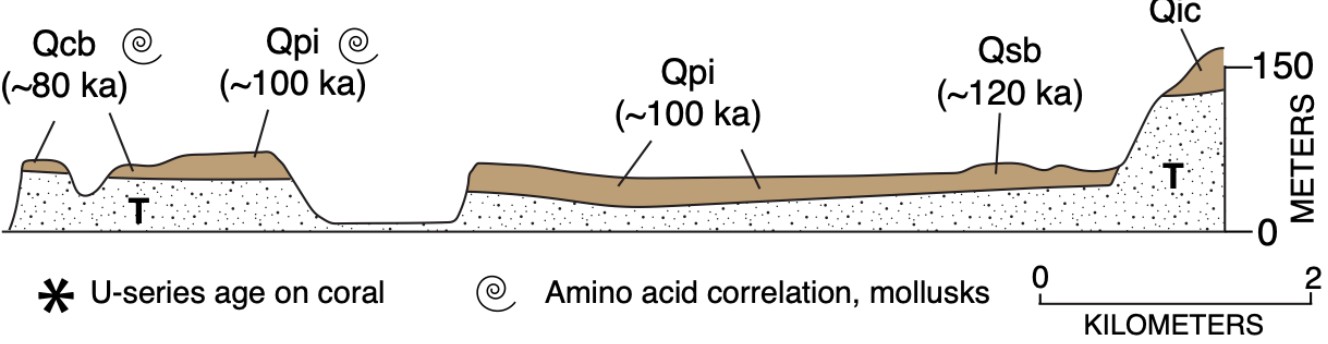

**Figure 19: Cross sections of marine terrace deposits in coastal Oregon in the Newport-Yaquina Bay area (a), Cape**
**Arago-Coquille Point area (b), and Cape Blanco area (c), and estimated ages based on U-series dating of corals,**
**amino acid geochronology of mollusks, and degree of soil development. Cross sections from Kelsey et al. (1996)**
**for (a), McInelly and Kelsey (1990) for (b), and Kelsey (1990) for (c); geochronological data from Kennedy et**
**al. (1982), Muhs et al. (1990, 2006), and Kelsey et al. (1996). Deposit abbreviations as defined in Figure 18**
**caption.**





**Figure 20. Photographs for the Pioneer (~100 ka, MIS 5c?) and Cape Blanco (~80 ka, MIS 5a?) marine terraces (a) in the Cape Blanco area, with closeups wave-cut bench on Miocene sandstone, shell bed, and terrace sediments (b, c). All photographs by D.R. Muhs.**

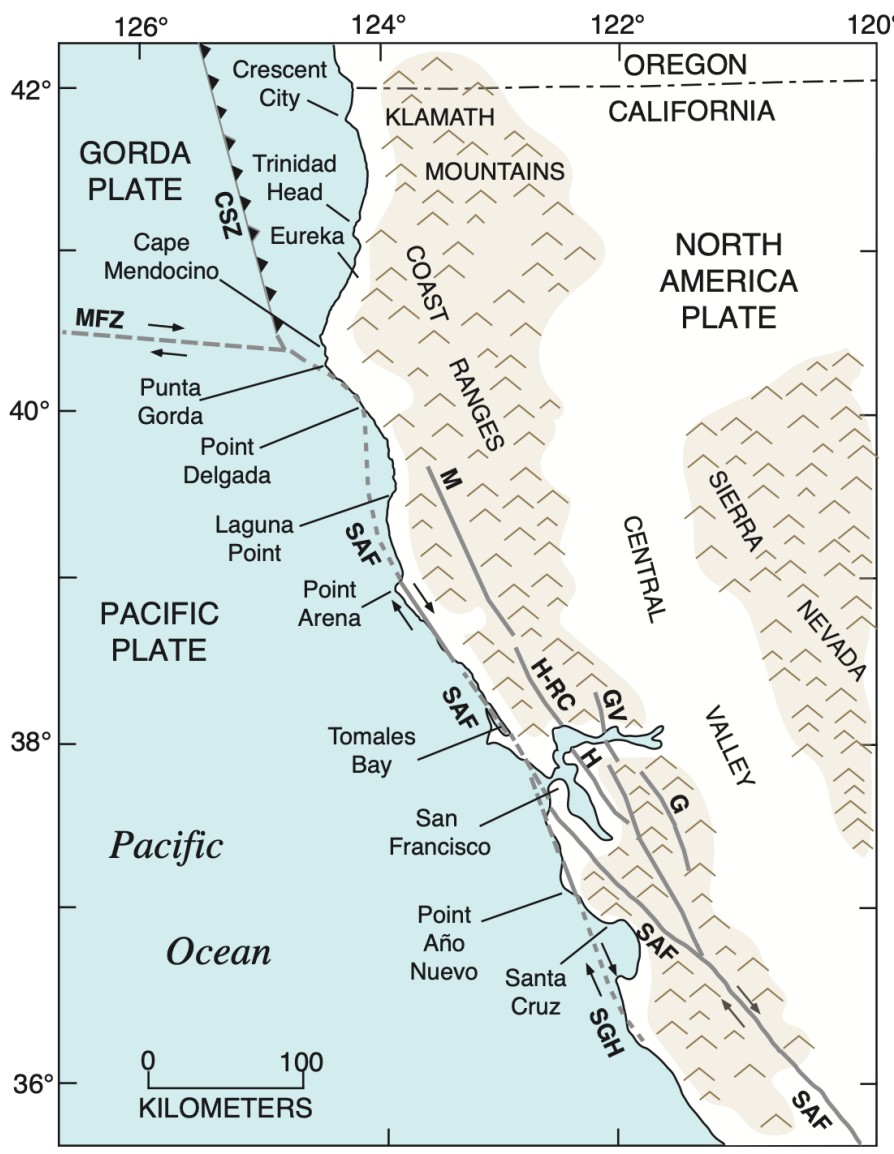

**Figure 21.  Map of coastal northwestern California showing geographic and structural features and locations discussed**
**in the text.**



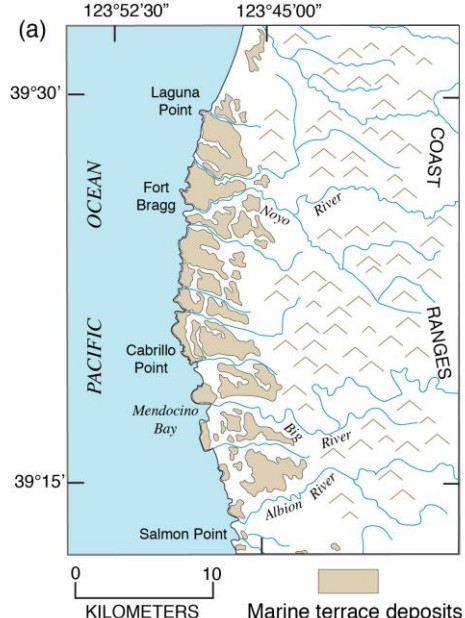

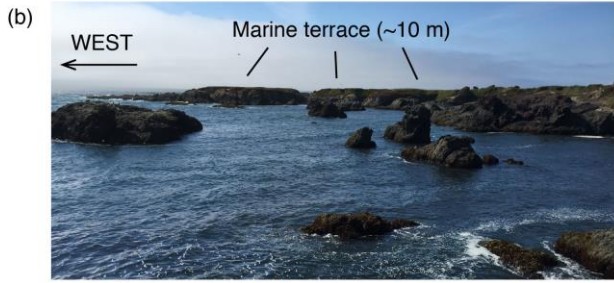

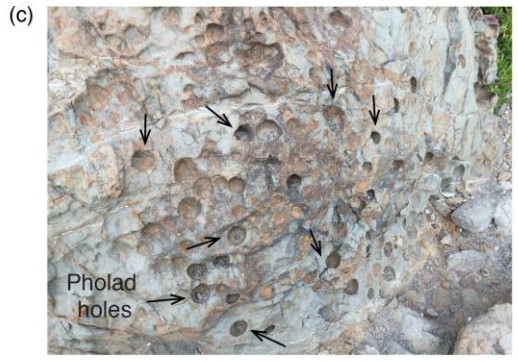

**Figure 22: (a) Map showing the distribution of marine terraces in the Laguna Point-Fort Bragg-Cabrillo Point area (redrawn from Jennings and Strand, 1960); (b) photograph of 10-m-high marine terrace, correlated to MIS 5e (Merritts and Bull, 1989); (c) pholad holes in outer edge of 10-m-high, MIS 5e terrace. Photographs by D.R. Muhs.**



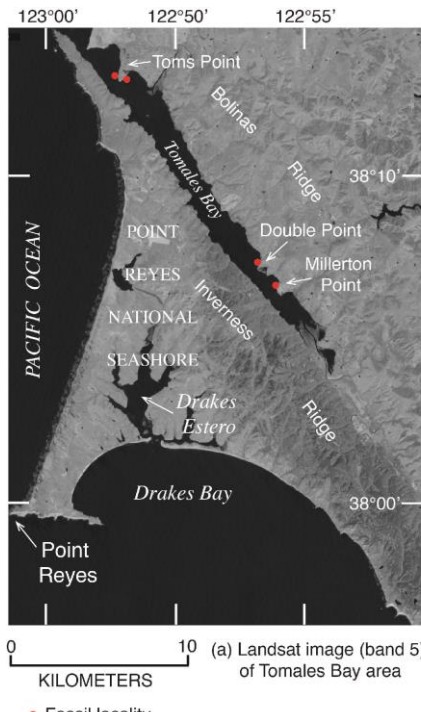

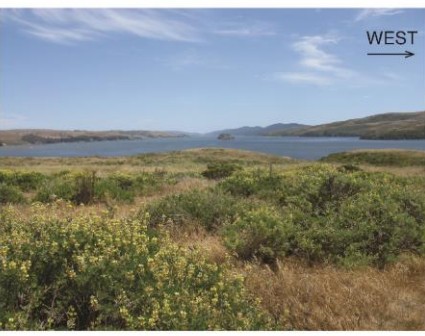

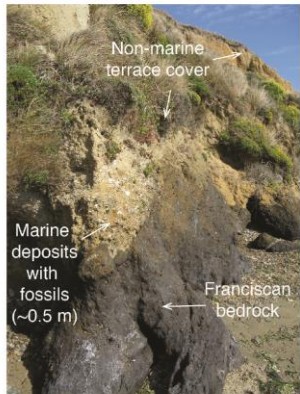

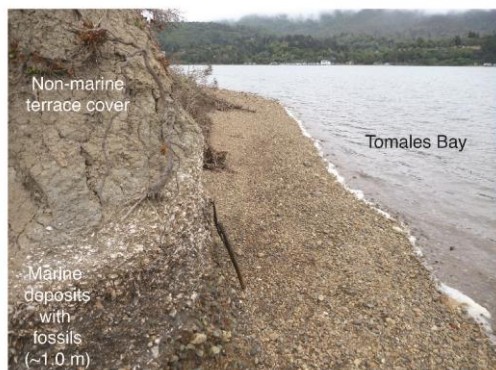

Figure 23: (a) Landsat band 5 image (from U.S. Geological Survey) of the Tomales Bay, California area, with fossil localities of the Millerton Formation (filled red circles) from Johnson (1962); formation is correlated to MIS 5e by thermoluminescence (Grove et al., 1995) and amino acid geochronology (Muhs and Groves, 2018). (b) Ground photograph from Toms Point, looking south to Tomales Bay. (c) Marine terrace deposits with fossils exposed on wave-cut bench at Toms Point. (d) Marine deposits with fossils exposed at Millerton Point. Photographs by D.R. Muhs.

**Figure 24. Map of a portion of the central coast of California, from Santa Cruz to just north of Point Año Nuevo, showing marine terraces, fossil localities, and location of the San Gregorio fault zone (solid gray lines; dashed where uncertain). Marine terrace inner edges redrawn from Bradley and Griggs (1976) and Weber et al. (1979); location of the San Gregorio fault zone from Weber et al. (1979) and Weber (1990).**





**Figure 25.** Map of geographic and structural features in southern California and localities referred to in the text. Grey lines are faults from Jennings (1994). AI, Anacapa Island; SBI, Santa Barbara Island.



**Figure 26.** Map of marine terrace deposits (brown shades), terrace inner edges (black lines), fossil localities (open red circles), landslide deposits (gray shades), and faults in the Palos Verdes Hills area, Los Angeles County, California. Redrawn from Woodring et al. (1946). PVS, Palos Verdes Sand, part of which is of MIS 5e age (Muhs and Groves, 2018); PDM, Paseo del Mar terrace (~80 ka, MIS 5c); G, Gaffey terrace (~120 ka, MIS 5e; Muhs et al., 2006). LACMIP, Natural History Museum of Los Angeles County Invertebrate Paleontology fossil locality numbers; WBK, fossil localities of Woodring et al. (1946).

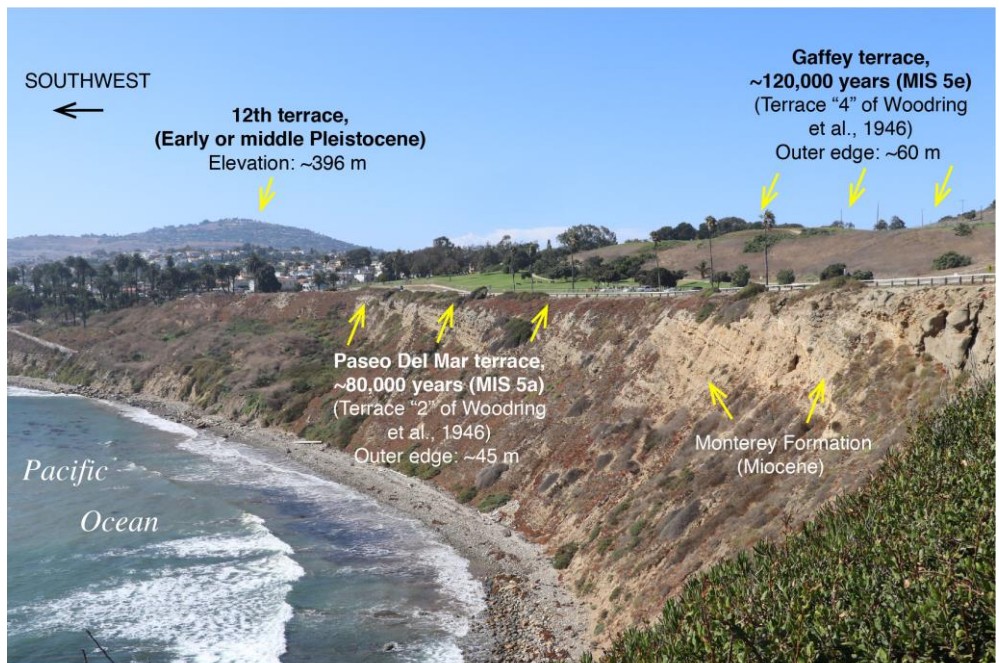

**Figure 27: View of outer edges of the Paseo del Mar (~80 ka) and Gaffey (~120 ka) terraces, looking northwest from Point Fermin (see Fig. 26). On left side of photograph, a small fragment of the 12th terrace is visible (=LACMIP loc. 1304 on Fig. 26). Photograph by D.R. Muhs.**

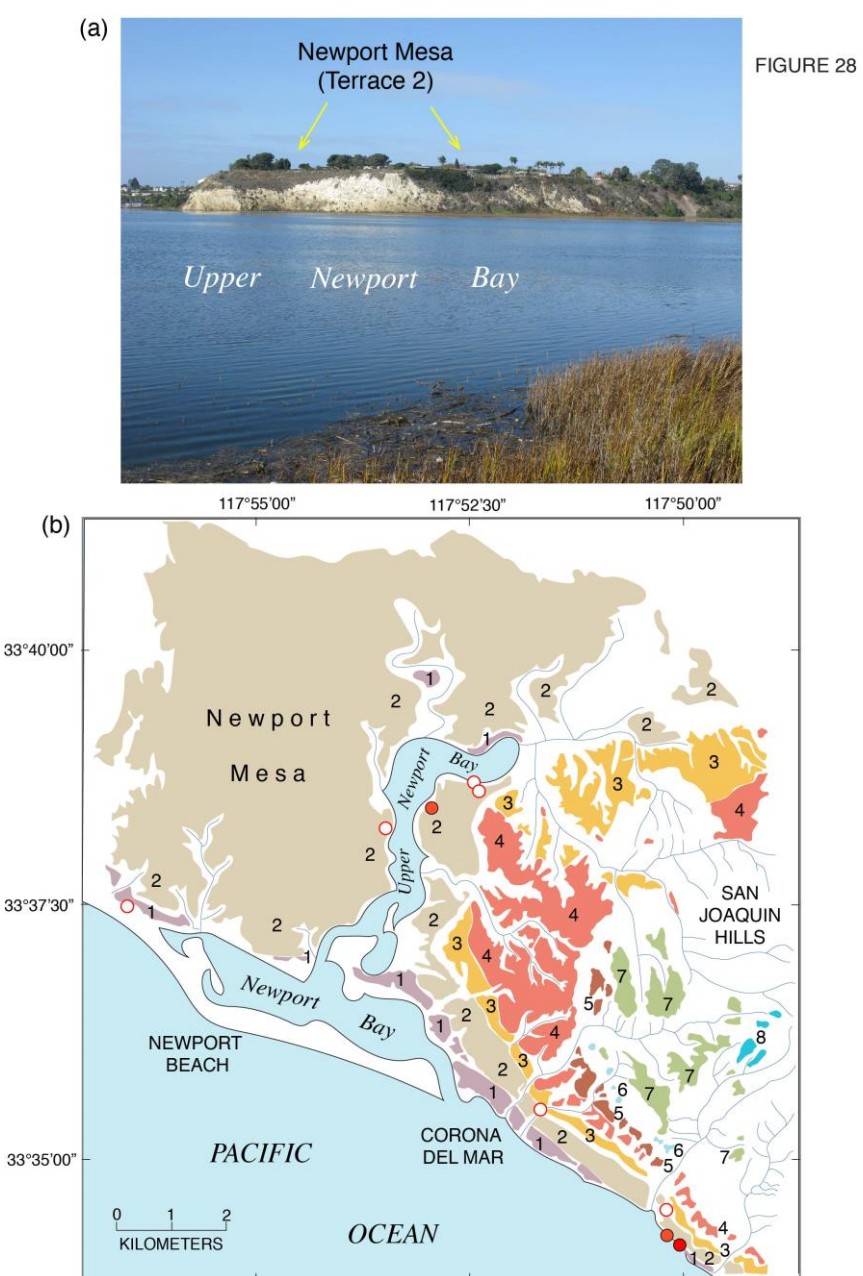

Figure 28: (a) Photograph of Newport Mesa (terrace 2) of MIS 5e age, view to the west, upper Newport Bay in the foreground. (b) Map of marine terraces in the Newport Bay area, redrawn from Vedder et al. (1957, 1975) and Grant et al. (1999). Filled red circles are fossil localities with U-series ages from Grant et al. (1999); open red circles are amino acid geochronology fossil localities of Wehmiller et al. (1977a). Photograph in (a) by D.R. Muhs.

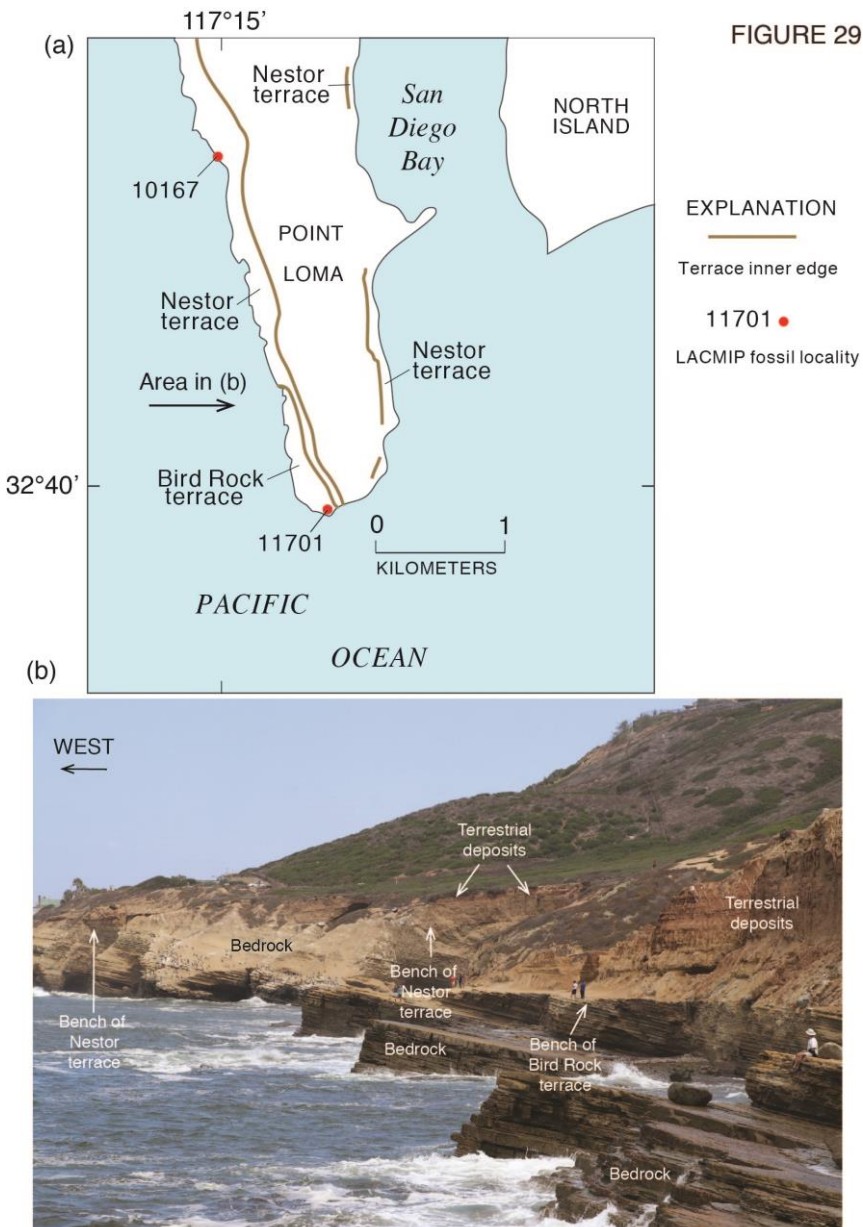

Figure 29: (a) Map showing marine terrace inner edges (redrawn from Kern, 1977) and LACMIP fossil localities where U-series ages are reported by Muhs et al. (1994; 2002b); corals from the Nestor terrace date to ~120 ka (MIS 5e) and ~100 ka (MIS 5c); those from the Bird Rock terrace date to ~80 ka (MIS 5a). (b) Photograph showing outer edges of the Nestor and Bird Rock terraces on the west coast of Point Loma, looking north (see location in (a)). Photograph by D.R. Muhs.

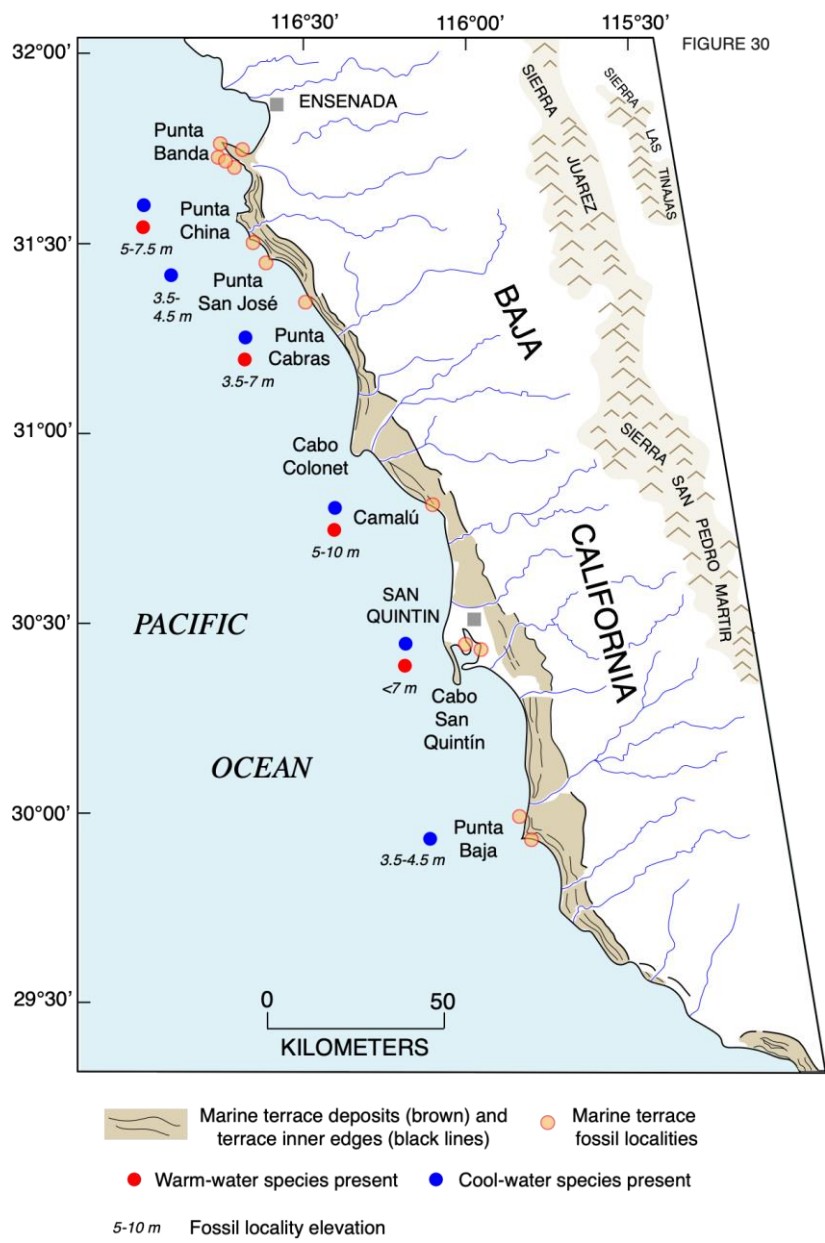

Figure 30: Map of northwestern Baja California, showing marine terrace deposits (brown shades), terrace inner edges (black lines), and fossil localities (red/orange circles). Punta Banda fossil localities are shown in Figure 31. South of Punta Banda, most fossil localities are undated, but based on elevations (shown), many likely date to some part of MIS 5 and could contain mixes of fossils of two ages, with cool-water (blue dots) and warm-water (red dots) faunas (see text for discussion). Marine terrace deposit and inner edge mapping redrawn from Orme (1980).



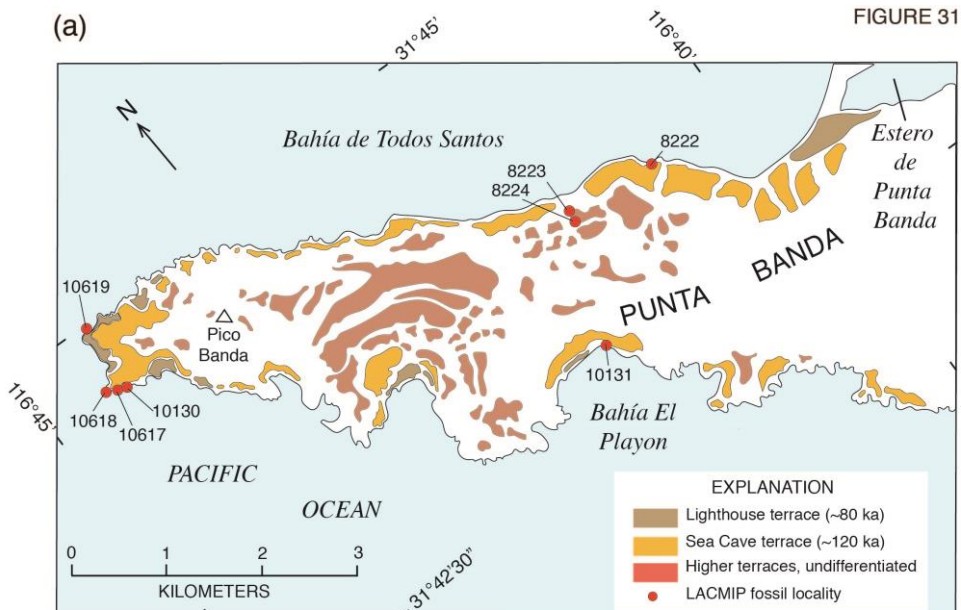

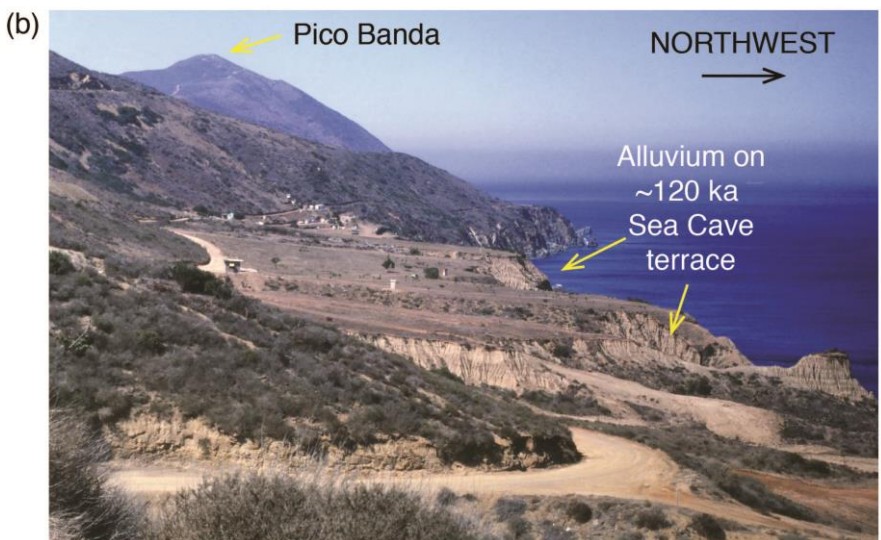

Figure 31: (a) Map of the Punta Banda area, south of Ensenada, Baja California (see Fig. 30 for location). Lighthouse terrace is dated to MIS 5a (~80 ka) and Sea Cave terrace is dated to MIS 5e (~120 ka), both by U-series on corals (Rockwell et al., 1989; Muhs et al., 2002b). Terrace mapping is redrawn from Rockwell et al. (1989). (b) Photograph of the ~120 ka Sea Cave terrace on the northeast side of Punta Banda, showing thick alluvial cover. Photograph by D.R. Muhs.

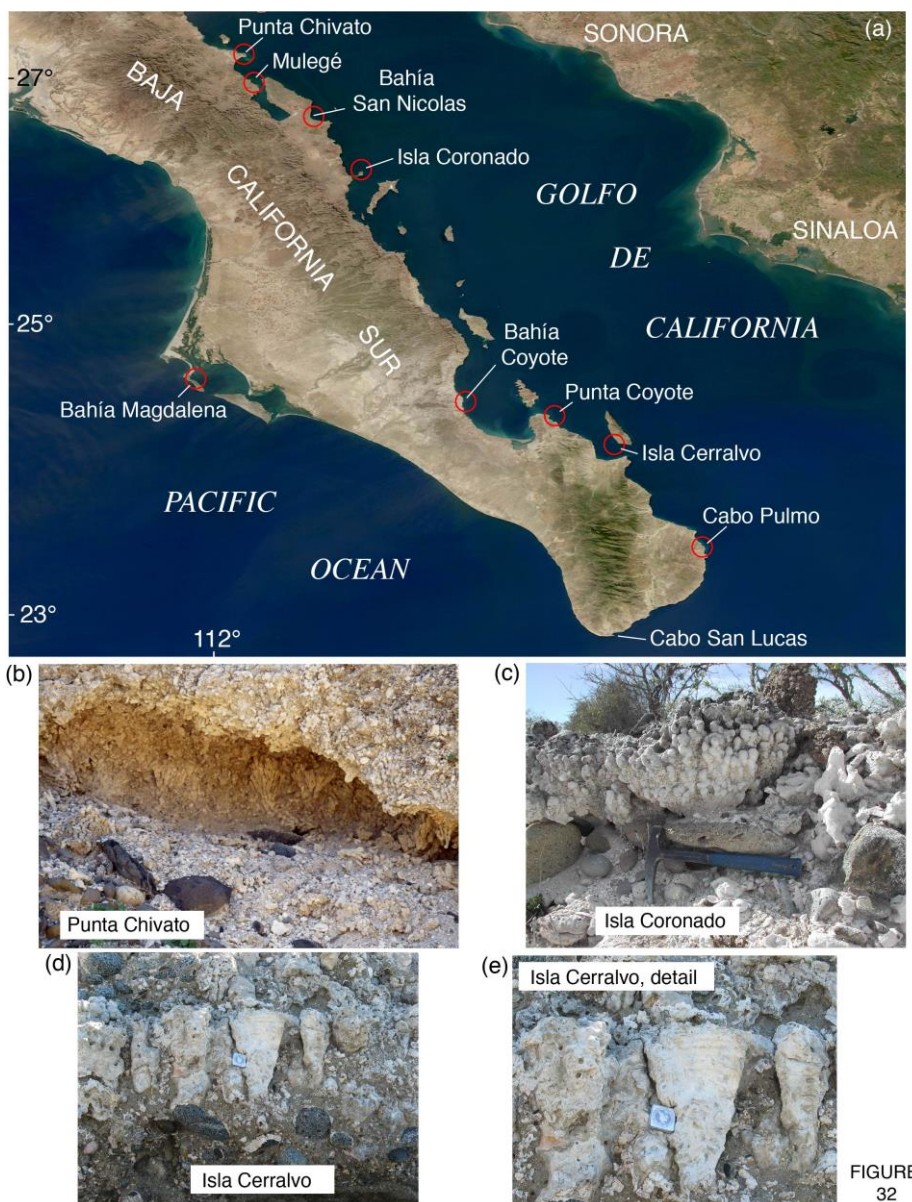

Figure 32: (a) Image of southernmost Baja California Sur and parts of Sonora and Sinaloa, acquired on 27 November 2011 using the MODIS instrument on the Aqua satellite (courtesy of the NASA Rapid Response Team). Red circles indicate marine terrace localities where U-series ages on corals have yielded MIS 5e ages (see Table S1). (b), (c), (d), (e) Photographs of growth-position corals dated to MIS 5e from Punta Chivato, Isla Coronado, and Isla Cerralvo (see Johnson, 2002; Johnson et al., 2007; Tierney and Johnson, 2012). All photographs are courtesy of Markes Johnson, Williams College.

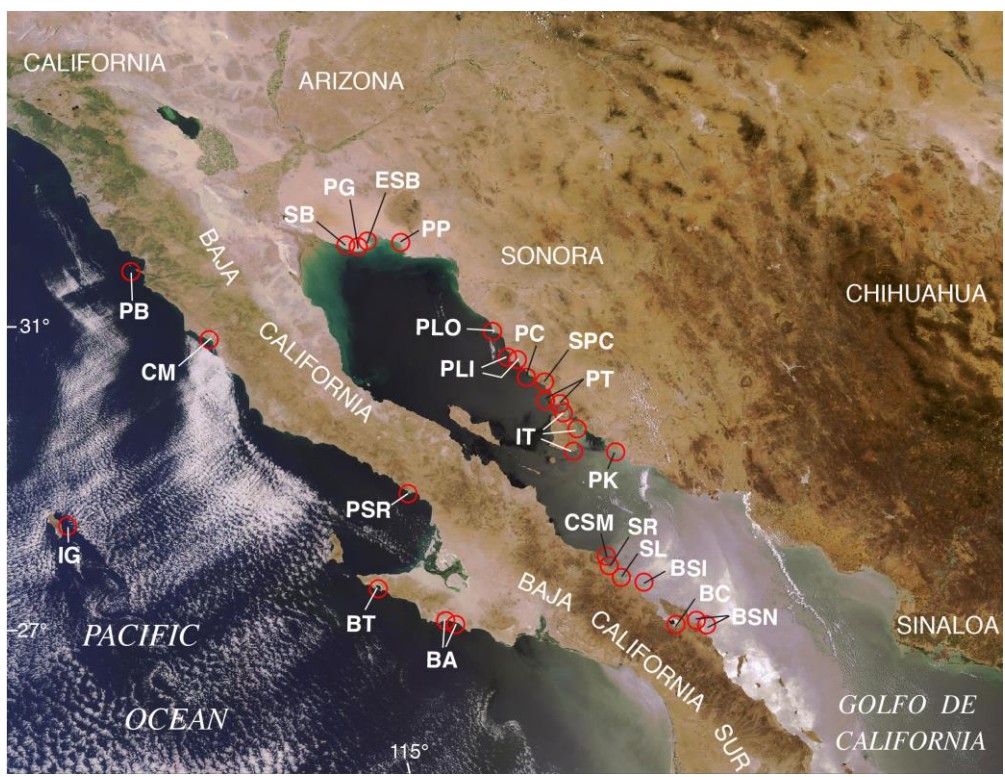

**Figure 33: Satellite image of northwestern Mexico (Baja California, Baja California Sur, Sonora, and parts of adjacent areas) showing localities where amino acid geochronology of fossil mollusks has yielded MIS 5e ages (Ortlieb, 1987, 1991; Valentine, 1980; Woods, 1980; Emerson et al., 1981; Keenan et al., 1981). Also shown for reference are two U-series-dated MIS coral localities, Punta Banda (PB) and Isla Guadalupe (IG). Image acquired 28 May 2006 by the Medium Resolution Imaging Spectrometer (MERIS) onboard the Envisat satellite, courtesy of the European Space Agency. Abbreviations of amino acid localities (see Table S2): CM, Camalú; PSR, Punta Santa Rosalíllíta; BT, Bahía Tortuga; BA, Bahía Asunción; BSN, Bahía San Nicolas North; BC, Bahía Concepción; BSI, Bahía Santa Inés; SL, San Lucas; SR, Santa Rosalia; CSM, Caleta Santa Maria; SB, Salina la Borrascosa; PG, Punta Gorda; ESB, East of Salina la Borrascosa; PLO, Puerto Lobos; PLI, Puerto Libertad; PC, Punta Cuevas; SPC, Southeast of Punta Cuevas; PT, Punta Tepopa; IT, Isla Tiburón; PK, Punta Kino.**



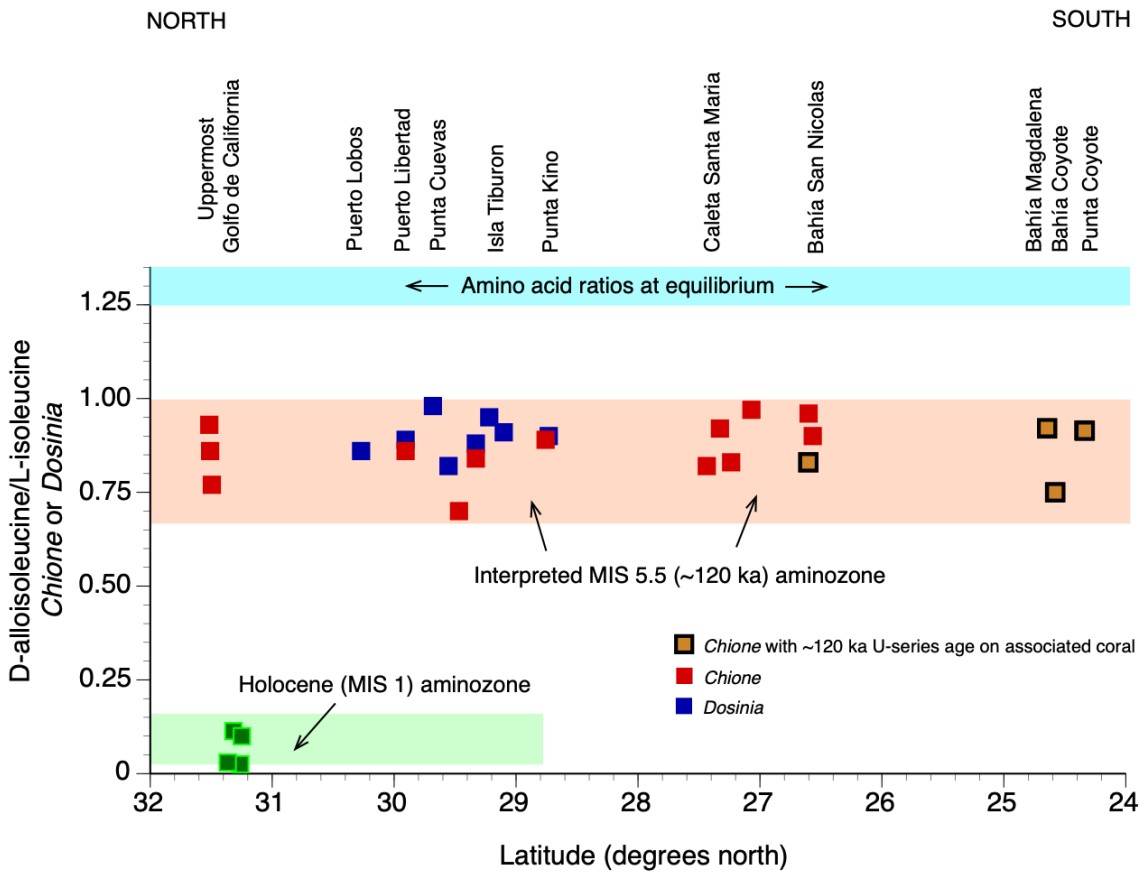

**Figure 34: Plot showing ratios of D-alloisoleucine to L-isoleucine in fossil *Chione* (red squares) and *Dosinia* (blue squares) as a function of latitude in Baja California, Baja California Sur, and Sonora (localities listed in Table S2). Amino acid data from Ortlieb (1987, 1991), and Umhoefer et al. (2014). Also shown (solid green squares) are ratios of D-alloisoleucine to L-isoleucine in late Holocene, radiocarbon-dated *Chione* shells from the upper Golfo de California (data from Martin et al., 1996) and ratios of D-alloisoleucine to L-isoleucine (blue shade) at equilibrium (from Miller and Mangerud, 1985). Pink shade defines aminozone correlated to MIS 5e, based on calibration with U-series-dated deposits that also contain *Chione* shells (gold squares).**

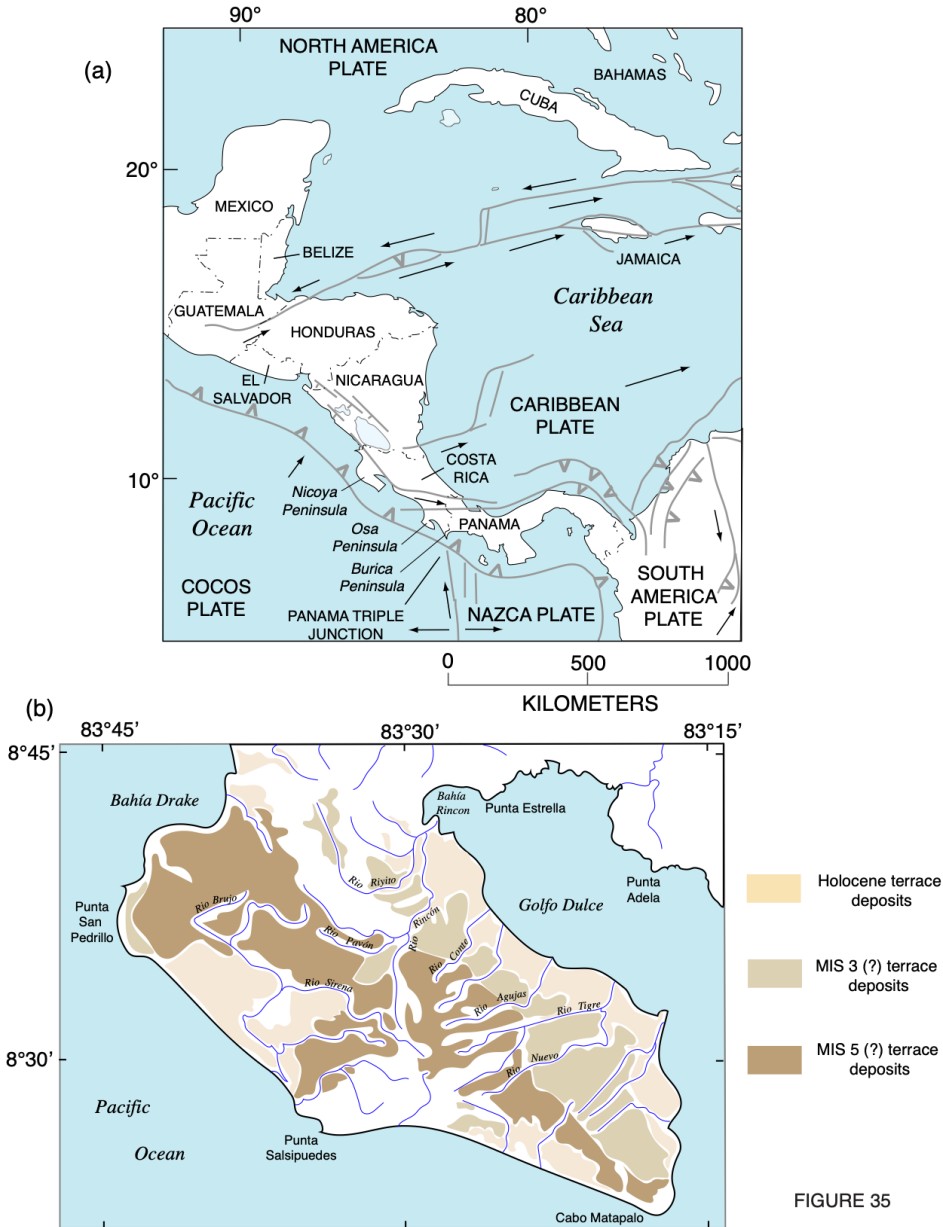

Figure 35: (a) Map of Central America, showing structural features (redrawn from Mann, 2007, and Pindell and Kennan 2009), lithospheric plates, directions of present plate movements (arrows), and localities referred to in text. (b) Map showing marine terrace deposits on the Osa Peninsula of Costa Rica (redrawn from Gardner et al., 2013); legend shows possible correlation to marine isotope stages (MIS).

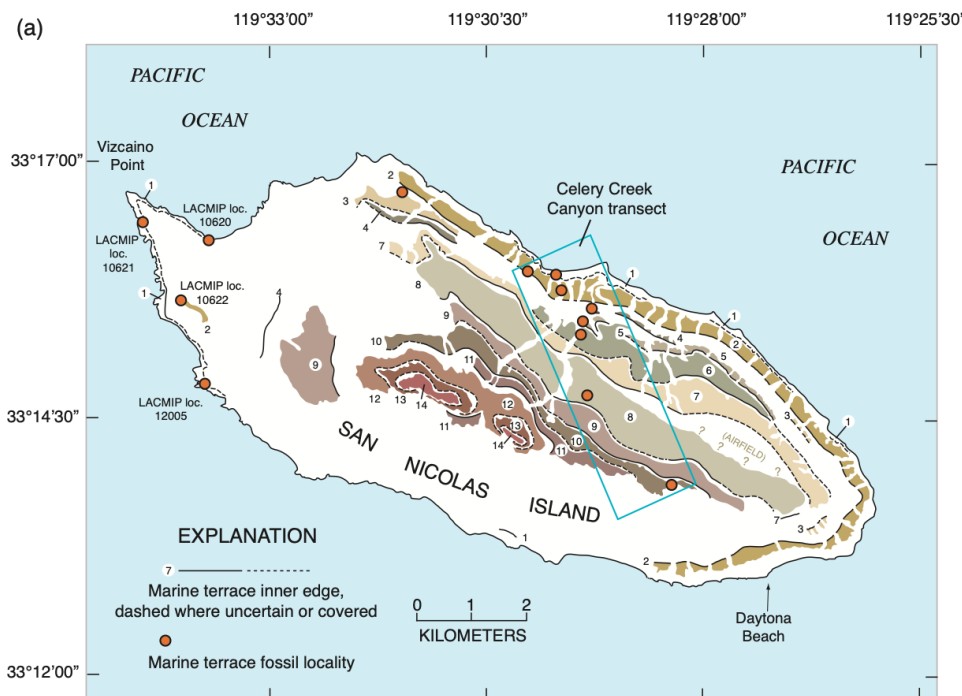

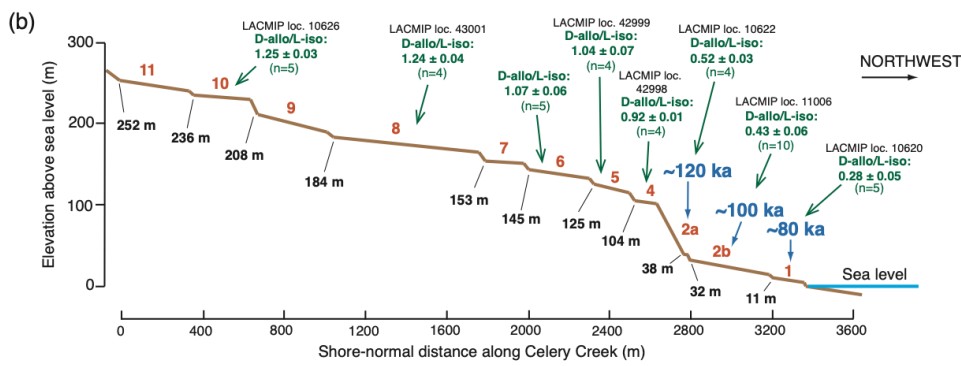

**Figure 36.** **(a) Map of marine terraces on San Nicolas Island, California (from Vedder and Norris, 1963; Muhs et al., 2012); orange dots are fossil localities. (b) Topographic profile across the lowest 11 terraces in the Celery Creek Canyon area (blue boxed area in (a)), showing terrace numbers, shoreline angle elevations (from Muhs et al., 2018), and (green lettering) ratios of the amino acids D-alloisoleucine to L-isoleucine in the fossil gastropod *Tegula* (data mostly from Muhs, 1985). Assuming a constant rate of uplift, terrace 4 could be ~390 ka (~MIS 11). Note that large jump in elevation between terraces 2a and 4 is mirrored by large increase in amino acid ratios, indicating the likelihood that terraces representing MIS 9 (~300 ka) and MIS 7 (~200 ka) were removed**



**by sea cliff retreat during MIS 5e.**

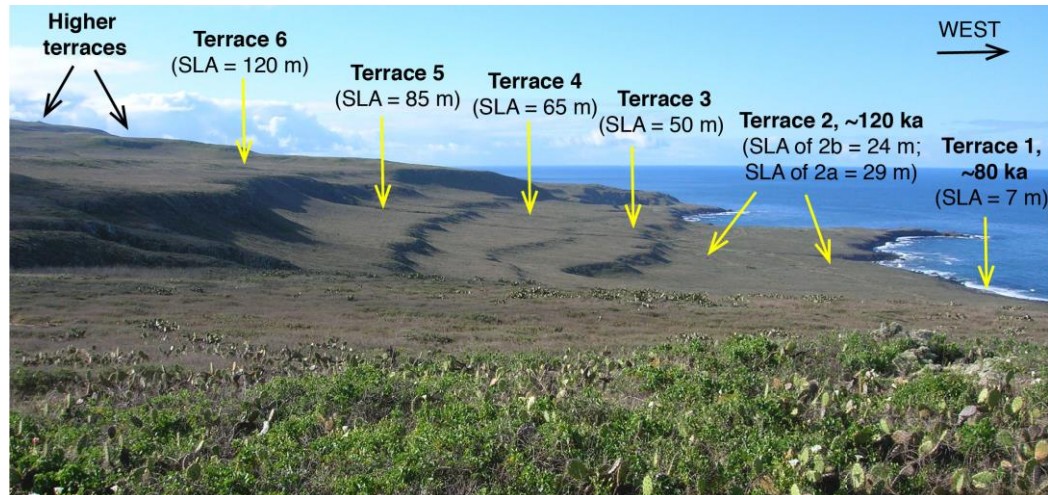

**Figure 37: View of the lowest six marine terraces on the west coast of San Clemente Island, California, looking south. Terrace 2b has been dated to MIS 5e (~120 ka) by TIMS U-series methods on coral (Muhs et al., 2002b); terrace 1 is estimated to be ~80 ka (MIS 5a) based on amino acid ratios in mollusks (Muhs, 1983). Terrace 6 could be as old as ~600 ka or more, if the rate of uplift has been constant over time. Photograph by D.R. Muhs.**

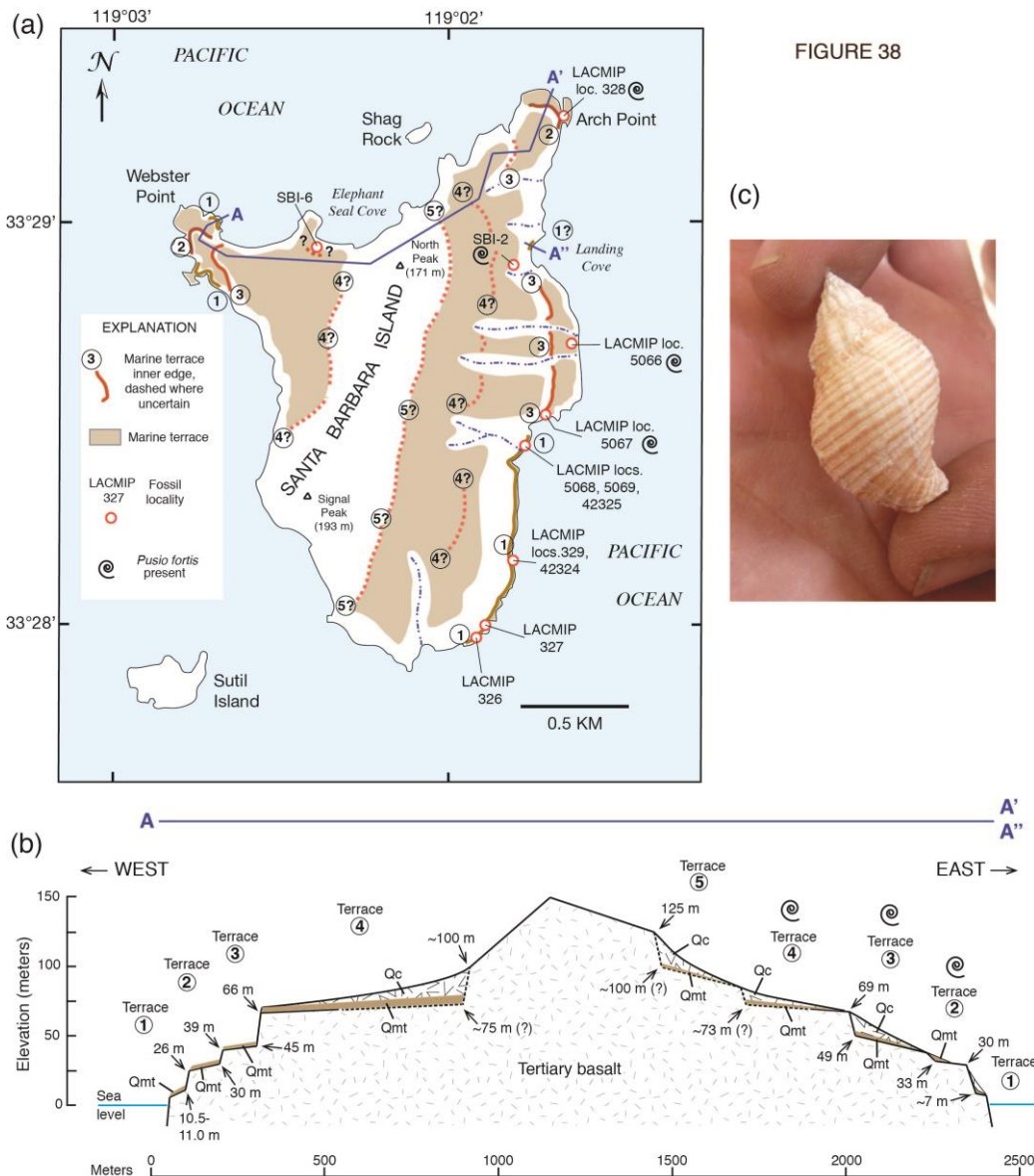

FIGURE 38

Figure 38. Use of an extinct species as a biostratigraphic marker. (a) Map of marine terraces and fossil localities on Santa Barbara Island, California (from Muhs and Groves, 2018), along with (b) topographic profile (location shown in (a)). Also shown in both (a) and (b) are terrace deposits that host the extinct fossil gastropod *Pusio fortis* (formerly *Calicantharus fortis*), shown in (c). Note that in (a) and (b), this taxon is not found on terrace 1, whose deposits contain a mix of fossils dating to MIS 5e and 5c, but is found on higher terraces, indicating that it likely became extinct before MIS 5e. Photograph in (c) is from 69-m-high terrace deposits on San Miguel Island, by D.R. Muhs. ´

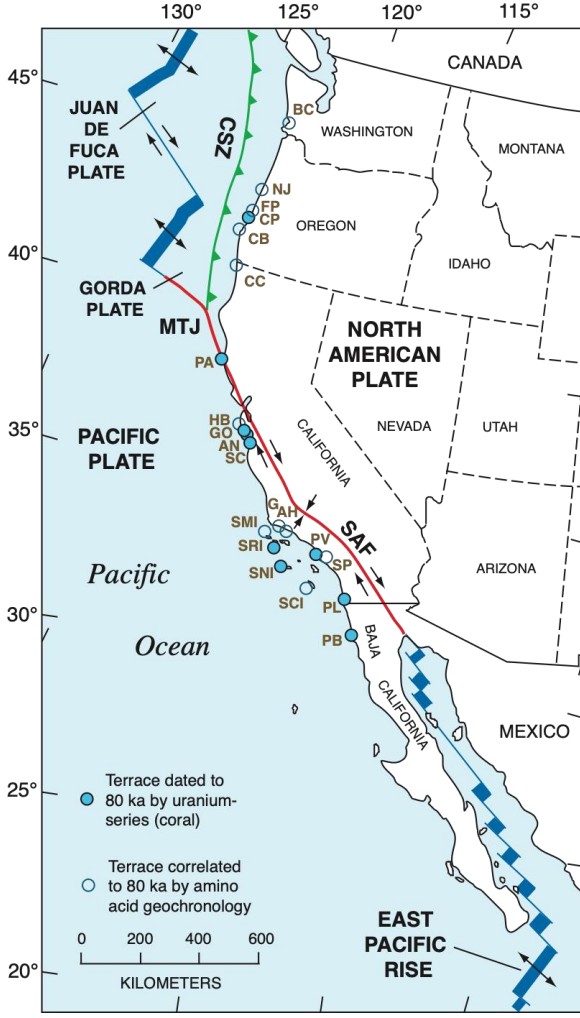

**Figure 39. Map of the Pacific Coast of North America with structural features as shown in Figure 2, but also plotted are localities (filled blue circles) where U-series ages of corals dating to MIS 5a (~80 ka) have been reported and localities (open blue circles) where amino acid ratios in mollusks permit correlation to MIS 5a. Abbreviations are keyed to Table S3 and are as follows: BC, Bay Center, Willapa Bay; NJ, Newport Jetty; FP, Five Mile Point; CP Coquille Point; CB, Cape Blanco; CC, Crescent City; PA, Point Arena; HB, Half Moon Bay; GO, Green Oaks Creek; AN, Point Año Neuvo; SC, Santa Cruz; G, Gaviota; AH, Arroyo Hondo; SMI, San Miguel Island; SRI, Santa Rosa Island; SNI, San Nicolas Island, PV, Palos Verdes Hills; SP, San Pedro; SCI, San Clemente Island; PL, Point Loma; PB, Punta Banda; CSZ, Cascadia Subduction Zone; MTJ, Mendocino Triple Junction; SAF, San Andreas Fault.**



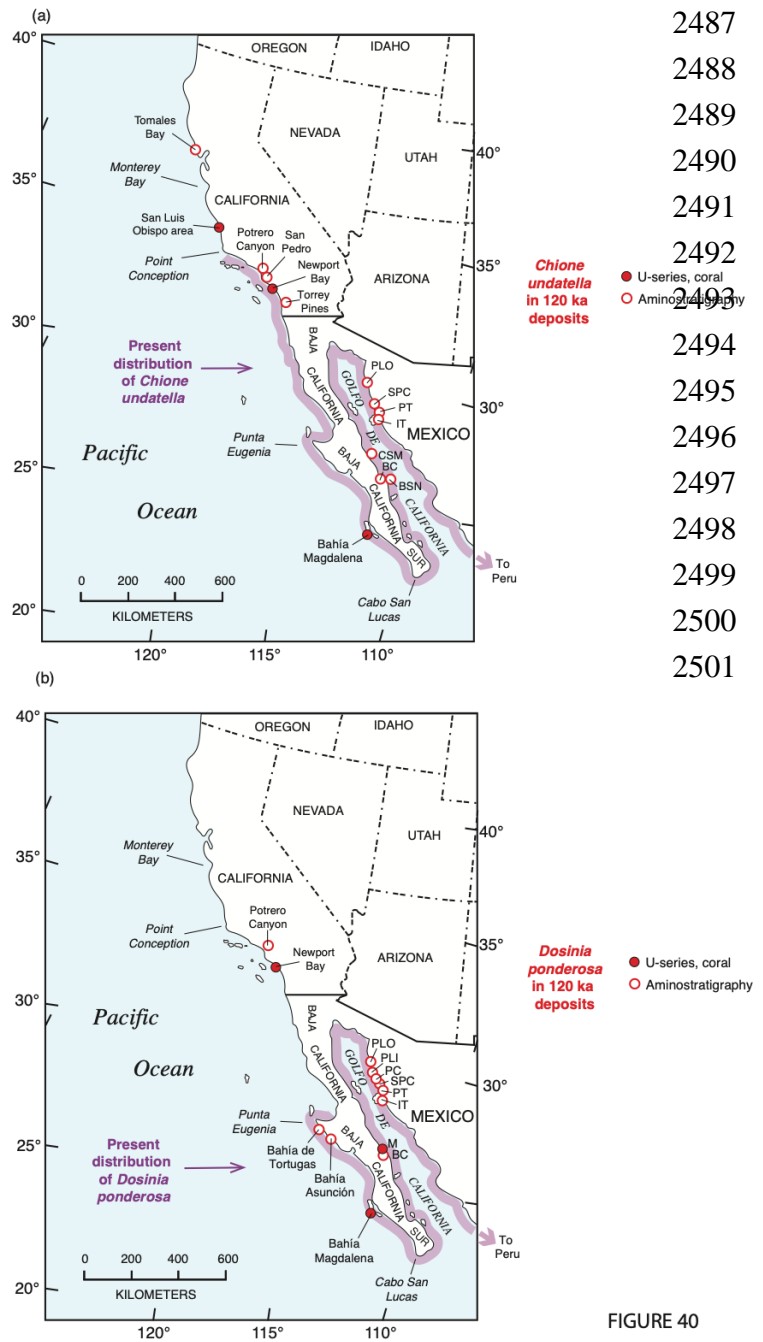

**Figure 40: Examples of extralimital southern**
**species of bivalves indicating sea surface**
**temperatures higher than present during MIS 5e**
**(filled and open circles); also shown in light**
**purple shading are the modern ranges of these**
**taxa (from Coan and Valentich-Scott, 2012): (a)**
***Chione undatella* and (b) *Dosinia ponderosa*. Age**
**and fossil data compiled from sources as follows:**
**Jordan (1936), Valentine (1956, 1960), Kanakoff**
**and Emerson (1959), Johnson (1962), Omura et**
**al. (1979), Emerson (1980), Emerson et al. (1981),**
**Kennedy et al. (1982), Ashby et al. (1987),**
**Keenan et al. (1987), Ortlieb (1987, 1991), Grove**
**et al. (1995), Grant et al. (1999), Kennedy (2000),**
**and Muhs and Groves (2018).**

FIGURE 40



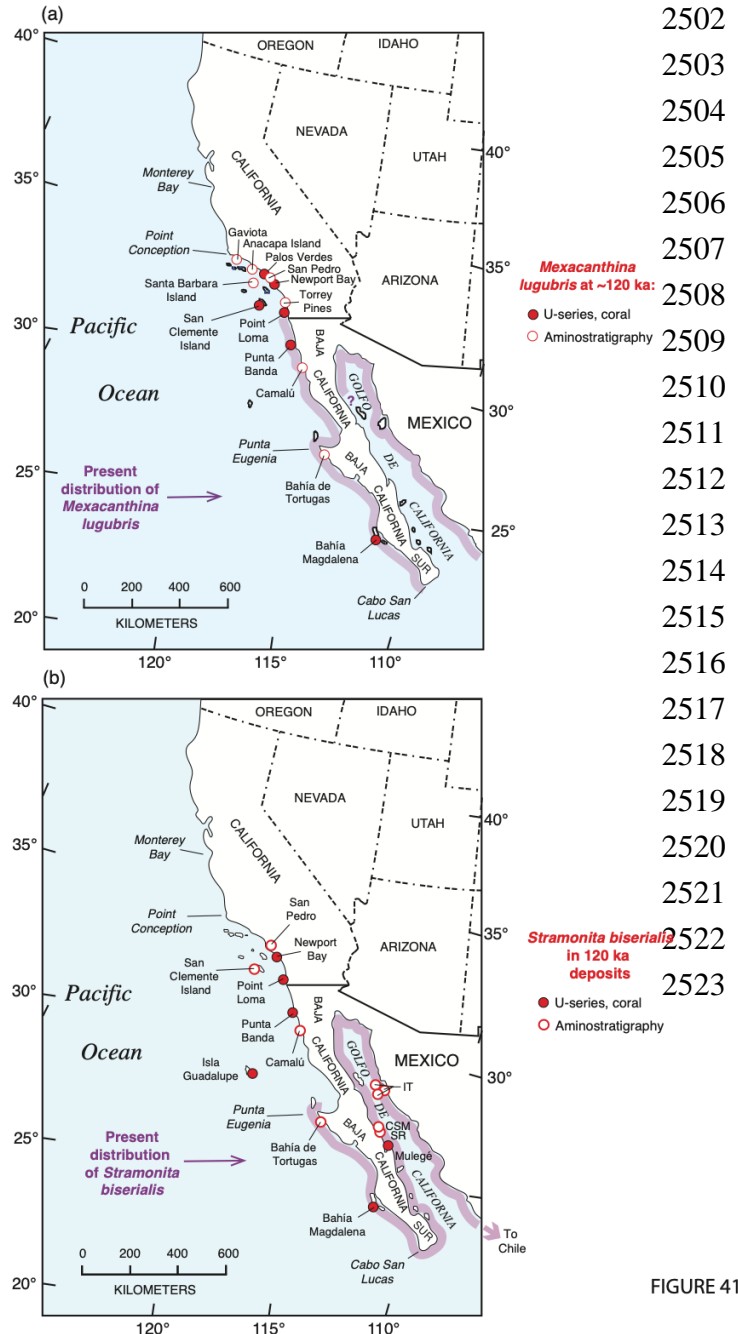

**Figure 41: Examples of extralimital southern**
**species of gastropods indicating sea surface**
**temperatures higher than at present during MIS**
**5e (filled and open circles); also shown in light**
**purple shading are the modern ranges of these**
**taxa (from Bertsch and Aguilar Rosas, 2016 and**
**Keen, 1971): (a)** *Mexacanthina lugubris* **and (b)**
***Stramonita biserialis*. Age and faunal data**
**compiled from Jordan (1936), Kanakoff and**
**Emerson (1959), Valentine (1962, 1980), Lipps et**
**al. (1968), Kern (1977), Ashby et al. (1979), Omura**
**et al. (1979), Lindberg et al. (1980), Emerson et al.**
**(1981), Ortlieb (1987), Rockwell et al. (1989),**
**Muhs et al. (1983, 1992, 2002b, 2014a), Grant et al.**
**(1999), Wehmiller and Pellerito (2015). Note that**
**the presence of** *Stramonita biserialis* **on San**
**Clemente Island is from a recent discovery of this**
**taxon by L.T. Groves, Natural History Museum of**
**Los Angeles County, from the NOTS Pier terrace**
**(Muhs, 1983), correlated to the Eel Point terrace**
**(~120 ka; Muhs et al., 2002b) by amino acid ratios**
**in mollusks.**

FIGURE 41



(a)

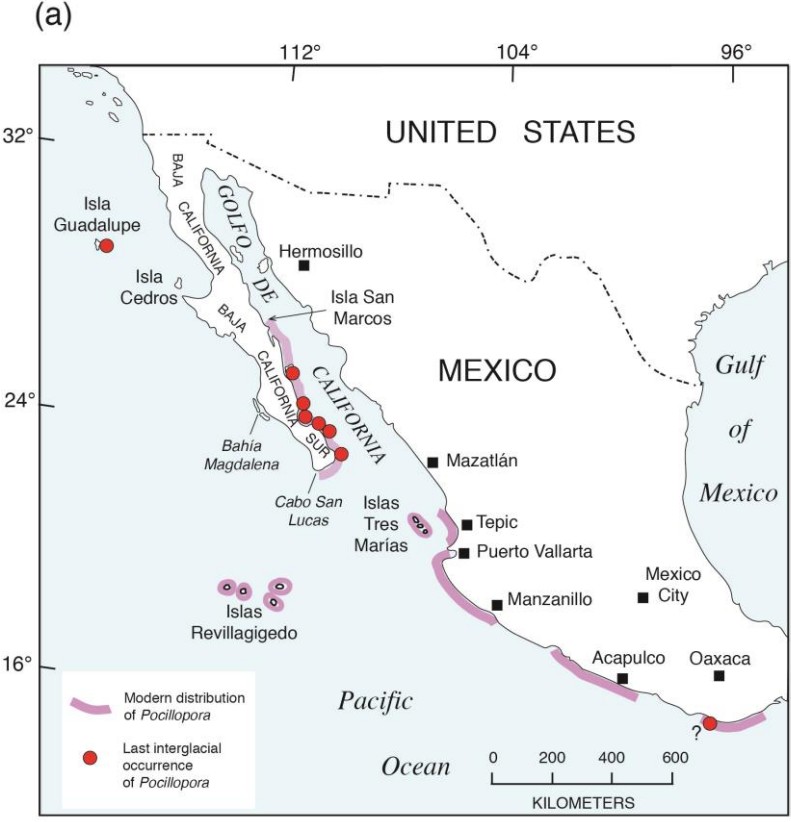

(b)

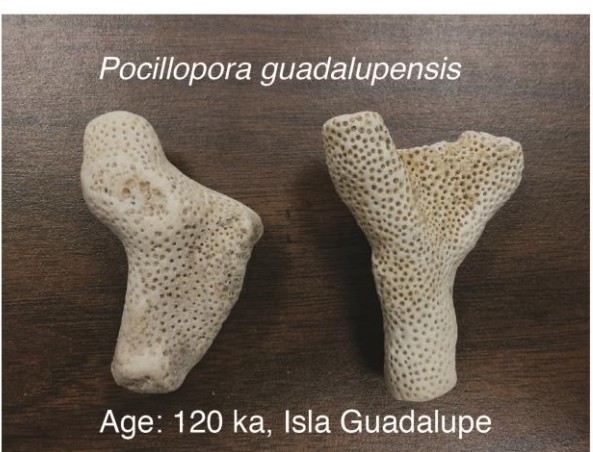

**Figure 42: (a) Map showing the modern distribution of species of the hermatypic coral genus *Pocillopora* (purple shading; from Reyes-Bonilla and López-Pérez, 1998) and U-series-dated, MIS 5e occurrences of this genus elsewhere in the region. MIS 5e age and faunal data from: Durham (1980), Lindberg et al. (1980), Sirkin et al. (1990), Szabo et al. (1990), Muhs et al. (2002b), DeDiego-Forbis et al. (2004), Johnson et al. (2007), and Tierney and Johnson (2012). Occurrence of fossil *Pocillopora* on the Mexican coast south of Oaxaca is undated, but could be of MIS 5e age, and is from Palmer (1928). (b) Photograph of fragments of *Pocillopora guadalupensis* from Isla Guadalupe, Mexico, dated to ~120 ka (Muhs et al., 2002b). Photograph by D.R. Muhs.**

FIGURE 42





**Figure 43: Map of a part of the**
**Pacific Coast of North America**
**showing marine invertebrate faunal**
**zones (from Valentine, 1966) and**
**their correlation with mean annual**
**sea surface temperatures**
**(temperature data from U.S.**
**National Oceanic and Atmospheric**
**Administration).**

Figure 43



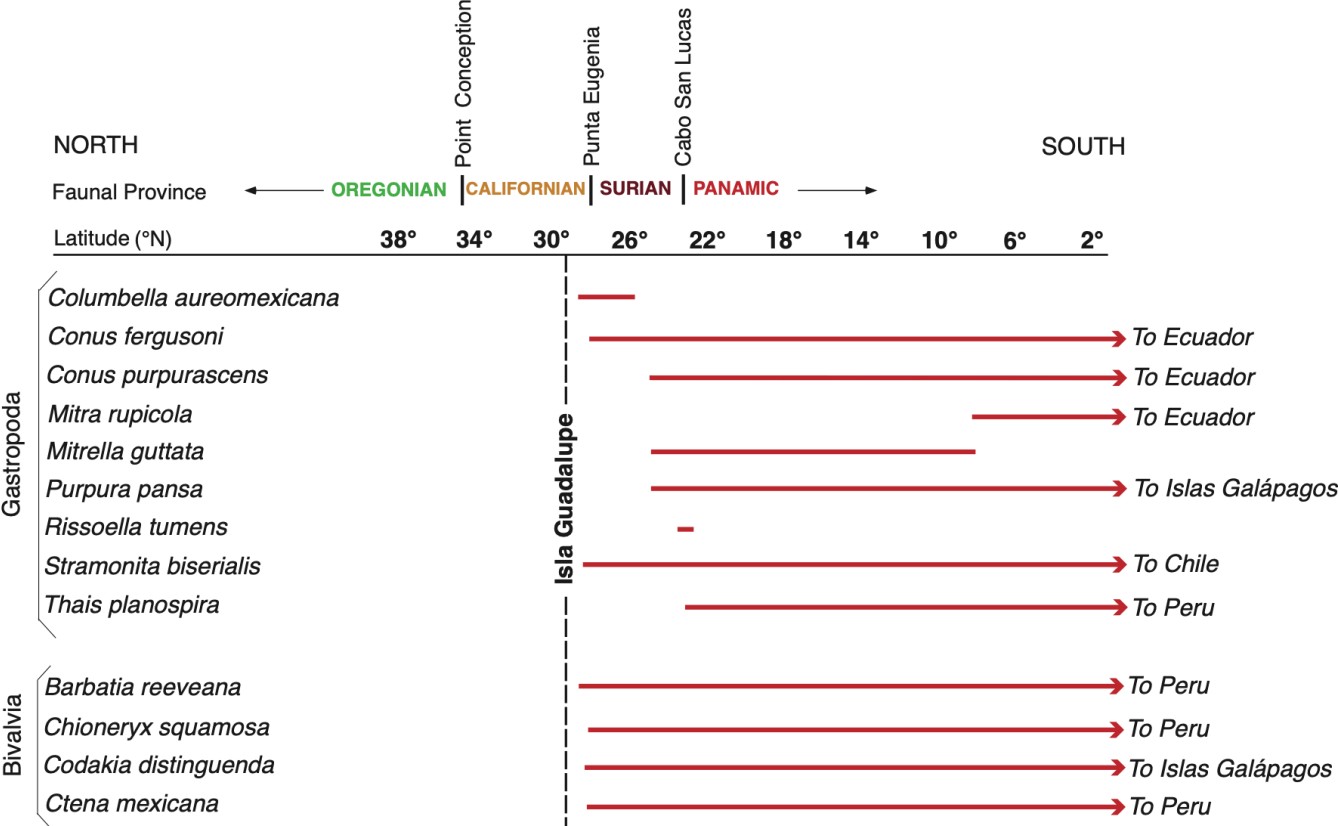

Figure 44: Extralimital southern species of gastropods and bivalves (from Lindberg et al., 1980) in marine deposits of Isla Guadalupe, Mexico (see Fig. 42 for location) dated to MIS 5e by Muhs et al. (2002b), and their modern latitudinal distribution (from Keen, 1971 for gastropods; from Coan and Valentich-Scott, 2012 for bivalves). Note also that the deposits also contain the extralimital southern genus of coral *Pocillopora* (see Fig. 42), and the Indo-Pacific gastropod *Naria cernica* (see text for discussion).

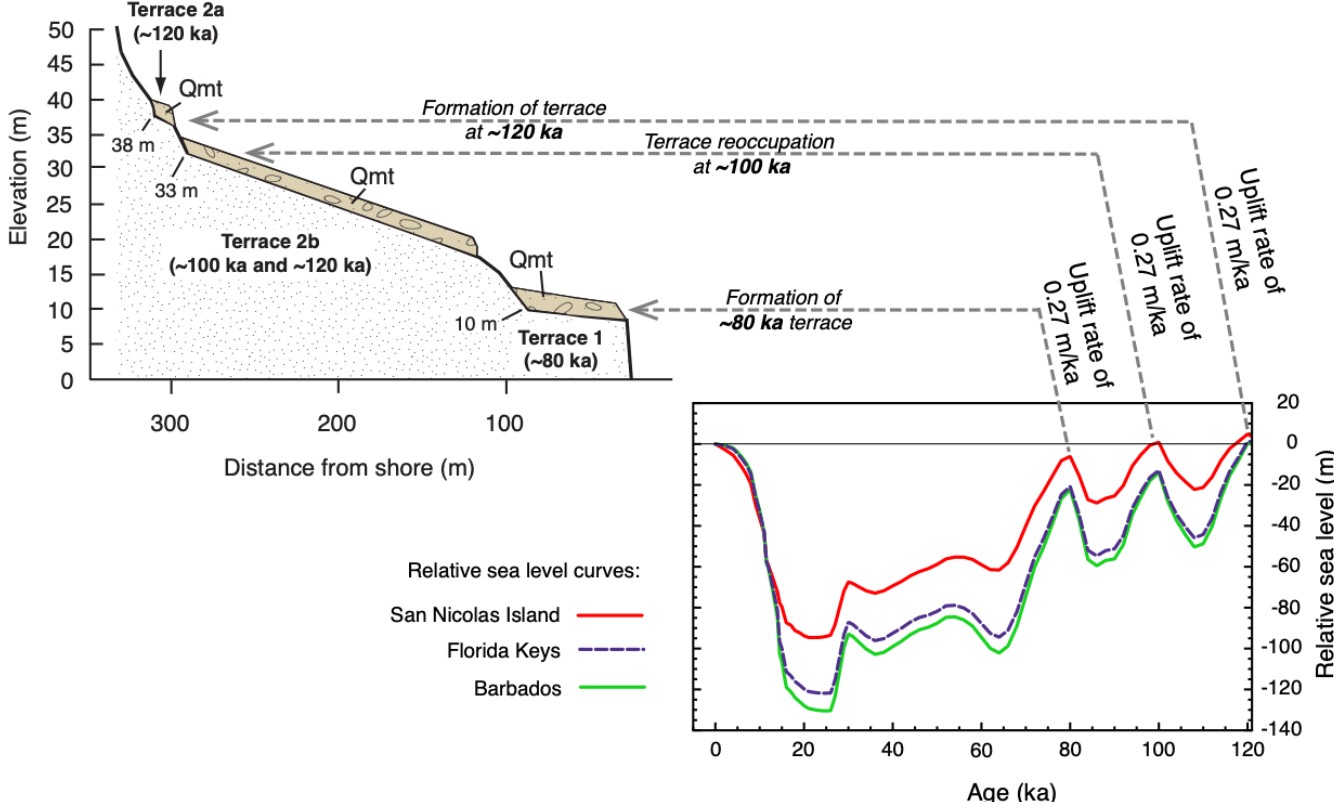

**Figure 45: Cross section of the lowest marine terraces on San Nicolas Island, California and modeled sea level curves for this island, the Florida Keys, and Barbados (from Muhs et al., 2012), showing link between sea level history, uplift rate, and terrace geomorphology, influenced by glacial isostatic adjustment (GIA) processes. Qmt, Quaternary marine terrace deposits.**



| Name of RSL indicator | Description of RWL | Description of IR |
|---|---|---|
| Marine terrace elevation | Junction of the wave-cut platform and sea cliff | Shoreline angles typically are within a meter or less of mean sea level. |
| Pholadidae elevation | Elevation of rock-boring mollusks in growth position | Bivalves in the Pholadidae family typically live within the intertidal zone and usually at depths of ~10 m or less, providing a minimum paleo-sea level elevation |
| Coral reef terrace elevation | Uppermost growth-position coral reef colony is a minimum elevation; habitat depth measurement needs to be added for better accuracy | In southern Baja California Sur and adjacent parts of mainland Mexico and Central America, colonial hermatypic corals in growth position (e.g., *Porites* and *Pocillopora*) typically live in the intertidal zone and are usually found at depths of ~10 m or less, providing a minimum paleo-sea level elevation |

**Table 1: different types od RSL indicators, reference water level (RWL) and indicative range (IR) on the Pacific**

**Coast of North America.**



| Measurement technique | Description | Typical accuracy |
|---|---|---|
| Hand level and tape | Hand level and metered tape and/or use of transit and stadia | On the order of tens of centimeters |
| Aneroid altimeter | American Paulin System aneroid altimeter | On the order of 0.5 meter |
| Global Positioning System (GPS) | Uses satellite array and triangulation with post-processing to increase accuracy | On the order of tens of centimeters if satellite geometry is favorable |

**Table 2: measurement techniques used to establish the elevation of MIS 5e shorelines on the Pacific Coast of North**
**America.**