# Peer review of "MIS 5e sea-level history along the Pacific Coast of North America"

_Earth System Science Data, 2021_

## Author Comment (AC1)

**REPLIES BY THE AUTHOR IN BOLD, BLUE FONT**

**RC1**: 'Comment on essd-2021-345', Jessica Creveling, 01 Nov 2021
In the ESSD submission "MIS 5e sea-level history along the Pacific Coast of North America", author Daniel Muhs presents a thoroughly researched and illustrated compendium of Last Interglacial sea level markers for the North and Central America Pacific coast  from southern Canada to central America. This is a valuable contribution across multiple themes, including:

- A systematic, north-to-south guide to MIS 5e (or purported 5e) sedimentary deposits and marine terraces (erosional and constructional) from British Columbia, Canada to the Osa Peninsula, Costa Rica with a rich discussion of the primary descriptions of these locales.

- A description of relevant geochronological methods used to date these indicators, with a broad bibliography for each method.

- A more concise, but equally informative summary of older and younger terraces situated above and below purported/dated MIS 5e indicators, with implications for local uplift histories.

- A nuanced discussion of how the MIS 5 (sensu lato) paleozoogeography informs regional climate constructions and bears on questions of a warming world.

- And a thoughtful summary of remaining controversies and how these can direct future inquiry across multiple methods, from GIA to geochronology to field geomorphology.

I have read the submission twice in full and learned from both readings. Muhs' writing and illustrations are exceptionally clear. I found nothing to quibble with and I believe that any reader, regardless of expertise or experience, will gain from this remarkable synthesis. I advocate for the publication of this submission without any change from its current format.

Best regards,

Jessica Creveling
**Citation**: https://doi.org/10.5194/essd-2021-345-RC1

**I thank Jessica for reading the paper twice and for her kind comments on it.**

**--Daniel R. Muhs**

---

## Author Comment (AC2)

**REPLIES BY THE AUTHOR IN BOLD, BLUE FONT**

**RC2**: 'Comment on essd-2021-345', John Wehmiller, 11 Nov 2021

J. F. Wehmiller review of Muhs: **MIS 5e sea-level history along the Pacific Coast of North America**

**Review begun Oct. 25, 2021**

**Comments are inserted at appropriate positions in the guidelines below; reviewer comments are in italics.**

**Review guidelines – step by step**

For future reuse and reinterpretation, it is mandatory for the user to be assured about research data quality. It is the aim of ESSD to provide quality assessment for data sets which are already included in permanent repositories.

Thus, when reviewing a paper in ESSD, we would like you to review not just the manuscript but, more importantly, the data set itself. For your guidance, a step-by-step review approach is suggested:

**1. Read the manuscript:** are the data and methods presented new? Is there any potential of the data being useful in the future? Are methods and materials described in sufficient detail? Are any references/citations to other data sets or articles missing or inappropriate?

Is the article itself appropriate to support the publication of a data set?

*Wehmiller (italics used here and following for comments):*

*This is an extremely comprehensive manuscript, both in terms of text and the data set(s) that is summarizes. Aside from some minor editorial comments in the text (in the form of pdf "sticky notes") I have no recommendations for changes. It is very noteworthy that, in spite of the wealth of information presented by Muhs, there are so many places along the Pacific where the data simply do not permit an "absolute" assignment of a "last interglacial age" to a specific terrace. The quthor does an excellent job of summarizing the state of our knowledge and also the uncertainties about the age assignments. Muhs has also done an excellent job of reviewing the history of marine terrace studies along the entire coastline – a remarkable accomplishment, perhaps more than would be expected for an atlas focused on MIS 5e.* **Thank you for your kind comments!**

*I note, however, that I have reviewed the "WALIS spreadsheet" that is associated with the manuscript and have made many comments about entries in this spreadsheet. My version of the spreadsheet is attached. More on this in following sections. This will likely require some discussions between the authors and WALIS colleagues.*

**Special issue editor Alessio Rovere has followed all of your suggested changes for the WALIS spreadsheet and uploaded a revised version. Thank you for suggesting the changes that have improved this.**

**2. Check the data quality:** is the data set accessible via the given identifier? Is the data set complete? Are error estimates and sources of errors given (and discussed in the article)? Are the accuracy, calibration, processing, etc. state of the art? Are common standards used for comparison?

Is the data set significant – unique, useful, and complete?

*The data tables created by Muhs (S1, S2, S3) are accurate and consistent with the manuscript. The "WALIS spreadsheet, however, is inaccurate or confusing in many places. It is an attempt to "force" a lot of old data into a set of expectations that can only be met with great difficulty, given the methods available at the time of sample collection and analysis. There are entries in this spreadsheet that are best left to the original publications, and I question the value of entering some methodological data for certain samples but not entering comparable data for other samples. For consistency, it should be all or none. I have made numerous comments in the AAR section of the spreadsheet, as I personally was involved in analyzing many of the samples listed in that spreadsheet. Consequently, I know better than anyone else the subtleties of the method(s) and the associated results.*

**Same reply as above: special issue editor Alessio Rovere has followed all of your suggested changes for the WALIS spreadsheet and uploaded a revised version. Thank you for suggesting the changes that have improved this.**

**3. Consider article and data set:** are there any inconsistencies within these, implausible assertions or data, or noticeable problems which would suggest the data are erroneous (or worse). If possible, apply tests (e.g. statistics). Unusual formats or other circumstances which impede such tests in your discipline may raise suspicion.

Is the data set itself of high quality?

*I have reviewed S2 and was actually involved in multiple discussions with the author prior to the preparation of S2. It is a fair and accurate summary of a large amount of data, and it appears to be consistent with the manuscript itself. Both are organized in a consistent geographic manner that is easy to follow.*

*The "WALIS spreadsheet" for the AAR data has many entries that are inaccurate or misleading. Comments are inserted where needed, but there may be other revisions when the author and WALIS colleagues review my comments.*

**See replies above.**

**4. Check the presentation quality:** is the data set usable in its current format and

size? Are the formal metadata appropriate? Check the publication: is the length of the article appropriate? Is the overall structure of the article well structured and clear? Is the language consistent and precise? Are mathematical formulae, symbols, abbreviations, and units correctly defined and used? Are figures and tables correct and of high quality?

*The article is definitely well structured and easy to follow, as it proceeds from north to south; tables S1-S3 follow this structure as well. The manuscript is long, primarily because it has to discuss ALL the possible MIS 5 sites, even though many of them are only qualitatively data (or not dated at all). This need arises because so many sites are of the potential MIS 5e age, but their ages cannot be constrained to the precision expected in the WALIS project.*

**Thank you.**

Is the data set publication, as submitted, of high quality?

*The dataset as presented in S1, S2, and S3 is clear, well explained and of high quality. It is easily citable in the current format. The data in the WALIS spreadsheet (S2) is potentially confusing, inconsistent or ambiguous – it attempts to include values for all D/L results from every sample, with I recommend not even trying to include all the AAR data in this spreadsheet and simply refer readers to the original manuscripts. In some cases the original manuscript(s) are referred to, but not in others. This inconstancy would confuse potential users. The authors of those manuscripts would presumably have the insights to report what they feel are the most reliable results.*

**See replies above.**

**Finally:** By reading the article and downloading the data set, would you be able to understand and (re-)use the data set in the future?

*The data in S1, S2, and S3 are useful. The AAR data in the "WALIS spreadsheet" are not useful and any user would have to resort the primary literature.*

**See replies above.**

**Rating**

Reviewers are asked to decide how well the respective data sets presented by an article and the article itself meet the following criteria (rated 1–4, excellent–poor):

**Significance**

Is there any potential of the data being useful? This is clearly the most important decision. There are at least three sub-criteria to evaluate:

• **Uniqueness:** it should not be possible to replicate the experiment or observation on a routine basis. Thus, any data set on a variable supposed or suspected to reflect changes in the Earth system deserves to be considered unique. This is also the case for cost-intensive data sets which will not be replicated due to financial reasons. A new or improved method should not be trivial or obvious.

• **Usefulness:** it should be plausible that the data, alone or in combination with other data sets, can be used in future interpretations, for the comparison to model output or to verify other experiments or observations. Other possible uses mentioned by the authors will be considered.

• Completeness: a data set or collection must not be split intentionally, for example, to increase the possible number of publications. It should contain all data that can be reviewed without unnecessary increase of workload and can be reused in another context by a reader.

*I rate the data quality as #1 for each of the above criteria – data as presented in S1, S2, and S3. The WALIS spreadsheet (attached) is ranked as #3 or #4... it needs a lot of work.*

**Thank you for the kind comments on S1, S2, and S3; for the WALIS spreadsheet, please see replies above.**

**Data quality**

The data must be presented readily and accessible for inspection and analysis to make the reviewer's task possible. Even if a data set submitted is the first ever published (on a parameter, in a region, etc.), its claimed accuracy, the instrumentation employed, and methods of processing should reflect the "state of the art" or "best practices". Considering all conditions and influences presented in the article, these claims and factors must be mutually consistent. The reviewer will then apply his or her expert knowledge and operational experience in the specific field to perform tests (e.g. statistical tests) and cast judgement on whether the claimed findings and its factors – individually and as a whole – are plausible and do not contain detectable faults.

*As noted in the "WALIS spreadsheet" and elsewhere, I have personally been involved in the analysis of many of the AAR samples reported in S2 and the WALIS*

*spreadsheet. Therefore, I am quite familiar with the data and the form in which it is presented. I conferred with the author regarding the production his data table S2. I feel that the WALIS spreadsheet needs a large amount of work to correct errors, improves references, and make the entries internally consistent. I am willing to consult with the author(s) as needed.*

**See replies above.**

**Presentation quality**

Long articles are not expected. Regarding the style, the aim is to develop stereotypical wording so that unambiguous meaning can be expressed and understood without much effort. The article should express clearly what has been found, where, when, and how. The article text and references should contain all information necessary to evaluate all claims about the data set or collection, whether the claims are explicitly written down in the article, or implicit, through the data being published or their metadata. The authors should point to suitable software or services for simple visualization and analysis, keeping in mind that neither the reviewer nor the casual "reader" will install or pay for it.

*The manuscript is indeed long, but its length is necessary because of the long history of the work and the comprehensive nature of the discussion and review of the literature. The author has been the major player in the past three decades of Pacific coast geochronology and draws upon his lengthy involvement in this work.*

**Thank you.**

**Access review, peer review, and interactive public discussion (ESSDD)**

Manuscripts submitted to ESSD at first undergo a rapid access review by the topical editor (initial manuscript evaluation), which is not meant to be a full scientific review but to identify and sort out manuscripts with obvious major deficiencies in view of the above principal evaluation criteria.

If they are not immediately rejected, they will be posted on the Earth System Science Data Discussions (ESSDD) website, the discussion forum of ESSD, where they are subject to full

**peer review and interactive public discussion.**

**Peer-review completion (ESSD)**

At the end of the interactive public discussion, the authors may make their final response and submit a revised manuscript. Based on the referee comments, other relevant comments, and the authors' response in the public discussion, the revised

manuscript is re- evaluated and rated by the topical editor. If rated excellent or good in all of the principal criteria and specific aspects listed above, it will normally be accepted for publication in ESSD. Additional advice from the referees in the evaluation and rating of the revised manuscript will be requested by the topical editor if the public discussion in ESSDD is not sufficiently conclusive.

*This reviewer rates the overall manuscript as excellent.*

**Thank you.**

**CC2**: ['Reply on RC2'](), Alessio Rovere, 25 Nov 2021
As topical editor of this Special Issue and the leading scientist behind the WALIS database idea, I would like to thank John Wehmiller for the specific comments on the database, mostly centered on the Amino Acid Racemization data table.

For the readers, I wish to clarify that the main aim of having the data in a standardized format is not to "supersede" original publications. Instead, it is to make the data contained therein more usable and available in a ready-to-analyze form. However, as John Wehmiller points out in his comment, nothing can indeed substitute the expertise of the person/team who originally collected the data, which should always be credited and to whom questions may be directed in case of doubts.

However, we should recognize that good data most often outlive our careers. Also, it is becoming increasingly clear that we cannot rely on publishers (that are, most often, companies that may eventually go out of business) to store data in the long term. Therefore, we need to prepare the best possible standardized templates and keep them in long-term archives. This is the driving concept of the entire WALIS special issue (and the associated database). Of note is that both data and papers are archived for long-term preservation. Data is stored in Zenodo, copies of papers are sent to the German National Library and other repositories, all independent from the publisher.

I would also like to point out that WALIS is not a "fixed" atlas, but it can be updated and improved through suggestions from the scientific community. The Special Issue serves to collect data for Version 1.0, to which hopefully many more will follow. The comments received from John Wehmiller on the AAR spreadsheet are exactly what WALIS needs to grow and improve through time. He knows better than anyone else those data, as he was the one who did the original analyses and wrote the original manuscripts from where the data was reported.

I encourage anyone who has their work cited in WALIS to look over the data they produced and get in touch with the editorial team if they notice discrepancies or suggest corrections. A track-changes system is in place in WALIS so that creating a new

record without losing the original entry is possible, and credit to the "reviewer" will be given. John Wehmiller has set an excellent example of how to do that.
**Citation**: https://doi.org/10.5194/essd-2021-345-CC2

**RC3**: 'Reply on CC2', John Wehmiller, 29 Nov 2021
Related to this discussion, it should be pointed out that the AAR community initiated an effort in 2010 to establish a program for archival data storage.  The results of this effort can be found at the US NOAA NEIC Paleoclimatology site (link below).   Datafiles on the site may not have some specific information that would be sought by WALIS users but the overall structure and content of the site should be of interest to the WALIS community.  Ideally, links between the NOAA site or other databases could be established so that WALIS community is aware of these other sources of information.  Several publications and datasets discussed in the Muhs WALIS paper are listed in the NOAA site, and others remain to be added.

In the 50+-year history of the AAR method, as with many other methods, technological advances have affected the type and quality of data.  In the early years, only one amino acid D/L value could be measured; newer methods allowed multiple D/L measurements with varying degrees of confidence. Data digitization and more rapid and more sensitive analytical procedures further enhanced the method.  Comparing D/L values obtained in the past two decades to those from the 1960's and 1970's must be done with extreme care, as inter-lab (and even intra-lab) differences can be significant.

For those studies that report D/L values for multiple amino acids, it is important to note that, in almost all cases, the investigator(s) emphasized results for certain amino acids.  The reasons for relying on specific amino acid results are usually discussed in the primary literature.  It is important that the WALIS database include this information if it is to be consistent with the original work.

NOAA Paleoclimatology Data site

https://www.ncei.noaa.gov/products/paleoclimatology

**Search for "Paleodata", then enter "Racemization"**

**Citation**: https://doi.org/10.5194/essd-2021-345-RC3

"STICKY NOTE" COMMENTS BY JOHN WEHMILLER:

**Page: 30**

Author: WEHMILLER Subject: Sticky Note Date: 11/1/2021 3:45:53 PM -04'00'
The Woods and Camalu data are cited in the WALIS spreadsheet and/or in Wehmiller & Pellerito 2015

**Thank you for catching this. I now cite Wehmiller and Pellerito (2015) here as well.**

**Page: 36**

Author: WEHMILLER Subject: Sticky Note Date: 11/1/2021 3:50:38 PM -04'00' do you need to change the name?

**This is actually an amusing situation. The original genus name was *Tegula*. For a brief period, this was changed to *Chlorostoma*, but then changed back again to *Tegula*! So, the "old" name is now the "new" name again.**

**Page: 37**

Author: WEHMILLER Subject: Sticky Note Date: 11/2/2021 2:48:39 PM -04'00' Figure 36 shows the SNI data but text refers to Figure 37 - better check figure numbering

**Thanks—fixed this.**

Author: WEHMILLER Subject: Sticky Note Date: 11/1/2021 4:20:48 PM -04'00' fig 36b?

**Thanks—fixed this too.**

Author: WEHMILLER Subject: Sticky Note Date: 11/2/2021 2:49:59 PM -04'00'
I'm not a fan of the parabolic model, especially for the A/I values for the high terraces that are essentially at equilibrium. However, I agree that these model ages are simply that - model ages. We have similar estimates in the 1977 Open File and 1993 SEPM publications for the terraces in the 100-200 m range using different modeling approaches.

**Agreed: this is why I put in the qualifying sentence at the end.**

Author: WEHMILLER Subject: Sticky Note Date: 11/1/2021 4:29:40 PM -04'00' We have AAR on the Linda Vista terrace

**Thank you for catching this, which I overlooked. I now mention here that your data in Wehmiller et al. (1977a) show that the Lindavista terrace has D/L ratios near equilibrium in *Tivela*, indicating considerable antiquity (which is consistent with the new cosmogenic ages).**

**Page: 39**

Author: WEHMILLER Subject: Sticky Note Date: 10/26/2021 10:22:44 AM -04'00' post-dated (i.e., younger than)

**Oh my goodness... a huge thank-you for catching THIS: fixed...!!!**

**Page: 40**

Author: WEHMILLER Subject: Sticky Note Date: 11/1/2021 4:35:26 PM -04'00'
As I recall, the AA data for Saxidomus and Mya from Coquille and Cape Blanco are not inter-generically consistent. You can't go into this issue here, but it's one of the nasty unresolved issues.

**Yes, I agree.  I wish we had more data on *Mya* from dated localities, which might clear some of this up, but it's a less-common beast.**

---

## Author Comment (AC3)

**REPLIES BY THE AUTHOR IN BOLD, BLUE FONT**

**CC1**: 'Comment on essd-2021-345', Barbara Mauz, 22 Nov 2021
I truly enjoyed reading this well-written paper. It exemplifies how to review a subject and how to link early ideas to today's thinking. Notwithstanding, after around half of the text I was puzzled.

**Thank you for your kind comments and I am glad you enjoyed reading it.  I am sorry to hear that halfway through it you were puzzled.**

With the Walis project aims in mind, i.e. standardising model-independent approaches for determining sea-level index points: what exactly is the RSL indicator here? Is it a marine terrace, a coral terrace, a shoreline angle?

**Depending on where you are, geographically, the RSL indicator is different things. Over most of the Pacific Coast of North America, from southern Canada almost to southern Baja California, the RSL is the shoreline angle of an *erosional* marine terrace.  South of there, where *constructional* coral reef terraces can be found (although rarely), it is more complex.  In some places, there are erosional marine terraces (i.e., wave-cut benches, with shoreline angles), but coral reefs may have grown ON the wave-cut terrace.  Thus, in those situations, you may have two RSL indicators, the shoreline angle AND the paleo-sea level that would be implied by the depth range of the taxa within the coral reef.  But, even there, the shoreline angle is the better one, as it is very close to mean sea level, as discussed in the text.**

Ok, the angle is a clear concept and easy to identify in the field (unless it is covered by slope deposits as depicted in Fig. 3 ) but how is the spatial relationship defined between the fossil dated, the terrace surface and the shoreline angle? In Fig. 3 a "*simple*" and a "*complex*" case of a marine terrace is depicted and the terrace deposit is composed of pebbles and (transported?) molluscs fossils. The complex case would be the one where an additional terrace forms above the lower one, hence the subsiding coast is the complex example. What about uplifting coasts and reef platforms? None of the U-series dated corals was in primary growth position suggesting that the sample was collected from the subtidal reef slope or interior platform.

**I'm glad that you understand the concept of a shoreline angle, because that is crucial to much of this review.  I believe I have explained the concept sufficiently in the text and there are diagrams and photographs showing both modern and ancient ones.**

**Fig. 3b was intended to show how two marine terraces could be masked by overlying deposits to appear as if only one terrace is present.  The lower terrace apparent only in cross section is younger than the upper terrace; thus, this is not (as you describe) a subsiding coast, but an uplifting coast.  I have modified the figure caption a bit to make this clear.**

**As for the fossils, yes, on most the Pacific Coast of North America, north of southern Baja California anyway, virtually all mollusks and corals are transported.  Hardly anything is ever in growth position on this high-energy coast.  This is very similar to what I have observed in Spain (Canary Islands and Mallorca) and Italy (Sardinia), so I am surprised you have not observed this yourself with your experience in the Mediterranean.  Do these transported fossils have *exactly* the same age as the wave-cut bench and its associated shoreline angle?  Clearly, they cannot be *exactly* the same age, because the bench has to have formed first (at least initially) and anything on top of it (sand, gravel, fossils) has to have been deposited there later.  Having said that, the bench and the fossils are certainly close in age, and any difference in age is likely not capable of being resolved with the dating methods we currently have in use.  We see this in the age of beach-collected corals on the modern Pacific Coast, where these specimens yield U-series ages of only a few hundred years, indicating a time of growth similar to the age of the modern wave-cut bench, which is still forming.**

**You mention "subtidal reef slope" with regard to these corals: that does not apply here.  These are tiny, solitary, ahermatypic corals (see Fig. 8a); they do not form reefs.  On the open (outer) Pacific Coast, you have to get as far south as southernmost Baja California (see Fig. 7a) before you find large colonial corals, and only from there south do you find hermatypic corals that build reefs.**

Fig. 6 illustrates the method approach using photos from the LACMIP locality – how does this approach relate to WALIS? How do pholads allow to estimate max shoreline angle elevation? I do not find the terms 'bench' and 'pholads' in Rovere et al.

**This is all explained in the text.  "Bench" (in this context meaning *wave-cut* bench) has been a geomorphic term in use for decades and is well understood by geomorphologists everywhere.  "Wave-cut bench" appears in the AGI "Glossary of Geology" (5th edition, 2011) on page 720.  "Pholad" refers to bivalve mollusks of the family Pholadidae, which are the rock-boring mollusks.  This is one of the very few examples of fossils that can sometimes be found in growth position, because they are still found in the holes that they themselves bored in the bench.  Some of these taxa have specific depth ranges, such as *Penitella penita*, the example I used.  It is typically found in waters less than 10 m deep.  Thus, if**

**you find it in growth position in fossil form, you are no lower than 10 m below the former shoreline associated with that fossil.**

**I can't answer the question about why "bench" and "pholad" do not appear in Rovere et al., but the terms have been around for decades and are well established.**

Does this affect the data presented in the WALIS database? In record ID3832 (arbitrarily selected) the indicator is 'marine terrace' (sensu Rovere, I guess), the coral used for dating (Porites panamensis) was found in 3.1 m elevation (Table S1; no uncertainty), the shoreline angle is at 8.7±1.6 m (this is a 18% error; the elevation measurement technique is given as 'not reported'), RWL is -0.03 m and IR is 1.06 m and, logically, WALIS calculates the sea level to have been at 8.73±1.68 m.

**Well, you cannot really calculate a paleo-sea level for the last interglacial period (MIS 5e) (as you have done above) anywhere on the Pacific Coast of North America because you cannot assume tectonic stability anywhere. I tried to emphasize this throughout the course of the manuscript. Most parts of the coast are uplifting (based on observations of multiple terraces at a range of elevations) and to calculate a paleo-sea level for MIS 5e, you would have to know the uplift rate. How would you do that? In most places here and elsewhere, uplift rates are calculated MIS 5e terrace elevations (on the uplifting coast) and an assumed paleo-sea level from distant, tectonically stable coastlines. Otherwise, you are in the middle of an exercise in circular reasoning.**

According to the IUCN database Porites panamensis occurs on coral reef communities growing on rocky substrates, at depth ranging 0 - 36 m. Following Hallmann et al. who followed Hibbert et al. 2016, 2018, this database (together with OBIS) is regarded as being the standard for coral-based SLIPs in WALIS – am I wrong? Daniel Muhs indicates 0 - 10 m for all corals in the study area following Glynn and Ault (2000) who focused on coral life history and population dynamics since the closure of the central-American seaway. There is no explanation as to how one of the key parameter in WALIS, that is the RWL, was inferred or calculated and it looks as if the tidal range, albeit minor, was not taken into consideration at all.

**With all due respect to Hibbert et al. (2016, 2018), both excellent papers, I suspect that the authors themselves have not done any depth measurements of *Porites panamensis* along the Pacific Coast of North America. Thus, I have gone with the depth range given by Peter W. Glynn, who is probably one of the leading field-based authorities on eastern Pacific corals. He has been studying corals in this region for 50 years. If you look at Glynn and Ault's (2000) paper that I referenced,**

**their information on corals in this region was taken from many observations, over many years, by many researchers, and they indicate (see caption for their Figure 2) that the maximum shelf depth where corals (including *P. panamensis*) mainly occur in both the Gulf of California and mainland Mexico is about 10 meters. Recent collections of this species by marine biologists have confirmed this general depth range (see Zapata & Lozano-Cortés, 2015; *Marine Biodiversity Records* 8, 1-4; Cabral-Tena et al., 2013, *Marine Ecology Progress Series* 476, 1-8). This is also consistent with what Verrill described as its depth when he first identified and described this species (from Baja California) in 1870 (Verrill, 1870, *American Journal of Science*). Thus, I regard Glynn and Ault's (2000) statement as authoritative.**

No doubt, ocean currents, ENSO cycles, the virtual absence of extended rocky shelfs and the geological history of the north-central American coast, all together control the shape of marine terraces, reef assemblages, reef construction, growth and shape. I feel that 'marine terrace' does not describe the sea-level indicator(s) that occur on the north-central American coast.

**Well, it would be hard to disagree with the first sentence here that all those variables influence marine terraces and reefs on the Pacific Coast of North America.**

**As for the second sentence, with regard to the term "marine terrace," I guess we just have to agree to disagree. I think that "marine terrace," a term that has been in common use around the world, starting in the 19th century and continuing into the 20th and 21st centuries, is well-established and is understood by geomorphologists.**

Lastly, I feel the paper would benefit if Figs 3, 18, 19, 21, 28, 36a include a key and/or a scale and if standards for numbers are followed: value and corresponding uncertainty must have the same number of digits.

**Why would Figure 3 need a "key"? Everything is pointed to in the figure itself (with arrows) or in the caption text. Same goes for the other figures noted...you just have to look closely, as some of these are within the boundaries of the map itself (for example, on the map of San Nicolas Island).**

**Citation**: https://doi.org/10.5194/essd-2021-345-CC1

---

## Author Comment (AC4)

**REPLIES BY THE AUTHOR IN BOLD, BLUE FONT**

**RC4**: 'Comment on essd-2021-345', Hayley C. Cawthra, 30 Nov 2021
It has been a great pleasure to review Daniel Muhs' review paper 'MIS 5e sea-level history along the Pacific Coast of North America'. This is a long and complex geographic area to cover, with a lengthy history of investigations of its Quaternary deposits, and the author has described the records in a carefully thought out and clear way, with excellent figures to supplement the text. I felt that the accompanying database of sea-level records for the WALIS compilation was well referred to and it cross references well.

**Thank you very much for these kind comments.**

In addition to the MIS 5e deposits and their relevance in a context of RSL, this review lays out the role of antecedent structure in the deposition and preservation of these features; biological and geological indicators of sea level and the ranges of accuracy associated with them; benefits and limitations of various dating methods; and a useful comparison to younger (Holocene) and pre-MIS 5e sea-level records. The final section 'Future research directions' provides useful tips for topics that could merit additional work, based both on a thorough literature review and observations based on personal field experience.

**Thank you.**

I have only three very minor suggestions and one point to consider:

To further expand on the limitations of using marine terraces as indicators of RSL, considering repeated reoccupation by sea-level stillstands

**This is an excellent point, and we certainly have evidence on the Pacific Coast of repeated reoccupation of terraces (for example, the evidence from both U-series dating and faunas of mixes of ~100 ka (MIS 5c) and ~120 ka (MIS 5e) fossils on terraces at several localities. I allude to this in the "Introduction" (section 1), but I have added some text reminding readers of this now in the "Sea level indicators" (section 2).**

In Lines 35 and 40: the use of the word 'complex' had me thinking of an igneous intrusive complex, so perhaps just refer to 'the last interglacial'?

**Another good point, and after looking the manuscript over, I realized I never use this phrase again, so why use it here and complicate matters? I have eliminated it in the two lines you refer to and I follow your suggestion and simply call it "the last**

**interglacial". For clarification that I am talking about MIS 5.5 alone, I also added a line at the end of this paragraph as follows: "Some investigators also consider that MIS 5.5 alone is the last interglacial (*sensu stricto*)."**

Please re-run a check of abbreviations as I noted that some were expanded more than once in the main text (GIA, LIG, RSL).

**Good catch! I found a number of repeats and have now fixed them all. Thanks!**

I have no doubt that this paper will be well received by the community. Again, it was a pleasure to review.

**Thank you.**

Kind regards,

Hayley Cawthra
**Citation**: https://doi.org/10.5194/essd-2021-345-RC4